# Functional Decomposition and Shapley Interactions
# for Interpreting Survival Models

**Sophie Hanna Langbein** [1 2] **Hubert Baniecki** [3 4] **Fabian Fumagalli** [5 6]
**Niklas Koenen** [1 2] **Marvin N. Wright** [1 2] **Julia Herbinger** [1]

## Abstract

Hazard and survival functions are natural, interpretable targets in time-to-event prediction tasks such as patient survival and disease progression modeling, but their inherent non-additivity fundamentally limits standard additive explanation methods. We introduce Survival Functional Decomposition (SurvFD), a principled approach for analyzing feature interactions in machine learning survival models. By decomposing higher-order effects into time-dependent and time-independent components, SurvFD offers a previously unrecognized perspective on survival explanations, explicitly characterizing when and why additive explanations fail. Building on this theoretical decomposition, we propose SurvSHAP-IQ, which extends Shapley interactions to time-indexed functions, providing a practical estimator for higher-order, time-dependent interactions. We validate the framework on simulated data and demonstrate its utility through cancer survival applications, including multi-modal breast cancer prognosis combining histopathology with clinical features. Together, SurvFD and SurvSHAP-IQ establish an interaction- and time-aware interpretability approach for survival modeling, with broad applicability across medicine, healthcare and other time-to-event prediction tasks.

## 1. Introduction

Understanding whether effects vary across subgroups, patient characteristics, or co-exposures is critical for clinical and public health decisions. Such *interactions* are important in survival analysis, e.g., between genetic and environmental factors (Minelli et al., 2011), obesity and treatment effects (Hanai et al., 2014; Jensen et al., 2008), or age and tumor markers (Julkunen & Rousu, 2025; Nielsen & Grønbæk, 2008; Stehlik et al., 2010). Machine learning survival models can automatically capture complex, non-linear, and time-varying interactions without prior specification (Barnwal et al., 2022; Ishwaran et al., 2008; Wiegrebe et al., 2024), but their opacity limits interpretability and clinical utility (Baniecki et al., 2025b; Langbein et al., 2025).

A principled way to formalize feature interactions is functional decomposition (FD), which additively separates prediction functions into main and interaction effects (Hooker, 2004; 2007; Owen, 2013; Stone, 1994). In survival analysis, however, FD is more challenging because predictions are intrinsically time-dependent. A key limitation is that standard FD does not distinguish between *time-independent* and *time-dependent* effects. Moreover, we show that although additive decomposition is natural on the log-hazard scale, transformations to interpretable scales, such as hazard or survival functions, induce additional time-dependent effects and interactions not present on the log-hazard scale (cf. Cor. 3.4, Prop. 3.5). This motivates a principled FD approach for understanding time-dependence and interactions in survival models and the need to quantify interactions across scales and time (cf. Eq. (9)).

Recently, Shapley-based interaction indices have been used to estimate interactions based on FD (Bordt & von Luxburg, 2023; Fumagalli et al., 2023; Grabisch & Roubens, 1999; Sundararajan et al., 2020; Tsai et al., 2023), but are largely restricted to scalar outcomes, leaving survival-specific decompositions and their estimation underdeveloped. This prevents a systematic and theoretically grounded quantification of feature interactions in time-to-event modeling. We now review existing approaches and their limitations.

**Related work.** In statistical survival models, interactions are assessed via (1) *subgroup analyses* (effect modification, VanderWeele, 2009) or (2) explicit interaction (product) terms. Their interpretation is model-specific: interactions represent departures from multiplicativity in CoxPH models

[1]Leibniz Institute for Prevention Research and Epidemiology – BIPS [2]Faculty of Mathematics and Computer Science, University of Bremen [3]University of Warsaw [4]Centre for Credible AI, Warsaw University of Technology [5]LMU Munich, MCML [6]Bielefeld University. Correspondence to: Marvin N. Wright <wright@leibniz-bips.de>.

*Proceedings of the 43rd International Conference on Machine Learning*, Seoul, South Korea. PMLR 306, 2026. Copyright 2026 by the author(s).

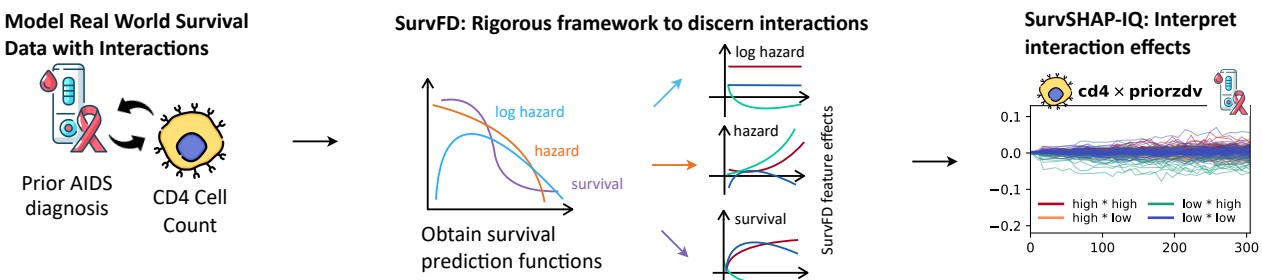

*Figure 1.* SurvFD and SurvSHAP-IQ facilitate the interpretation of interactions in survival models. (images: Flaticon.com)

and from additivity in additive hazard models (Aalen, 1980; Rod et al., 2012). In contrast, interaction analysis in machine learning survival models is limited. Existing local, model-agnostic explanation methods—such as SurvLIME (Kovalev et al., 2020), SurvSHAP(t) (Krzyziński et al., 2023), JointLIME (Chen et al., 2025), and GradSHAP(t) (Langbein et al., 2025)—do not capture interactions, leaving their detection and quantification largely unaddressed.

So far, FD has not been widely adopted in survival analysis. Huang et al. (2000) use *functional ANOVA* decomposition (Hooker, 2004) on the log-hazard, yielding flexible non-linear, time-dependent effects via splines. Mercadier & Ressel (2021) extend the Hoeffding–Sobol decomposition to homogeneous co-survival functions, decomposing joint survival for multiple events into main and interaction effects. Yet, none of these approaches provide a principled decomposition of survival models across time and prediction scales.

**Contributions.** Our work advances the literature in three ways: **(1) SurvFD.** We formalize functional decomposition for survival models (SurvFD), recovering additive, any-order interaction effects, split into time-dependent and time-independent components. We characterize SurvFD across log-hazard (Thms. 3.2 & 3.3), hazard, and survival functions (Prop. 3.5, Cor. 3.4) and its behavior under feature dependencies (Thm. 3.6). **(2) SurvSHAP-IQ.** Building on SurvFD, we extend Shapley interaction quantification to survival models. SurvSHAP-IQ provides estimates of higher-order interactions that can be visualized to interpret time-dependent survival predictions. **(3) Empirical validation.** We show that SurvSHAP-IQ accurately recovers interaction effects across several simulated prediction functions while satisfying local accuracy. We further demonstrate its utility for interpreting cancer survival models on multiple real-world datasets including multi-modal data.

## 2. Background

This section gives the necessary background on survival analysis, functional decomposition, and Shapley-based interpretation of survival models, as foundations for SurvFD theory and its practical implementation in SurvSHAP-IQ.

**General notation.** Let $\mathbb{D} = \{(\boldsymbol{x}^{(i)}, y^{(i)}, \delta^{(i)}) : i = 1, \ldots, n\}$ denote a survival dataset, where $\boldsymbol{x}^{(i)} = (x_1^{(i)}, \ldots, x_p^{(i)}) \in \mathcal{X}$ is the $p$-dimensional vector of *predictive features* for *individual i*. We let $\boldsymbol{X} = (X_1, \ldots, X_p)$ denote the corresponding random vector of features with support $\mathcal{X}$, and $X_j$ its $j$-th component. For each data point, $y^{(i)} = \min(t^{(i)}, c^{(i)})$ represents the observed time, defined as the minimum of the true event time $t^{(i)} \in \mathbb{R}_0^+$ and the censoring time $c^{(i)} \in \mathbb{R}_0^+$. The binary event indicator $\delta^{(i)} \in \{0, 1\}$ equals 1 if the event occurs ($t^{(i)} < c^{(i)}$) and 0 if the observation is censored ($t^{(i)} > c^{(i)}$). We consider a single time-to-event setting with one event type (the event of interest), excluding competing and recurrent events.

### 2.1. Survival Analysis

Survival analysis aims to characterize a set of functions mapping combinations of values from the feature space $\mathcal{X}$ and the time domain $\mathcal{T}$ to a scalar outcome. In the following, we consider the hazard function $h$ and survival function $S$.

**Definition 2.1** (Hazard function). The **hazard** (aka **risk**) **function** $h : \mathcal{X} \times \mathcal{T} \to \mathbb{R}_0^+$ gives the instantaneous event risk at time $t$, conditional on survival up to $t$ and observed features $\boldsymbol{x} \in \mathcal{X}$:

$$h(t|\boldsymbol{x}) := \lim_{\Delta t \to 0} \frac{\mathbb{P}(t \leq T \leq t + \Delta t | T \geq t, \boldsymbol{x})}{\Delta t}. \quad (1)$$

**Definition 2.2** (Survival function). The **survival function** $S : \mathcal{X} \times \mathcal{T} \to [0, 1]$ describes the probability of the time-to-event being at least $t \geq 0$ conditional on the observed features $\boldsymbol{x} \in \mathcal{X}$:

$$S(t|\boldsymbol{x}) := \mathbb{P}(T \geq t|\boldsymbol{x}) = \exp\left(-\int_0^t h(u|\boldsymbol{x})du\right). \quad (2)$$

A common choice for the hazard function is the general multiplicative hazard model (Oakes, 1977):

$$h(t|\boldsymbol{x}) = h_0(t) \exp\big(G(t|\boldsymbol{x})\big), \quad (3)$$

where $h_0(t)$ is a baseline hazard and $G(t|\boldsymbol{x})$ a (possibly time-dependent) *risk score* linking features $\boldsymbol{x}$ to the hazard.

A well-known special case is the *Cox proportional hazards (CoxPH) model* (Cox, 1972), which usually assumes $G(t|\boldsymbol{x}) = \boldsymbol{x}^\top \boldsymbol{\beta}$ with $\boldsymbol{\beta} \in \mathbb{R}^p$, i.e., the model considers neither interactions between features nor time-dependent effects, which implies constant hazard ratios over time — the *proportional hazards assumption*.

In this work, we consider a generalized version of the risk score incorporating potentially time-dependent non-linear and interaction terms of arbitrary order

$$G(t|\boldsymbol{x}) = \sum_{M \subseteq P} \beta_M \prod_{j \in M} g_j(x_j)\, l_j(t),$$

where $M \subseteq P = \{1, \ldots, p\}$ indexes feature subsets, $\beta_M \in \mathbb{R}$ denotes the associated main or interaction effect, $g_j : \mathbb{R} \to \mathbb{R}$ is a fixed (non-linear) feature transformation, and $l_j(t)$ captures time dependence (possibly constant).

## 2.2. Functional Decomposition

Functional decomposition (FD) expresses a model's prediction function in terms of main and interaction effects over all feature subsets $M \subseteq P \coloneqq \{1, \ldots, p\}$, with complement $\bar{M} \coloneqq P \setminus M$. Any square-integrable function $F : \mathcal{X} \to \mathbb{R}$ on a $p$-dimensional feature space admits the following decomposition:

$$F(\boldsymbol{x}) = f_\emptyset + \sum_{\emptyset \neq M \subseteq P} f_M(\boldsymbol{x}), \qquad (4)$$

where $f_\emptyset \coloneqq \int F(\boldsymbol{x})\, d\mathbb{P}_{\boldsymbol{X}}$ and $f_M$ is the *pure effect* of subset $M$, defined via the inclusion–exclusion principle (Hoeffding, 1948; Hooker, 2004; 2007):

$$f_M(\boldsymbol{x}) \coloneqq \int F(\boldsymbol{x})\, d\mathbb{P}_{\boldsymbol{X}_{\bar{M}}} - \sum_{Z \subset M} f_Z(\boldsymbol{x}). \quad (5)$$

Here, $\mathbb{P}_{\boldsymbol{X}_{\bar{M}}}$ denotes a reference distribution over the complementary features $\boldsymbol{X}_{\bar{M}}$, whose choice uniquely determines the FD. The resulting $f_M$ represents a *pure main effect* when $|M| = 1$ and a *pure interaction effect* when $|M| \geq 2$. Different choices of reference distribution yield different FDs with distinct properties and interpretations. Following Fumagalli et al. (2025), we distinguish between marginal and conditional FD.

**Marginal FD vs. conditional FD.** Marginal FD is defined via the joint marginal distribution $\mathbb{P}_{\boldsymbol{X}_{\bar{M}}} = \mathbb{P}(\boldsymbol{X}_{\bar{M}})$, ignoring dependencies between $\bar{M}$ and $M$, whereas conditional FD integrates over the joint conditional distribution $\mathbb{P}_{\boldsymbol{X}_{\bar{M}}} = \mathbb{P}(\boldsymbol{X}_{\bar{M}}|\boldsymbol{X}_M = \boldsymbol{x}_M)$ to account for such dependencies. Marginal FD is unique (Fumagalli et al., 2025) and reduces to functional ANOVA (Hoeffding decomposition) under feature independence, yielding orthogonal components (Hoeffding, 1948; Hooker, 2004). Conditional FD is unique but enforcing hierarchical orthogonality is generally costly (Chastaing et al., 2015; Hooker, 2007; Rahman,

2014). Marginal FD underpins common model-centric explanations such as partial dependence (Friedman, 2001), permutation feature importance (Fisher et al., 2019), and interventional SHAP (Lundberg & Lee, 2017), which are "true to the model". In contrast, conditional FD yields interpretations "true to the data" (Chen et al., 2020), underpinning methods such as conditional feature importance (Strobl et al., 2008) and observational SHAP (Olsen et al., 2022).

## 2.3. Shapley Values for Survival Models

Our goal in this work is to interpret interactions in survival model predictions. To quantify the attribution of feature $j \in P$ for instance $\boldsymbol{x} \in \mathcal{X}$, we define a time-dependent value function $\nu : \mathcal{P}(P) \times \mathcal{T} \to \mathbb{R}$, where $\mathcal{P}(P)$ is the power set of $P$. In the machine learning setting, the value function corresponds to model predictions for a feature subset $M$, integrated over a reference distribution $\mathbb{P}_{\boldsymbol{X}_{\bar{M}}}$ of the remaining features $\bar{M} \coloneqq P \setminus M$ at a particular timepoint $t$:
$\nu(t|M) \coloneqq \mathbb{E}\left[S(t|\boldsymbol{X})|\boldsymbol{X}_M = \boldsymbol{x}_M\right] - \mathbb{E}\left[S(t|\boldsymbol{X})\right].$

Following Krzyziński et al. (2023), the time-dependent Shapley value for feature $j$ is defined as

$$\phi_j(t|\boldsymbol{x}) = \sum_{M \subseteq P \setminus \{j\}} w_M \times \Big(\nu(t|M \cup \{j\}) - \nu(t|M)\Big). \quad (6)$$

with $w_M = [(p - |M| - 1)!\, |M|!]/p!$. This formulation satisfies the Shapley axioms (Lundberg & Lee, 2017) for each timepoint $t \in \mathcal{T}$: (1) **symmetry**, i.e., attributions are invariant to feature ordering; (2) **linearity**, i.e., attributions are linear in $\nu(t|\cdot)$; (3) **dummy**, i.e., features with no effect on $\nu(t|\cdot)$ receive zero attribution; and (4) **efficiency/local accuracy**, i.e., attributions sum to $S(t|\boldsymbol{x}) - \mathbb{E}\left[S(t|\boldsymbol{X})\right]$.

As discussed in Sec. 2.2, this definition corresponds to interventional SHAP when the value function uses the joint marginal distribution and to observational SHAP when it uses the joint conditional distribution; both are defined via marginal and conditional FD (Bordt & von Luxburg, 2023). These Shapley values quantify only individual feature attributions in the survival context, not explicit feature interactions, which we address in the following section.

## 3. Methodology

We first introduce a general class of survival prediction functions that defines *ground-truth* time-dependent and time-independent interaction structures in Sec. 3.1. We then propose *SurvFD*, a functional decomposition for survival prediction functions in Sec. 3.2 and formalize under which assumptions SurvFD recovers ground-truth interactions of the log-hazard as well as analyze the effects of transformations to alternative prediction functions and feature dependencies. Finally, we propose an estimation method for interaction effects up to a pre-specified order in Sec. 3.3.

## 3.1. Ground-Truth Assumptions

We assume the functional relationships between survival and hazard functions to risk score $G$ given in Eqs. (2) and (3). Since time dependence may arise only for certain feature subsets, the power set $\mathcal{P}(P)$ is partitioned into time-dependent subsets $\mathcal{I}_d$ and time-independent subsets $\mathcal{I}_{id}$. Assuming (generally unknown) ground-truth partitions satisfying $\mathcal{I}_d \cap \mathcal{I}_{id} = \emptyset$ and $\mathcal{I}_d \cup \mathcal{I}_{id} = \mathcal{P}(P)$, the ground-truth function $G : \mathcal{X} \times \mathcal{T} \to \mathbb{R}$ admits the following generalized additive representation:

$$G(t|\boldsymbol{x}) = \sum_{M \in \mathcal{I}_d} g_M(t|\boldsymbol{x}) + \sum_{M \in \mathcal{I}_{id}} g_M(\boldsymbol{x}). \quad (7)$$

This generalizes the CoxPH model to allow for time-dependent, non-linear, and interaction effects.

Thus, the *log-hazard function* yields the ground-truth decomposition into time-dependent and time-independent effects with baseline hazard $b(t) = \log h_0(t)$

$$\log h(t|\boldsymbol{x}) = b(t) + \sum_{M \in \mathcal{I}_d} g_M(t|\boldsymbol{x}) + \sum_{M \in \mathcal{I}_{id}} g_M(\boldsymbol{x}). \quad (8)$$

Formally, we distinguish **time-dependent** vs. **time-independent effects** $g_M$ for a feature set $M \subseteq P$:

**Time-dependent:** $M \in \mathcal{I}_d$, if $g_M(t|\boldsymbol{x})$ varies with $t$, i.e., $\exists t_1 \neq t_2 : g_M(t_1|\boldsymbol{x}) \neq g_M(t_2|\boldsymbol{x})$ for some $\boldsymbol{x} \in \mathcal{X}$.

**Time-independent:** $M \in \mathcal{I}_{id}$, if $g_M(t|\boldsymbol{x})$ does not depend on $t$, i.e., $\forall t_1 \neq t_2 : g_M(t_1|\boldsymbol{x}) = g_M(t_2|\boldsymbol{x})$ holds for all $\boldsymbol{x} \in \mathcal{X}$. In this case, we write $g_M(t|\boldsymbol{x}) = g_M(\boldsymbol{x})$. By definition, this also includes non-influential feature sets (i.e., those with $g_M \equiv 0$).

## 3.2. Functional Decomposition for Survival (SurvFD)

Understanding feature effects on survival often requires separating *time-dependent* from *time-independent* attributions. Therefore, we generalize the FD definition of Eq. (4) to survival prediction functions.

**Definition 3.1** (SurvFD). Let $F : \mathcal{X} \times \mathcal{T} \to \mathbb{R}$ be a square-integrable survival prediction function on a $p$-dimensional feature space $\mathcal{X}$ and time domain $\mathcal{T}$. Its decomposition into **pure effects** is

$$F(t|\boldsymbol{x}) = f_\emptyset(t) + \sum_{\emptyset \neq M \subseteq P} f_M(t|\boldsymbol{x}) \quad (9)$$

$$= f_\emptyset(t) + \sum_{M \in \mathcal{I}_d^\star} f_M(t|\boldsymbol{x}) + \sum_{M \in \mathcal{I}_{id}^\star} f_M(\boldsymbol{x}),$$

where $\mathcal{I}_d^\star$ and $\mathcal{I}_{id}^\star$ denote the sets of feature subsets with time-dependent and time-independent effects, respectively, satisfying $\mathcal{I}_d^\star \cap \mathcal{I}_{id}^\star = \emptyset$ and $\mathcal{I}_d^\star \cup \mathcal{I}_{id}^\star = \mathcal{P}(P)$. Pure effects $f_M$ are defined analogous to Eq. (5).

In practice, the true partitions $\mathcal{I}_d$ and $\mathcal{I}_{id}$ are unknown. SurvFD provides an additive decomposition of any survival prediction function into time-dependent and time-independent main and interaction effects via the sets $\mathcal{I}_d^\star$ and $\mathcal{I}_{id}^\star$. While such a decomposition always exists, it is not unique and depends on the prediction target and model form. We now show in which cases SurvFD recovers the ground-truth decomposition of Eq. (8) and how prediction target transformations and feature correlations influence it.

### 3.2.1. SURVFD WITH INDEPENDENT FEATURES

We begin with the simplest case of independent features. In this setting, marginal and conditional FD coincide with the *functional ANOVA decomposition* (Hoeffding, 1948; Hooker, 2004), as the reference distribution $\mathbb{P}_{\boldsymbol{X}_{\bar{M}}}$ reduces to the product of the marginal distributions in both cases. We derive how the FD in Def. 3.1 differs depending on the survival prediction function.

**Log-hazard function.** The log-hazard in Eq. (8) admits an additive decomposition into ground-truth time-dependent and time-independent components. Under the following assumptions, this decomposition coincides with the SurvFD representation in Eq. (9).

**Theorem 3.2.** *Let* $\log h(t|\boldsymbol{x})$ *be given as in Eqs.* (7) *and* (8) *with ground-truth sets* $\mathcal{I}_d$ *and* $\mathcal{I}_{id}$. *Assume that features are mutually independent. If either*

(i) $G(t|\boldsymbol{x})$ *is linear in* $\boldsymbol{x}$ *including interactions, or*

(ii) $G(t|\boldsymbol{x})$ *is an additive main effect model,*

*then for SurvFD in Eq.* (9): $\mathcal{I}_d^\star = \mathcal{I}_d$ *and* $\mathcal{I}_{id}^\star = \mathcal{I}_{id}$.

For general log-hazard functions, SurvFD in Eq. (9) may not exactly recover ground-truth effects: a single time-dependent effect in the ground truth $G$ can make lower-order time-independent subsets appear time-dependent, while higher-order supersets remain time-independent.

**Theorem 3.3.** *Let* $G(t|\boldsymbol{x}) = g_Z(t|\boldsymbol{x}) + \sum_{M \in \mathcal{I}_{id}} g_M(\boldsymbol{x})$ *and* $\mathcal{I}_d = \{Z\}$, *with* $Z$, $2 \leq |Z| < p$, *being the only time-dependent set in* $G$, *and assume features are mutually independent. Then for SurvFD of* $\log h(t|\mathbf{x})$:
1. *For any* $L \subset Z$, *it may occur that* $L \in \mathcal{I}_d^\star$ *while* $L \in \mathcal{I}_{id}$ *(downward propagation).*
2. *For any* $L \supset Z$ *with* $L \in \mathcal{I}_{id}$, *it holds that* $L \in \mathcal{I}_{id}^\star$ *(no upward propagation).*

**Hazard and survival function.** The exponential transformation from the log-hazard to the hazard induces a multiplicative form. Applying SurvFD to $h(t|\boldsymbol{x})$ and $S(t|\boldsymbol{x})$ entangles effects, making it harder to separate time-dependent from time-independent components, reducing consistency with the log-hazard and causing both downward and upward propagation of time-dependent effects.

**Corollary 3.4.** *Let* $G(t|\boldsymbol{x}) = g_Z(t|\boldsymbol{x}) + \sum_{M \in \mathcal{I}_{id}} g_M(\boldsymbol{x})$ *and* $\mathcal{I}_d = \{Z\}$, *with* $Z$, $2 \leq |Z| < p$, *being the only time-dependent set in* $G$, *assuming features are mutually independent. Then, for the SurvFD of* $h(t|\boldsymbol{x})$ *and* $S(t|\boldsymbol{x})$ *(Eqs. (1) and (2)), subsets or supersets of* $Z$ *may appear in* $\mathcal{I}_d^\star$ *while belonging to* $\mathcal{I}_{id}$ *(downward and upward propagation).*

Moreover, the non-linear transforms to hazard and survival naturally induce additional feature interactions, underscoring the need to quantify them.

**Proposition 3.5.** *Let* $G(t|\boldsymbol{x}) = \boldsymbol{x}^\top \boldsymbol{\beta}$ *be a CoxPH model with mutually independent features and* $\beta_j \neq 0$ *for all* $j$. *With* $h(t|\boldsymbol{x})$ *and* $S(t|\boldsymbol{x})$ *defined as in Eqs. (1) and (2), the SurvFD of* $h(t|\boldsymbol{x})$ *and of* $S(t|\boldsymbol{x})$ *exhibits interaction effects.*

### 3.2.2. SURVFD WITH DEPENDENT FEATURES

When features are dependent, marginal and conditional FD differ. Marginal FD integrates over the marginal distribution $\mathbb{P}(\mathbf{X}_{\bar{M}})$ and thus breaks the association between features in $M$ and $\bar{M}$, ignoring correlations and reflecting the model rather than the data. Conditional FD accounts for feature dependencies through the conditional distribution $\mathbb{P}(\mathbf{X}_{\bar{M}}|\mathbf{X}_M = \mathbf{x}_M)$, providing a decomposition that better reflects the data. Consequently, for the log-hazard, marginal FD deviates from the ground truth only if the model learned the effects, whereas for conditional FD, even non-influential features can appear as influential time-dependent effects.

**Theorem 3.6.** *Let* $G(t|\boldsymbol{x})$ *be as in Eq. (7). Let* $x_j$ *with* $j \in \mathcal{I}_{id}$ *be a feature with no direct effect on* $G$, *and assume* $x_j$ *depends on a time-dependent feature* $x_k$ *with* $k \in \mathcal{I}_d$. *Then the marginal SurvFD component of* $\log h(t|\boldsymbol{x})$ *is zero*

$$\mathbb{P}(\mathbf{X}_{\{\bar{j}\}}) = \mathbb{P}(\boldsymbol{X}) \implies f_{\{j\}}^{\mathrm{marginal}}(t|\boldsymbol{X}) \equiv 0.$$

*while the conditional SurvFD component is non-zero*

$$\mathbb{P}(\boldsymbol{X}_{\{\bar{j}\}}|X_j = x_j) \neq \mathbb{P}(\boldsymbol{X}_{\{\bar{j}\}}) \implies f_{\{j\}}^{\mathrm{cond.}}(t|\boldsymbol{X}) \not\equiv 0.$$

It follows that while the independent feature case allows for identical interpretations between marginal and conditional FD, for both independent and dependent features the multiplicative transformations of the hazard and survival functions may lead to time-dependent effects and interactions that differ from the ground-truth log-hazard decomposition. For dependent features, the resulting decomposition additionally depends on the chosen reference distribution, depending on whether the primary interest is in interpreting the model or the data.

### 3.3. Shapley Interactions for Survival (SurvSHAP-IQ)

Building on SurvFD, we introduce **SurvSHAP-IQ, the first Shapley interaction quantification method for time-dependent survival outcomes**. SurvSHAP-IQ extends Shapley interaction quantification (Bordt & von Luxburg, 2023) to time-indexed survival predictions and provides a practical estimator of SurvFD (interaction) components up to an *explanation order* $k = 1, \ldots, p$.

**Definition 3.7** (SurvSHAP-IQ decomposition)**.** At any fixed timepoint $t$, Shapley interactions with $k = 2$ decompose a survival prediction function $F(t|\boldsymbol{x})$ additively into constant, first and second-order components as

$$F(t|\boldsymbol{x}) = \phi_\emptyset^{(2)}(t) + \underbrace{\sum_{j=1}^p \phi_{\{j\}}^{(2)}(t|\boldsymbol{x})}_{\text{individual effects}} + \underbrace{\sum_{\{i,j\}:i \neq j} \phi_{\{i,j\}}^{(2)}(t|\boldsymbol{x})}_{\text{interaction effects}},$$

and more generally for arbitrary $k$-th order, as

$$F(t|\boldsymbol{x}) = \sum_{M \subseteq P:|M| \leq k} \phi_M^{(k)}(t|\boldsymbol{x}),$$

where $\phi_{\{i\}}^{(k)}(t|\boldsymbol{x})$ are the individual effects, and $\phi_M^{(k)}(t|\boldsymbol{x})$ with $|M| \geq 2$ the interaction effects of order $|M|$.

For $k = 1$, Shapley interactions reduce to the time-dependent Shapley value $\phi_{\{i\}}^{(1)}(t|\boldsymbol{x}) = \phi_i(t|\boldsymbol{x})$ (Eq. (6)), recovering standard feature attributions. For $k > 1$, they yield a finer-grained decomposition that explicitly quantifies feature interactions in survival predictions. To obtain an axiomatic interaction estimator, we adopt the n-Shapley values (Bordt & von Luxburg, 2023), which are grounded in the Shapley interaction index (Grabisch & Roubens, 1999). The n-Shapley values are constructed such that the top-order coefficients exactly coincide with the Shapley interaction index, i.e., for $|K| = k$ it holds

$$\phi_K^{(k)}(t|\boldsymbol{x}) = \sum_{M \subseteq P \setminus K} \frac{1}{(p-k+1)\binom{p-k}{|M|}} \Delta_K(M), \quad (10)$$

where $\Delta_K(M)$ denotes the *discrete derivative*

$$\Delta_K(M) := \sum_{L \subseteq K} (-1)^{|K|-|L|} \nu(t|M \cup L).$$

Following Sec. 2.3, the value function $\nu : \mathcal{P}(P) \times \mathcal{T} \to \mathbb{R}$ is defined at timepoint $t \in \mathcal{T}$ as the expected prediction over a reference distribution of the complement features $\bar{M}$,

$$\nu(t|M) := \mathbb{E}_{\boldsymbol{X}_{\bar{M}}}[F(t|\boldsymbol{X})|\boldsymbol{X}_M = \boldsymbol{x}_M] - \mathbb{E}_{\boldsymbol{X}}[F(t|\boldsymbol{X})],$$

where the reference distribution is either the joint marginal $\mathbb{P}(\boldsymbol{X}_{\bar{M}})$ (marginal/interventional variant) or the joint conditional $\mathbb{P}(\boldsymbol{X}_{\bar{M}}|\boldsymbol{X}_M = \boldsymbol{x}_M)$ (conditional/observational variant), in line with the marginal vs. conditional FD distinction in Sec. 2.2. For $|K| = 1$ this reduces to the standard marginal contribution $\Delta_{\{i\}}(M) = \nu(t|M \cup \{i\}) - \nu(t|M)$, recovering the Shapley value (Eq. (6)). In App. A.6, we

further show how $\Delta_{\{i\}}(M)$ relates to the SurvFD decomposition in Eq. (9).

Following the marginal vs. conditional FD distinction (Sec. 2.2), the two choices of the value function $\nu$ (underlying Eq. (10)) yield interventional or observational Shapley interaction explanations, respectively. Here, we focus on the **marginal (interventional)** variant, prioritizing **faithfulness to the model** over the data distribution. While the SurvFD decomposition directly separates time-dependent from time-independent effects, SurvSHAP-IQ provides complementary insight by quantifying **feature-level interaction effects** and returning an attribution vector **evaluated at selected timepoints** $t \in \mathcal{T}$. The distinction between time-dependent ($\mathcal{I}_d^\star$) and time-independent effects ($\mathcal{I}_{id}^\star$) can then be recovered by visualizing these attributions over time (see Sec. 4).

**Implementation and evaluation.** Computing Shapley interactions for survival models requires an exponential number of evaluations of $\nu$ for selected timepoints $t \in \mathcal{T}$, which quickly becomes infeasible in practice. To address this, we extend existing *SHAP-IQ* approximation methods (Fumagalli et al., 2023; 2024; Muschalik et al., 2024) to the survival setting, yielding a **practical implementation** of *SurvSHAP-IQ*. To evaluate the accuracy of the methods, we use the time-dependent adaptation of the *local accuracy* measure (Krzyziński et al., 2023), extending it to interactions. This metric quantifies the decomposition error, i.e., the difference between an individual's survival prediction and the dataset average at a given timepoint (see Eq. (B35)).

## 4. Empirical Validation

The detailed validation setup is described in App. B, and the code for reproducing all results is available on GitHub.

### 4.1. Experiments with Simulated Data and Risk Scores

The purpose of the simulated experiments is twofold: (1) to validate the theoretical FD results for various survival functions using SurvSHAP-IQ as its practical implementation, and (2) to demonstrate that SurvSHAP-IQ can accurately decompose both ground-truth and model prediction functions, thereby facilitating local interpretation.

**Setup.** Corresponding to Sec. 3.2, we consider increasingly complex model structures for the risk score $G(t|\boldsymbol{x})$ with a focus on different interactions. For clarity, we restrict the analysis to second-order interactions. Ten simulation scenarios are constructed, shown in Fig. 2, with a linear $G(t|\boldsymbol{x})$ in a purple block, and a generalized additive (gen. add.) $G(t|\boldsymbol{x})$ in an orange block, serving as baselines. To each baseline, time-feature (in pink), feature-feature (in green), and/or time-feature-feature interactions (in blue) are added systematically. We simulate data for all scenarios by generating $n = 1{,}000$ observations per experiment,

*Figure 2.* Ten simulation scenarios for the risk score $G(t|\boldsymbol{x})$, ranging from linear (top, purple block) to generalized additive (bottom, orange block) baselines. To each baseline, time-dependent main effects (pink), feature–feature interactions (green), and time-feature–feature interactions (blue) are added systematically.

with event times drawn from standard exponential (non-)PH models specified by Eq. (3). The three features $x_1, x_2, x_3$ are sampled from $\mathcal{N}(0, 1)$, with coefficients $h_0(t) = 0.03$, $\beta_1 = 0.4$, $\beta_2 = -0.8$, $\beta_3 = -0.6$, $\beta_{12} = -0.5$, and $\beta_{13} = 0.2$. The maximum follow-up time is $t = 70$. Crucially, such a setup satisfies the basic assumptions required for the SurvFD results in Sec. 3.1. For each setting, the exact order-2 SurvSHAP-IQ decomposition is computed (cf. Eq. (10)) using the ground-truth log-hazard, hazard, and survival functions on the full dataset. The resulting attributions are plotted over time for a single randomly selected observation $[x_1 = -1.265, x_2 = 2.416, x_3 = -0.644]$, and Savitzky-Golay smoothing is applied to obtain smooth curves. We note that the choice of observation is arbitrary and leads to the same conclusions w.r.t. the SurvFD results.

**Ground-truth decompositions.** The top- and bottom-left plots (scenario (**9**) and (**10**)) in Figure 3 show SurvSHAP-IQ attributions of the ground-truth log-hazard function that validate Thm. 3.2 and Thm. 3.3. For $G(t|\boldsymbol{x})$ with one time-dependent main effect and otherwise time-independent main and interaction effects (scenario (**9**), top-left plot), the pure SurvFD effects match the additive components of $G(t|\boldsymbol{x})$. That is, it shows time-constant effects for all time-independent components ($x_2, x_3, x_1x_2, x_1x_3$), zero attribution for the absent interaction ($x_2x_3$), and a time-varying effect for the time-dependent $x_1$. In contrast, when $G(t|\boldsymbol{x})$ includes a time-dependent interaction (scenario (**10**), bottom-left), the pure SurvFD effects do not necessarily correspond to the components of $G(t|\boldsymbol{x})$. In this example, $x_1$, which is time-independent in $G(t|\boldsymbol{x})$, retains part of the time-dependent effect of $x_1x_3$ (cf. Thm. 3.3).[1]

---

[1]Note that the attributions of time-independent features and interactions remain constant over time for both the log-hazard and hazard functions, which is the case for a time-constant baseline hazard only. In such cases, they can be aggregated into a single value, conceptually corresponding to the hazard ratio in CoxPH.

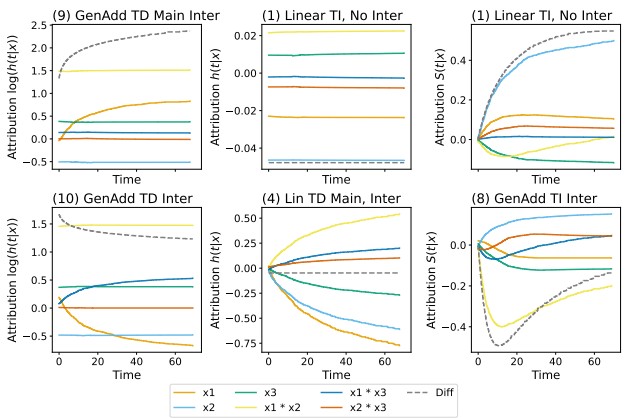

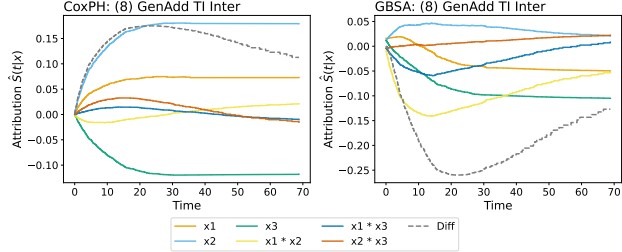

*Figure 4.* Exact SurvSHAP-IQ attribution curves for a selected observation computed on the predicted survival functions of CoxPH (**left**) and GBSA (**right**). For complete results see Fig. B.4 & B.5.

*Figure 3.* Exact SurvSHAP-IQ attribution curves for a selected observation computed on the ground-truth log-hazard (**left**), hazard (**middle**), and survival function (**right**). The difference between individual ground-truth values and the dataset average is plotted as a grey dashed line. The full results are shown in Figures B.1-B.5.

In Figure 3 (scenario **(1)**, top-middle-right plots), we see that even a linear $G(t|\boldsymbol{x})$ without interactions can yield non-zero attributions for higher-order feature interactions in the ground-truth hazard and survival functions (e.g., $x_1 x_2$). This is consistent with Prop. 3.5, which implies that even if $G(t|\boldsymbol{x})$ is solely defined by linear main effects, it can still produce non-zero higher-order effects in the SurvFD due to the non-linear transformations of the hazard and survival functions. The same holds for more complex scenarios with feature interactions and time-dependent main effects (scenario **(4)**, bottom-middle) or non-linear feature and interaction effects (scenario **(8)**, bottom-right), additionally highlighting that ground-truth time-independent effects may appear time-dependent in the SurvFD (corresp. Cor. 3.4). Further experiments in App. B.2 show the influence of feature dependencies on the SurvFD decomposition and how the decomposition varies depending on the chosen reference distribution including an empirical validation of Thm. 3.6.

**Decomposing model predictions.** Next, we fit a gradient boosting survival analysis (GBSA) model and a CoxPH model to the simulated training data, and compute SurvSHAP-IQ attributions for the survival functions on test data. Figure 4 shows an example for scenario **(8)**, where we defer the remaining results for all scenarios to App. B.1. The more flexible GBSA model fits the ground-truth generalized additive $G(t|\boldsymbol{x})$ with interactions better than the CoxPH without pre-specified interactions (C-index 0.70 vs. 0.66; IBS 0.15 vs. 0.16). This is further highlighted by the GBSA attribution curves being similar to those of the ground-truth survival attributions for scenario **(8)** (see Fig. 3).

**Local accuracy.** We compute the average local accuracy over time (Eq. (B35)) for the full datasets across all ten scenarios. For the ground-truth decompositions, **local accu-**

racy is consistently below 0.001 for survival, and below 0.00001 for hazard and log-hazard – it is slightly higher for survival due to approximation using the hazard. For the predicted survival functions from the CoxPH and GBSA models, the value remains below 0.015, **indicating near-perfect decomposition quality** (see Appendix Tab. B.3 and Fig. B.6 for complete results).

### 4.2. Experiments with Real-world Applications

We evaluate our methodology on real-world survival models without multiplicative hazard assumptions, using datasets from SurvSet (Drysdale, 2022) as well as an eye cancer study (Donizy et al., 2022) and the multi-modal TCGA-BRCA dataset (Lingle et al., 2016). Additional results are provided in Appendix B.3 and B.4.

**Gradient boosting survival analysis (GBSA).** We first analyze a GBSA model that predicts the survival time of patients diagnosed with uveal melanoma. It is trained with 100 trees of depth four on nine numerical features, achieving a validation C-index of 0.758 (Donizy et al., 2022). Figure 5 shows first- and second-order explanations for the validation set ($n = 77$), which is extended in Fig. B.11. Age, maximum tumor diameter, and mitotic rate emerge as the key predictors, as indicated by their high average absolute attribution values—consistent with established prognostic factors for uveal melanoma (Donizy et al., 2022).

The first-order Shapley values for age (Fig. 5, top-left) exhibit both positive and negative attributions across individuals, with the effect on the survival prediction tending to be positive at lower ages (blue) and negative at higher ages (red). Moving to second-order explanations (middle row) isolates the pure main effects, which exhibit substantially lower within-group variance. The variation has been effectively redistributed to the interaction panels (bottom row). This illustrates a general property of higher-order Shapley interactions that motivates SurvSHAP-IQ: **interactions absorbed into first-order attributions can manifest as spurious heterogeneity**, which higher-order decompositions disentangle.

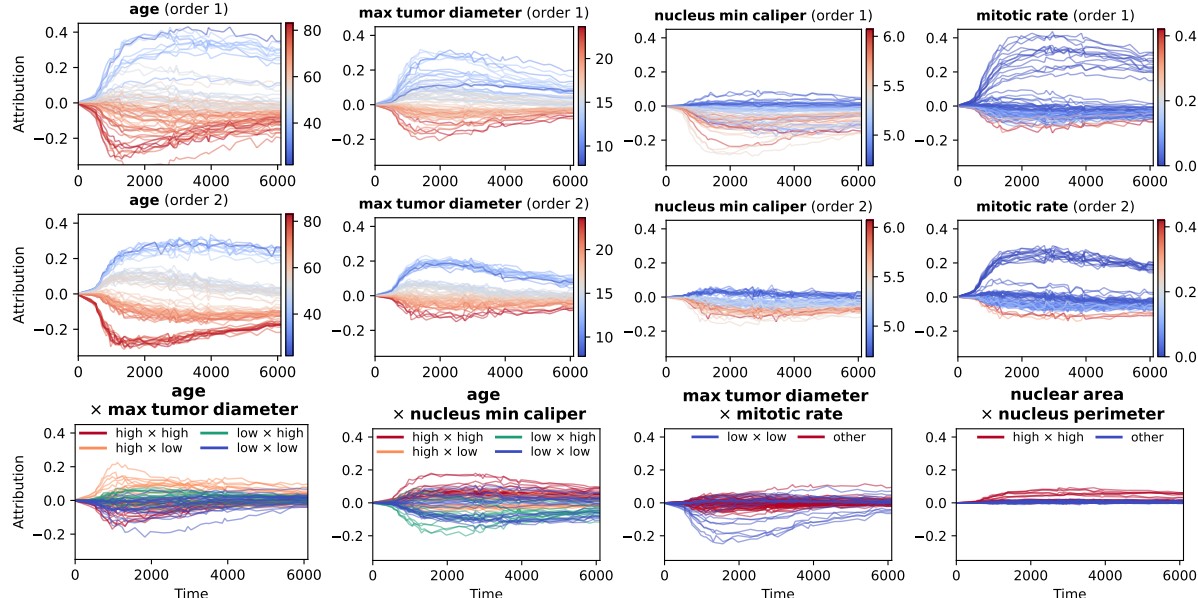

*Figure 5.* Explanation of a GBSA model predicting the survival of patients diagnosed with uveal melanoma. **Top:** Shapley attributions (order 1) for multiple observations (curves) with different feature values (in color). **Middle:** SurvSHAP-IQ individual effects (order 2). We observe how adding interaction terms decreases the in-group variance of attributions. **Bottom:** SurvSHAP-IQ interaction effects (order 2), where color denotes four groups of observations. Attributions are zero at $t = 0$ since $S(0|\boldsymbol{x}) = 1$ for all individuals, so $S(0|\boldsymbol{x}) - \mathbb{E}[S(0|\boldsymbol{X})] = 0$ and local accuracy enforces zero attributions. The remaining attribution effects are shown in Figure B.11.

SurvSHAP-IQ further uncovers strong pairwise interactions, enabling a finer-grained interpretation of survival predictions. While low maximum tumor diameter and low mitotic rate each individually increase predicted survival, their interaction effect is negative (Fig. 5, bottom-row), indicating that the joint contribution is smaller than the sum of the individual effects. This suggests that the two features carry partially redundant prognostic information in the learned model, which is consistent with both quantifying the same underlying disease burden (tumor size and proliferative activity). Such a joint structure is invisible to first-order attributions and demonstrates the practical value of higher-order explanations for survival models.

**Multi-modal TCGA-BRCA.** Second, we analyze the TCGA-BRCA dataset to predict overall survival for 990 breast cancer patients using histopathological whole-slide images (WSIs) and eight clinical features (e.g., age, surgery type, tumor stage, menopausal status). WSIs are provided as pre-extracted patches at $20\times$ magnification and encoded into 1,536-dimensional embeddings using the pre-trained vision transformer UNI2-h (Chen et al., 2024). Due to the large and varying number of patches per patient (523 on average), we apply a gated multiple-instance learning attention mechanism to weight patches before aggregation (Ilse et al., 2018). The resulting image representation is fused with clinical features and passed to a *DeepHit* head (Lee et al., 2018) to predict the probability mass function (PMF) of the event at timepoint $t$ (i.e., $\mathbb{P}(T = t|\boldsymbol{x})$) from which the

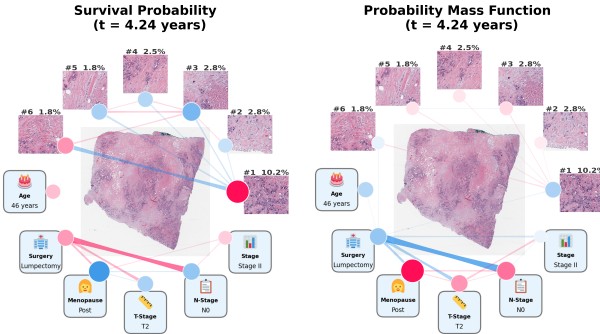

*Figure 6.* SurvSHAP-IQ attributions for a multi-modal survival deep learning model predicting a patient's (ID: TCGA-A7-A13D) survival probability (**left**) and probability mass function (**right**) at $t = 4.24$ years. Node size represents individual feature effects, edges indicate pairwise interaction effects between patches of histopathological WSI and clinical features.

discrete-time survival probabilities $S(t|\boldsymbol{x})$ can be computed. For SurvSHAP-IQ, we use the top six patches by attention weight, along with the eight clinical features, as players.

Figure 6 shows SurvSHAP-IQ network explanations for a single patient (ID: TCGA-A7-A13D) at $t = 4.24$ years, for survival probability (left) and PMF (right). Node sizes indicate individual attribution strength, and edges represent pairwise interactions; red denotes positive effects (increased survival or risk-accelerating PMF), while blue denotes negative effects (decreasing survival or risk-delaying PMF).

For the survival probability, the top-ranked patches by attention weights receive high attributions, including within-modality interactions. Among clinical features, the model reveals a strong interaction between node-negative status (N0) and lumpectomy: their joint effect on the survival prediction is favorable, recovering a clinically established relationship (Amin et al., 2017; Fisher et al., 2002). Interactions occur exclusively within, rather than across, modalities. This absence of cross-modal interactions is noteworthy, given that the model already includes tabular-image interaction terms at the encoding level before passing them to the DeepHit head; whether their limited contribution reflects the model's learned representation or a genuine biological independence remains an open question for further investigation. The PMF exhibits opposite effect directions, consistent with its negative relationship to survival, but the interaction patterns differ, particularly among image features, where some pairwise interactions visible in the PMF attenuate or change sign in the survival function and vice versa. This empirically extends Prop. 3.5, derived under the multiplicative hazards assumption (Eq. (3), (7)): in a DeepHit model trained without any proportional hazards constraint, the choice of prediction scale (survival probability vs. PMF) still induces distinct interaction structures, suggesting that scale-dependence of feature interactions is a broader phenomenon than our theoretical results currently formalize.

**Scalability.** While computing exact interaction-based explanations is feasible for low-dimensional models ($p \leq 10$), higher dimensions often require efficient approximations. We benchmark common algorithms on four datasets with 10–16 features (Bergamaschi et al., 2006; Knaus et al., 1995; Shedden et al., 2008; Simons et al., 1999). Figures 7 and B.12 show that the regression-based (kernel) approximator is the most faithful across survival tasks, which is consistent with prior work (Fumagalli et al., 2024; Muschalik et al., 2024). However, it requires the sampling budget to exceed the number of interaction effects, with instability evident at budget $2^6$ in Fig. 7. Once the budget grows beyond this regime, the regression-based approximator becomes the most accurate. Sampling-based methods (Monte Carlo, SVARM, Permutation) instead show smaller standard errors at low budgets, trading higher bias for lower variance; a pattern consistent with prior SHAP-IQ benchmarks (Fumagalli et al., 2024; Muschalik et al., 2024). To address scalability concerns, we further experiment with a machine learning model that predicts survival for patients with breast cancer using 70 genomic features in Appendix B.3.

## 5. Discussion

**Conclusion.** As machine learning models gain traction in survival analysis, ensuring an understanding of feature

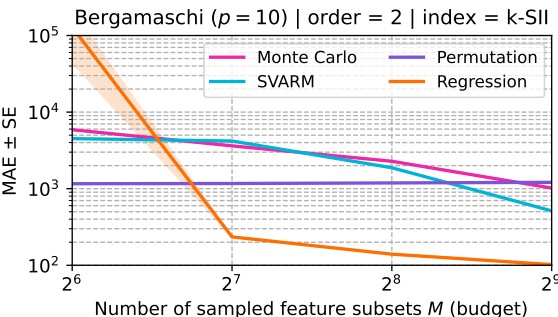

*Figure 7.* Error of different SurvSHAP-IQ approximators as a function of budget for the `Bergamaschi` task. Extended results for all tasks are in Figure B.12.

interactions is essential. We introduced the combination of SurvFD and SurvSHAP-IQ as a theoretically grounded approach to explain interactions in survival models, emphasizing the challenges of interpreting them across log-hazard, hazard, and survival scales. While the hazard and survival scales are clinically relevant, the additive log-hazard scale uniquely reflects the ground-truth interaction structure without transformation-induced artifacts. In practice, we recommend including second-order terms when extending analyses beyond additive effects, particularly in models that capture nonlinearities, since such interactions are intrinsic to the prediction function.

**Limitations and future work.** While our theoretical results (Thms. 3.2–3.6) assume the multiplicative hazard structure of Eq. (3), the SurvFD definition and SurvSHAP-IQ estimator extend to any square-integrable survival prediction function, including cause-specific log-hazards and log-intensities (preserving our guarantees) and non-additive targets such as cumulative incidence functions (transformation-induced interactions analogous to Prop. 3.5 and Cor. 3.4 are expected). Extending the theory to these settings is a natural direction for future work. Our empirical evaluation is restricted to second-order effects and medium-dimensional datasets (up to 70 features); high-dimensional applications such as genomics remain challenging due to the exponential growth in interaction terms with feature count and interaction order. Recent advances in scaling feature interaction methods to language and vision–language models suggest promising directions (Baniecki et al., 2025a; Butler et al., 2025), while pathway-level aggregation or dimensionality reduction prior to explanation offers a practical route in the meantime. Finally, the visualization and interpretability of interaction explanations remain open challenges for human-computer interaction research (Rong et al., 2024). Extending regional explanation frameworks such as Herbinger et al. (2024) and Herbinger et al. (2026) to survival and functional outputs may help identify subregions with weaker time-dependent interactions, enabling more interpretable additive explanations of hazard and survival functions within these regions.

## Impact Statement

This paper develops an interpretability methodology for survival models, with applications to clinical and biomedical prediction. Several considerations follow from our specific design choices.

SurvFD and SurvSHAP-IQ describe a model's learned behavior, not underlying causal structure. The attributions and interactions in our clinical examples reflect what the model relies on, not necessarily what drives patient outcomes, and should not substitute for further clinical validation. The choice between marginal and conditional reference distributions further encodes a normative decision about whether explanations should be true to the model or to the data. This choice has real consequences when sensitive or correlated features are involved, thus users should make it deliberately. Finally, our multi-modal application shows that SurvSHAP-IQ can attribute predictions to specific image regions, which supports auditing for spurious shortcuts but could equally be misused to retrofit unjustified narratives onto model outputs.

### ACKNOWLEDGEMENTS

The results shown here are in part based upon data generated by the TCGA Research Network: https://www.cancer.gov/tcga. Sophie Hanna Langbein and Niklas Koenen have been funded by the German Research Foundation (DFG) as part of the Research Unit "Lifespan AI: From Longitudinal Data to Lifespan Inference in Health" (DFG FOR 5347), Grant 459360854. Hubert Baniecki was supported by the Foundation for Polish Science (FNP), and the Polish Ministry of Education and Science within the "Pearls of Science" program, project number PN/01/0087/2022. Fabian Fumagalli gratefully acknowledges funding by the Deutsche Forschungsgemeinschaft (DFG, German Research Foundation): TRR 318/3 2026 – 438445824. Julia Herbinger and Marvin N. Wright gratefully acknowledge funding by the German Research Foundation (DFG), Emmy Noether Grant 437611051.

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

# Supplementary Materials

## A. Proofs

### A.1. Proof of Theorem 3.2

In Theorem 3.2, for the log-hazard $\log h(t|\boldsymbol{x}) = \log h_0(t) + G(t|\boldsymbol{x})$ and mutually independent features, we derive from each assumption (i) $G(t|\boldsymbol{x})$ is linear in $\boldsymbol{x}$ and (ii) $G(t|\boldsymbol{x})$ is additive in the features that $\mathcal{I}_d = \mathcal{I}_d^\star$ and $\mathcal{I}_{id} = \mathcal{I}_{id}^\star$. Additionally, without loss of generality, we can assume zero-centered features, i.e., $\mathbb{E}[X_i] = 0$. For both proofs, we consider the log-hazard function defined with ground-truth sets $\mathcal{I}_d$ and $\mathcal{I}_{id}$ given by (see Eq. (8) in the main text)

$$\log h(t|\boldsymbol{x}) = \log h_0(t) + G(t|\boldsymbol{x}) = \log h_0(t) + \sum_{M \in \mathcal{I}_d} g_M(t|\boldsymbol{x}) + \sum_{L \in \mathcal{I}_{id}} g_L(\boldsymbol{x}), \tag{A11}$$

where we use the separate feature sets $M$ and $L$ for notational convenience.

The pure FD effects (as defined in Eq. (5)) for some non-empty subset of features $T \in \mathcal{P}(P)$ for timepoint $t \in \mathcal{T}$ is recursively defined as:

$$f_T(t|\boldsymbol{x}) = \int F(t|\boldsymbol{x}) \, d\mathbb{P}_{\boldsymbol{X}_{\bar{T}}} - \sum_{S \subset T} f_S(t|\boldsymbol{x}) = \mathbb{E}_{X_{\bar{T}}}\left[F(t|\boldsymbol{X})|\boldsymbol{X}_T = \boldsymbol{x}_T\right] - \sum_{S \subset T} f_S(t|\boldsymbol{x}). \tag{A12}$$

The recursion above has the closed form (*Möbius Transformation*):

$$f_T(t|\boldsymbol{x}) = \sum_{S \subseteq T} (-1)^{|T|-|S|} \, \mathbb{E}_{\boldsymbol{X}_{\bar{S}}}\left[F(t|\boldsymbol{X}) \,\big|\, \boldsymbol{X}_S = \boldsymbol{x}_S\right]. \tag{A13}$$

In both cases (i) and (ii), we compute the pure effects to show that they are equivalent to the ground-truth components $g_T$ and therefore $\mathcal{I}_d^\star = \mathcal{I}_d$ and $\mathcal{I}_{id}^\star = \mathcal{I}_{id}$.

*Proof using assumption (i).* Due to the linearity assumption (i), we can write the function $G$ in a simple form using time-dependent functions $\beta_M(t)$ for time-dependent effects in $\mathcal{I}_d$ and time-independent coefficients $\beta_L$ for effects in $\mathcal{I}_{id}$, i.e.,

$$G(t|\boldsymbol{x}) = \sum_{M \in \mathcal{I}_d} \underbrace{\beta_M(t) \prod_{i \in M} x_i}_{=g_M(t|\boldsymbol{x})} + \sum_{L \in \mathcal{I}_{id}} \underbrace{\beta_L \prod_{j \in L} x_j}_{=g_L(\boldsymbol{x})}. \tag{A14}$$

**1. The $\emptyset$-component.** Using expression (A14), we can derive the baseline as

$$f_\emptyset(t) = \mathbb{E}_{\boldsymbol{X}}\left[\log h(t|\boldsymbol{X})\right] = \log h_0(t) + \mathbb{E}_{\boldsymbol{X}}[G(t|\boldsymbol{X})]$$

$$= \log h_0(t) + \sum_{M \in \mathcal{I}_d} \beta_M(t) \, \mathbb{E}_{\boldsymbol{X}}\left[\prod_{i \in M} X_i\right] + \sum_{L \in \mathcal{I}_{id}} \beta_L \, \mathbb{E}_{\boldsymbol{X}}\left[\prod_{j \in L} X_j\right]$$

$$= \log h_0(t).$$

The last step uses the feature independence and $\mathbb{E}[X_i] = 0$, implying that every non-empty product has mean zero, so only the log baseline hazard remains.

**2. Conditional expectations for the Möbius formula.** For any non-empty subset of features $T \in \mathcal{P}(P)$, we can calculate the joint marginal effect of the FD (which is required for the individual summands of the Möbius Transformation in

Eq. (A13)) for timepoint $t \in \mathcal{T}$ as

$$\mathbb{E}_{\boldsymbol{X}_{\bar{T}}}\big[\log h(t|\boldsymbol{X})\,\big|\,\boldsymbol{X}_T = \boldsymbol{x}_T\big]$$

$$= \log h_0(t) + \mathbb{E}_{\boldsymbol{X}_{\bar{T}}}\big[G(t|\boldsymbol{X})\,\big|\,\boldsymbol{X}_T = \boldsymbol{x}_T\big]$$

$$= \log h_0(t) + \sum_{M \in \mathcal{I}_d} \beta_M(t)\, \mathbb{E}_{\boldsymbol{X}_{\bar{T}}}\Big[\prod_{i \in M} X_i \,\Big|\, \boldsymbol{X}_T = \boldsymbol{x}_T\Big] + \sum_{L \in \mathcal{I}_{id}} \beta_L\, \mathbb{E}_{\boldsymbol{X}_{\bar{T}}}\Big[\prod_{j \in L} X_j \,\Big|\, \boldsymbol{X}_T = \boldsymbol{x}_T\Big]. \tag{A15}$$

By the assumed independence and zero means, the conditional product reduces to the fixed product, if all its indices lie in $T$, and to zero otherwise:

$$\mathbb{E}_{\boldsymbol{X}_{\bar{T}}}\Big[\prod_{i \in M} X_i \,\Big|\, \boldsymbol{X}_T = \boldsymbol{x}_T\Big] = \begin{cases} \prod_{i \in M} x_i, & M \subseteq T, \\ 0, & M \nsubseteq T, \end{cases}$$

$$\mathbb{E}_{\boldsymbol{X}_{\bar{T}}}\Big[\prod_{j \in L} X_j \,\Big|\, \boldsymbol{X}_T = \boldsymbol{x}_T\Big] = \begin{cases} \prod_{j \in L} x_j, & L \subseteq T, \\ 0, & L \nsubseteq T. \end{cases}$$

Hence together with Eq. (A15), it follows for the joint marginal effect

$$\mathbb{E}_{\boldsymbol{X}_{\bar{T}}}\big[\log h(t|\boldsymbol{X})\,\big|\,\boldsymbol{X}_T = \boldsymbol{x}_T\big] = \log h_0(t) + \sum_{\substack{M \in \mathcal{I}_d \\ M \subseteq T}} \beta_M(t) \prod_{i \in M} x_i + \sum_{\substack{L \in \mathcal{I}_{id} \\ L \subseteq T}} \beta_L \prod_{j \in L} x_j. \tag{A16}$$

**3. Calculate pure FD effects.** For each non-empty subset of features $S \subseteq T$, the pure effect is given by the Möbius Transformation (see Eq. (A13))

$$f_S(t|\boldsymbol{x}) = \sum_{T \subseteq S} (-1)^{|S|-|T|}\, \mathbb{E}_{\boldsymbol{X}_{\bar{T}}}\big[\log h(t|\boldsymbol{X})\,\big|\,\boldsymbol{X}_T = \boldsymbol{x}_T\big]. \tag{A17}$$

We now substitute the conditional representation from Eq. (A16):

$$f_S(t|\boldsymbol{x}) = \sum_{T \subseteq S} (-1)^{|S|-|T|} \log h_0(t) \quad + \sum_{T \subseteq S} (-1)^{|S|-|T|} \left( \sum_{\substack{M \in \mathcal{I}_d \\ M \subseteq T}} \beta_M(t) \prod_{i \in M} x_i \; + \sum_{\substack{L \in \mathcal{I}_{id} \\ L \subseteq T}} \beta_L \prod_{j \in L} x_j \right). \tag{A18}$$

The baseline term vanishes, which follows from using the binomial expansion and $S \neq \emptyset$ (i.e., $|S| > 0$):

$$\sum_{T \subseteq S} (-1)^{|S|-|T|} \log h_0(t) = \log h_0(t) \sum_{T \subseteq S} (-1)^{|S|-|T|}$$

$$= \log h_0(t) \sum_{k=0}^{|S|} \binom{|S|}{k} (-1)^{|S|-k}$$

$$= \log h_0(t)(-1)^{|S|}(-1+1)^{|S|} = 0.$$

Additionally, for the time-dependent term in Eq. (A18), it holds

$$\sum_{T \subseteq S} (-1)^{|S|-|T|} \sum_{\substack{M \in \mathcal{I}_d \\ M \subseteq T}} \beta_M(t) \prod_{i \in M} x_i = \sum_{T \subseteq S} (-1)^{|S|-|T|} \sum_{\substack{M \in \mathcal{I}_d \\ M \subseteq S}} \mathbb{1}_T(M) \beta_M(t) \prod_{i \in M} x_i$$

$$= \sum_{\substack{M \in \mathcal{I}_d \\ M \subseteq S}} \beta_M(t) \prod_{i \in M} x_i \left( \mathbb{1}_T(M) \sum_{T \subseteq S} (-1)^{|S|-|T|} \right)$$

$$= \sum_{\substack{M \in \mathcal{I}_d \\ M \subseteq S}} \beta_M(t) \prod_{i \in M} x_i \sum_{T: M \subseteq T \subseteq S} (-1)^{|S|-|T|},$$

where $\mathbb{1}_T(A)$ is 0 if $A \not\subseteq T$ else 1. Again, it follows from the binomial expansion

$$\sum_{T:\, M \subseteq T \subseteq S} (-1)^{|S|-|T|} = \sum_{k=0}^{|S|-|M|} \binom{|S|-|M|}{k} (-1)^{|S|-|M|-k} = (-1)^{|S|-|M|}(1-1)^{|S|-|M|} = \begin{cases} 1, & M = S, \\ 0, & M \subsetneq S. \end{cases} \quad \text{(A19)}$$

The same holds for the time-independent term in Eq. (A18). Since $\mathcal{I}_d$ and $\mathcal{I}_{id}$ are disjoint and $\mathcal{I}_d \cup \mathcal{I}_{id} = \mathcal{P}(P)$ (i.e., $S$ has to be in one of them), the pure effect is given by

$$f_S(t|\boldsymbol{x}) = \begin{cases} \beta_S(t) \prod_{i \in S} x_i, & S \in \mathcal{I}_d, \\ \beta_S \prod_{i \in S} x_i, & S \in \mathcal{I}_{id}, \end{cases} = \begin{cases} g_S(t|\boldsymbol{x}), & S \in \mathcal{I}_d, \\ g_S(\boldsymbol{x}), & S \in \mathcal{I}_{id}. \end{cases} \quad \text{(A20)}$$

**4. Identification of the sets.** The last expression (Eq. (A20)) of the pure effect $f_S$ for a subset $S \neq \emptyset$ of features depends on $t$ if and only if the corresponding ground-truth component is time-dependent. Therefore

$$\mathcal{I}_d^\star = \mathcal{I}_d \quad \text{and} \quad \mathcal{I}_{id}^\star = \mathcal{I}_{id}.$$

Since $\log h_0(t)$ contributes only to $f_\emptyset$, it does not affect these sets. This completes the proof under assumption (i). $\qquad \square$

*Proof using assumption (ii).* Due to the assumption of additivity in the main effects in (ii), we can write the function $G$ as sums over the individual time-dependent features in $M \in \mathcal{I}_d$ and time-independent features in $L \in \mathcal{I}_{id}$.

$$G(t|\boldsymbol{x}) = \sum_{M \in \mathcal{I}_d} g_M(t|\boldsymbol{x}) + \sum_{L \in \mathcal{I}_{id}} g_L(\boldsymbol{x}) = \sum_{\{i\} \in \mathcal{I}_d} g_{\{i\}}(t|\boldsymbol{x}) + \sum_{\{j\} \in \mathcal{I}_{id}} g_{\{j\}}(\boldsymbol{x}). \quad \text{(A21)}$$

Analogous to the proof using assumption (i), we want to show that under the assumption of mutually independent features, the time-dependent and time-independent functional components of the log-hazard function correspond exactly to the respective pure effects of the functional decomposition:

**1. The $\emptyset$-component.** Using expression in Eq. (A21), we can derive the baseline as

$$
\begin{aligned}
f_\emptyset(t) &= \mathbb{E}_{\boldsymbol{X}}\big[\log h(t|\boldsymbol{X})\big] \\[1mm]
&= \mathbb{E}_{\boldsymbol{X}}\Big[\log h_0(t) + \sum_{M \in \mathcal{I}_d} g_M(t|\boldsymbol{X}) + \sum_{L \in \mathcal{I}_{id}} g_L(\boldsymbol{X})\Big] \\[1mm]
&= \log h_0(t) + \sum_{\{i\} \in \mathcal{I}_d} \mathbb{E}_{X_i}\big[g_{\{i\}}(t|X_i)\big] + \sum_{\{j\} \in \mathcal{I}_{id}} \mathbb{E}_{X_j}\big[g_{\{j\}}(X_j)\big].
\end{aligned}
$$

In this case, the expectation terms do not cancel out as we do not impose the centering property of the functional decomposition. This means the component $g_\emptyset(t) = \log h_0(t)$ at this stage does not necessarily align with the pure FD component $f_\emptyset(t)$. Without explicitly requiring each first-order ground-truth component function to have zero mean, their expectations generally remain nonzero, so these attributions persist in this expression $f_\emptyset(t)$.

**2. Compute single-feature pure effect.** Let $M \subset P$ with $|M| = 1$ denote a single time-dependent feature index. By the recursive definition of FD components in Eq. (5), it holds

$$f_M(t|\boldsymbol{x}) = \mathbb{E}_{\boldsymbol{X}_{\bar{M}}}\big[\log h(t|\boldsymbol{X})|X_M = x_M\big] - f_\emptyset(t). \quad \text{(A22)}$$

We first compute the conditional expectation term. Substituting Eq. (A21) into Eq. (A22), we get (we omit the conditioning in the expectations for better clarity)

$$
\begin{aligned}
\mathbb{E}_{\boldsymbol{X}_{\bar{M}}}\big[\log h(t|\boldsymbol{X})\big] &= \mathbb{E}_{\boldsymbol{X}_{\bar{M}}}\Big[\log h_0(t) + \sum_{\{i\} \in \mathcal{I}_d} g_{\{i\}}(t|X_i) + \sum_{\{j\} \in \mathcal{I}_{id}} g_{\{j\}}(X_j)\Big] \\[1mm]
&= \log h_0(t) + \mathbb{E}_{\boldsymbol{X}_{\bar{M}}}\big[g_M(t|X_M)\big] + \sum_{\substack{\{i\} \in \mathcal{I}_d \\ i \notin M}} \mathbb{E}_{\boldsymbol{X}_{\bar{M}}}\big[g_{\{i\}}(t|X_i)\big] + \sum_{\{j\} \in \mathcal{I}_{id}} \mathbb{E}_{\boldsymbol{X}_{\bar{M}}}\big[g_j(X_j)\big]. \quad \text{(A23)}
\end{aligned}
$$

By independence of features, conditioning on $X_M = x_M$ does not affect the distribution of $X_i$ for $i \notin M$. Hence

$$\mathbb{E}_{\boldsymbol{X}_{\bar{M}}}\big[g_{\{i\}}(t|X_i)|X_M = x_M\big] = \mathbb{E}\big[g_{\{i\}}(t|X_i)\big] \quad \text{and} \quad \mathbb{E}_{\boldsymbol{X}_{\bar{M}}}\big[g_{\{j\}}(X_j)|X_M = x_M\big] = \mathbb{E}\big[g_{\{j\}}(X_j)\big],$$

and since $g_M(t|X_M)$ depends only on $X_M$, we have

$$\mathbb{E}_{\boldsymbol{X}_{\bar{M}}}\big[g_M(t|X_M)|X_M = x_M\big] = g_M(t|x_M).$$

Substituting these results in Eq. (A23) gives

$$\mathbb{E}_{\boldsymbol{X}_{\bar{M}}}\big[\log h(t|\boldsymbol{X})|X_M = x_M\big] = \log h_0(t) + g_M(t|x_M) + \sum_{\substack{\{i\}\in\mathcal{I}_d \\ i\notin M}} \mathbb{E}\big[g_{\{i\}}(t|X_i)\big] + \sum_{\{j\}\in\mathcal{I}_{id}} \mathbb{E}\big[g_{\{j\}}(X_j)\big].$$

Subtracting $f_\emptyset(t)$ yields

$$
\begin{aligned}
f_M(t|\boldsymbol{x}) &= \Big[\log h_0(t) + g_M(t|x_M) + \sum_{\substack{\{i\}\in\mathcal{I}_d \\ i\notin M}} \mathbb{E}\big[g_{\{i\}}(t|X_i)\big] + \sum_{\{j\}\in\mathcal{I}_{id}} \mathbb{E}\big[g_{\{j\}}(X_j)\big] \Big] \\
&\quad - \Big[\log h_0(t) + \sum_{\{i\}\in\mathcal{I}_d} \mathbb{E}_{X_i}\big[g_{\{i\}}(t|X_i)\big] + \sum_{\{j\}\in\mathcal{I}_{id}} \mathbb{E}_{X_j}\big[g_{\{j\}}(X_j)\big]\Big] \\
&= g_M(t|x_M) - \mathbb{E}_{X_M}[g_M(t|X_M)].
\end{aligned}
$$

All other terms cancel exactly. The same argument applies for a single time-independent feature $L$ with $|L| = 1$, giving $f_L(\boldsymbol{x}) = g_L(x_L) - \mathbb{E}_{X_L}[g_L(X_L)]$.

**3. Compute higher-order-feature pure effect.** We show now that all higher-order FD pure effects are 0. Therefore, we first define the joint pure effect of a time-dependent feature set $M$ with $|M| > 1$:

$$\mathbb{E}_{\boldsymbol{X}_{\bar{M}}}\big[\log h(t|\boldsymbol{X})|X_M = x_M\big] = \log h_0(t) + g_M(t|x_M) + \sum_{\substack{\{i\}\in\mathcal{I}_d \\ i\notin M}} \mathbb{E}\big[g_{\{i\}}(t|X_i)\big] + \sum_{\{j\}\in\mathcal{I}_{id}} \mathbb{E}\big[g_{\{j\}}(X_j)\big].$$

The pure effect of $M$ is defined by

$$f_M(t|\boldsymbol{x}) = \mathbb{E}_{\boldsymbol{X}_{\bar{M}}}[\log h(t|\mathbf{X})|\boldsymbol{X}_M = \mathbf{x}_M] - \sum_{\emptyset \neq S \subset M} f_S(t|\boldsymbol{x}) - f_\emptyset(t). \tag{A24}$$

We show now that the joint effect can be reformulated based on the emptyset and first-order FD components as follows:

$$
\begin{aligned}
f_\emptyset(t) + \sum_{\substack{\{i\}\in\mathcal{I}_d \\ i\in M}} f_{\{i\}}(t|x) &= \log h_0(t) + \sum_{\{i\}\in\mathcal{I}_d} \mathbb{E}_{X_i}\big[g_{\{i\}}(t|X_i)\big] + \sum_{\{j\}\in\mathcal{I}_{id}} \mathbb{E}_{X_j}\big[g_{\{j\}}(X_j)\big] + \\
&\qquad \sum_{\substack{\{i\}\in\mathcal{I}_d \\ i\in M}} \big(g_{\{i\}}(t|x_i) - \mathbb{E}_{X_i}\big[g_{\{i\}}(t|X_i)\big]\big) \\
&= \log h_0(t) + \sum_{\substack{\{i\}\in\mathcal{I}_d \\ i\in M}} g_{\{i\}}(t|x_i) + \sum_{\substack{\{i\}\in\mathcal{I}_d \\ i\notin M}} \mathbb{E}\big[g_{\{i\}}(t|X_i)\big] + \sum_{\{j\}\in\mathcal{I}_{id}} \mathbb{E}\big[g_{\{j\}}(X_j)\big] \\
&= \log h_0(t) + g_M(t|x_M) + \sum_{\substack{\{i\}\in\mathcal{I}_d \\ i\notin M}} \mathbb{E}\big[g_{\{i\}}(t|X_i)\big] + \sum_{\{j\}\in\mathcal{I}_{id}} \mathbb{E}\big[g_{\{j\}}(X_j)\big] \\
&= \mathbb{E}_{\boldsymbol{X}_{\bar{M}}}\big[\log h(t|\boldsymbol{X})|X_M = x_M\big]
\end{aligned}
$$

Based on the recursive formula of the Möbius Transfom, this leads to

$$f_M(t|\boldsymbol{x}) = \mathbb{E}_{\mathbf{X}_M}[\log h(t|\mathbf{X})|\mathbf{X}_M = \mathbf{x}_M] - \sum_{\emptyset \neq S \subset M} f_S(t|\boldsymbol{x}) - f_\emptyset(t) = 0 \tag{A25}$$

for all $|M| > 1$. The same also holds for a time-independent set $|L| > 1$, analogously.

**4. Identification of the sets.** We have shown that higher-order pure effects vanish $f_M(t|\boldsymbol{x}) = 0$, if $|M| > 1$ and first-order pure FD effects are equal to the additive components in $G(t|\boldsymbol{x})$, $f_M(t|\boldsymbol{x}) = g_M(t|\boldsymbol{x})$ for time-dependent as well as time-independent features. As a result the pure effect $f_S$ for a subset $S \neq \emptyset$ of features depends on $t$ if and only if the corresponding ground-truth component is time-dependent. Therefore

$$\mathcal{I}_d^\star = \mathcal{I}_d \quad \text{and} \quad \mathcal{I}_{id}^\star = \mathcal{I}_{id}.$$

This completes the proof under assumption (ii). $\qquad\square$

### A.2. Proof of Theorem 3.3

In Theorem 3.3, we assume $G(t|\boldsymbol{x}) = g_Z(t|\boldsymbol{x}) + \sum_{M \in \mathcal{I}_{id}} g_M(\boldsymbol{x})$ and $\mathcal{I}_d = \{Z\}$, with $Z$, $2 \leq |Z| < p$, being the only time-dependent set in $G$ and assume feature independence. Then:

(i) For any subset of the time-dependent set $L \subset Z$, $L$ may appear as time-dependent in the functional decomposition even when $L$ is time-independent in G (downward propagation).

(ii) Any superset of the time-dependent set $L \supset Z$ that is time-independent in $G$ remains time-independent in the functional decomposition (no upward propagation).

*Proof of (i) by Construction.* In Theorem 3.3, we assume $G(t|\boldsymbol{x}) = g_Z(t|\boldsymbol{x}) + \sum_{M \in \mathcal{I}_{id}} g_M(\boldsymbol{x})$, such that the log-hazard function can be written as

$$\log h(t|\boldsymbol{x}) = \log h_0(t) + g_Z(t|\boldsymbol{x}) + \sum_{M \in \mathcal{I}_{id}} g_M(\boldsymbol{x}), \tag{A26}$$

with $\{Z\} = \mathcal{I}_d$, $2 \leq |Z| < p$, the only time-dependent index set in $G$ and $M$ is the index set of time-independent features. We first show by construction that the pure effect $f_L$ of some subset $L \subset Z$ may appear as time-dependent in the functional decomposition of $\log h(t|\boldsymbol{x})$ even though $g_L$ is time-independent in $G$.

**Constructed example:** Consider the three-dimensional case $\boldsymbol{x} = (x_1, x_2, x_3)$ sampled from univariate standard Gaussians (i.e., $X_1, X_2, X_3 \overset{iid}{\sim} \mathcal{N}(0,1)$) and define

$$G(t|\boldsymbol{x}) = x_1^2 + x_2 + x_3 + x_1 x_2^2 t.$$

Here, the interaction term $g_{\{12\}}(t|\boldsymbol{x}) = x_1 x_2^2 t$ is time-dependent, while the first-order terms $g_{\{1\}}(\boldsymbol{x}) = x_1^2$, $g_{\{2\}}(\boldsymbol{x}) = x_2$ and $g_{\{3\}}(\boldsymbol{x}) = x_3$ are time-independent, such that $Z = \{1, 2\}$ and $\mathcal{I}_{id} = \{\{1\}, \{2\}, \{3\}\}$. We define set $L = \{1\} \in \mathcal{I}_{id}$, with $L \subset Z$, i.e., $g_L(\boldsymbol{x}) = g_{\{1\}}(\boldsymbol{x}) = x_1^2$ is time-independent.

**Calculate $f_L$:** The functional first-order effect of $x_1$ is defined as (see Eq. (5) and (A12))

$$\begin{aligned}
f_{\{1\}}(t|\boldsymbol{x}) &= \mathbb{E}_{(X_2, X_3)}[\log h(t|\mathbf{X})] - \mathbb{E}_{\boldsymbol{X}}[\log h(t|\mathbf{X})] \\
&= \mathbb{E}_{(X_2, X_3)}[\log h_0(t)] + \mathbb{E}_{(X_2, X_3)}[G(t|(x_1, X_2, X_3))] - \mathbb{E}_{\boldsymbol{X}}[\log h_0(t)] - \mathbb{E}_{\boldsymbol{X}}[G(t|\mathbf{X})] \\
&= \mathbb{E}_{(X_2, X_3)}[G(t|(x_1, X_2, X_3))] - \mathbb{E}_{\boldsymbol{X}}[G(t|\mathbf{X})], \tag{A27}
\end{aligned}$$

where the last step follows from the identity $\mathbb{E}_{(X_2, X_3)}[\log h_0(t)] = \log h_0(t) = \mathbb{E}_{\boldsymbol{X}}[\log h_0(t)]$ due to the feature independence of the baseline hazard. Now, we calculate the left summand of Eq. (A27):

$$\begin{aligned}
\mathbb{E}_{(X_2 X_3)}[G(t|(x_1, X_2, X_3))] &= \mathbb{E}_{(X_2 X_3)}[x_1^2 + X_2 + X_3 + x_1 X_2^2 t] \\
&= x_1^2 + \mathbb{E}_{X_2}[X_2] + \mathbb{E}_{X_3}[X_3] + t \cdot x_1 \mathbb{E}_{X_2}[X_2^2] \\
&= x_1^2 + t \cdot x_1, \tag{A28}
\end{aligned}$$

where we used the zero-centered features and $\mathbb{E}_{X_2}[X_2^2] = 1$. Now we compute the second component of Eq. (A27) as

$$\mathbb{E}_{\boldsymbol{X}}[G(t|\boldsymbol{X})] = \mathbb{E}_{X_1}[X_1^2] + \mathbb{E}_{X_2}[X_2] + \mathbb{E}_{X_3}[X_3] + t \cdot \mathbb{E}_{(X_1, X_2)}[X_1 X_2^2] = 1, \tag{A29}$$

where the last step uses the zero-centered features and $\mathbb{E}_{(X_1, X_2)}[X_1 X_2^2] = \mathbb{E}_{X_1}[X_1]\,\mathbb{E}_{X_2}[X_2^2] = 0$. Finally, we plug Eq. (A28) and Eq. (A29) in Eq. (A27).

$$f_{\{1\}}(t|\boldsymbol{x}) = x_1^2 + t \cdot x_1 - 1$$

Clearly, $f_{\{1\}}(t|\boldsymbol{x})$ depends on $t$, even though the first-order term in $g_{\{1\}}(\boldsymbol{x})$ in $G(t|\boldsymbol{x})$ is time-independent. This demonstrates that for the subset $L = \{1\} \subset Z = \{1, 2\}$, $L$ appears as time-dependent in the functional decomposition despite being time-independent in $G$.

$\square$

*Proof of (ii).* Let

$$G(t|\boldsymbol{x}) = g_Z(t|\boldsymbol{x}) + \sum_{M \in \mathcal{I}_{id}} g_M(\boldsymbol{x}),$$

where $Z$, with $2 \leq |Z| \leq p$ is the only time-dependent index set and every $g_M$ with $M \in \mathcal{I}_{id}$ is time-independent. Assume all features are mutually independent and let $f_M$ denote the functional pure effect of subset $M$ obtained from $G$ via the usual marginal projection (see Eq. (4)). Under feature independence, the functional decomposition is unique and yields for $\log h(t|\boldsymbol{x}) = \log h_0(t) + G(t|\boldsymbol{x})$:

$$\log h(t|\boldsymbol{x}) = f_\emptyset(t) + \sum_{\emptyset \neq S \subseteq P} f_S(t|\boldsymbol{x}),$$

where each $f_S$ is given by the recursive definition and explicit variant, the Möbius Transformation as in Eq. (A13), as

$$\begin{aligned}
f_S(t|\boldsymbol{x}) &= \mathbb{E}_{\boldsymbol{X}_{\bar{S}}}\big[\log h(t|\boldsymbol{x}) \,\big|\, \boldsymbol{X}_S = \boldsymbol{x}_S\big] - \sum_{N \subset S} f_N(t|\boldsymbol{x}) \\
&= \sum_{N \subseteq S} (-1)^{|S|-|N|} \mathbb{E}_{\boldsymbol{X}_{\bar{N}}}\big[\log h(t|\boldsymbol{x}) \,\big|\, \boldsymbol{X}_N = \boldsymbol{x}_N\big] \\
&= \sum_{N \subseteq S} (-1)^{|S|-|N|} \mathbb{E}_{\boldsymbol{X}_{\bar{N}}}\big[\log h_0(t) \,\big|\, \boldsymbol{X}_N = \boldsymbol{x}_N\big] + \sum_{N \subseteq S} (-1)^{|S|-|N|} \mathbb{E}_{\boldsymbol{X}_{\bar{N}}}\big[G(t|\boldsymbol{x}) \,\big|\, \boldsymbol{X}_N = \boldsymbol{x}_N\big] \\
&= \sum_{N \subseteq S} (-1)^{|S|-|N|} \mathbb{E}_{\boldsymbol{X}_{\bar{N}}}\big[G(t|\boldsymbol{x}) \,\big|\, \boldsymbol{X}_N = \boldsymbol{x}_N\big], \tag{A30}
\end{aligned}$$

where the last step follows similarly as in Step 3 in the proof of Theorem 3.2 due to the feature independence of the baseline hazard and the binomial expansion. We now aim to show that for $L$ with $|L| > |Z|$ the pure FD effect $f_L$ is time-independent. Using the definition of $G(t|\boldsymbol{x})$, we compute for $N \subseteq L$

$$\begin{aligned}
\mathbb{E}_{\boldsymbol{X}_{\bar{N}}}\big[G(t|\boldsymbol{x}) \,\big|\, \boldsymbol{X}_N = x_N\big] &= \mathbb{E}_{\boldsymbol{X}_{\bar{N}}}\Big[g_Z(t|\boldsymbol{x}) + \sum_{M \in \mathcal{I}_{id}} g_M(\boldsymbol{x}) \,\Big|\, \boldsymbol{X}_N = \boldsymbol{x}_N\Big] \\
&= \mathbb{E}_{\boldsymbol{X}_{\bar{N} \cap Z}}\big[g_Z(t|\boldsymbol{x}_Z) \,\big|\, \boldsymbol{X}_{N \cup \bar{Z}} = \boldsymbol{x}_{N \cup \bar{Z}}\big] + \sum_{M \in \mathcal{I}_{id}} \mathbb{E}_{\boldsymbol{X}_{\bar{N} \cap M}}\big[g_M(\boldsymbol{x}) \,\big|\, \boldsymbol{X}_{N \cup \bar{M}} = \boldsymbol{x}_{N \cup \bar{M}}\big],
\end{aligned}$$

since the expectation only needs to be taken over the variables used by the respective function $g$. Combining this with the non-recursive definition of the pure FD effect in Eq. (A30), we obtain the pure effect $f_L$ as (for convenience reasons, we omit the condition for the expectations)

$$f_L(t|\boldsymbol{x}) = \sum_{N \subseteq L} (-1)^{|L|-|N|} \mathbb{E}_{\boldsymbol{X}_{\bar{N} \cap Z}}\big[g_Z(t|\boldsymbol{x})\big] + C(\boldsymbol{x}),$$

where $C(\boldsymbol{x})$ is the remainder of aggregated time-independent effects. We now show that this time-dependent part of $f_L$ is zero by re-arranging the sum as

$$\sum_{N \subseteq L} (-1)^{|L|-|N|} \mathbb{E}_{\boldsymbol{X}_{\bar{N} \cap Z}}\big[g_Z(t|\boldsymbol{x})\big] = \sum_{S \subseteq Z} \mathbb{E}_{\boldsymbol{X}_{\bar{S}}}\big[g_Z(t|\boldsymbol{x})\big] \sum_{N \subseteq L : S = N \cap Z} (-1)^{|L|-|N|}.$$

For the inner sum, we can disjointly decompose $N = S \,\dot\cup\, R$ with $R \subseteq L \setminus Z$ and $S \cap R = \emptyset$, and thus

$$\sum_{N \subseteq L : S = N \cap Z} (-1)^{|L|-|N|} = \sum_{R \subseteq L \setminus Z} (-1)^{|L|-|S \cup R|} = \sum_{r=0}^{|L \setminus Z|} (-1)^{|L|-|S|-r} \binom{|L \setminus Z|}{r}$$

$$= (-1)^{|L|-|S|} \sum_{r=0}^{|L \setminus Z|} (-1)^r \binom{|L \setminus Z|}{r} = (-1)^{|L|-|S|}(-1+1)^{|L \setminus Z|} = 0,$$

since $|L \setminus Z| > 0$ due to $|L| > |Z|$. In conclusion, all time-dependent effects vanish in $f_L$, which finishes the proof. $\qquad\square$

## A.3. Proof of Corollary 3.4

In Corollary 3.4, we assume

$$G(t|\boldsymbol{x}) = g_Z(t|\boldsymbol{x}) + \sum_{M \in \mathcal{I}_{id}} g_M(\boldsymbol{x}),$$

and $\{Z\} = \mathcal{I}_d$, with $Z$, $2 \leq |Z| < p$, is the only time-dependent index set in $G$ and $\mathcal{I}_{id}$ is the index set of time-independent features. Then, for both the hazard $h(t|\boldsymbol{x}) = h_0(t)\exp(G(t|\boldsymbol{x}))$ and survival function $S(t|\boldsymbol{x}) = \exp\big(-\int_0^t h_0(u)\exp(G(u|\boldsymbol{x}))du\big)$, there exists

(i) a subset $L \subset Z$, which appears as time-dependent in the respective functional decomposition (i.e., $f_L(t|\boldsymbol{x})$ is time-dependent) even when $g_L(\boldsymbol{x})$ is time-independent in $G$ and

(ii) superset $L \supset Z$ such that the corresponding FD component $f_L(\cdot)$ becomes time-dependent even though $g_L(\cdot)$ is time-independent in $G$.

*Proof of (i) by Construction.* Consider the three-dimensional case $\boldsymbol{x} = (x_1, x_2, x_3)$ and define

$$G(t|\boldsymbol{x}) = x_3 + l_1(t)x_1 x_2,$$

where the interaction term $g_{\{1,2\}}(t|\boldsymbol{x}) = l_1(t)x_1 x_2$ is time-dependent on some non-constant function of time $l_1(t)$, while the first-order terms $g_{\{1\}}(\boldsymbol{x}) = 0$, $g_{\{2\}}(\boldsymbol{x}) = 0$ and $g_{\{3\}}(\boldsymbol{x}) = x_3$ are time-independent ($g_{\{1\}}(\boldsymbol{x}) = g_{\{2\}}(\boldsymbol{x}) = 0$ assumed for simplicity, but implies time-independence as stated in Sec. 3.1). Features are assumed to be mutually independent.

We define set $L = \{1\}$, with $L \subset Z = \{1,2\}$. The first-order FD effect for the hazard function $h$ of $x_1$ is defined by (see Eq. (5) and (A12))

$$f_L(t|\boldsymbol{x}) = \mathbb{E}_{(X_2, X_3)}[h(t|(x_1, X_2, X_3))] - \mathbb{E}_{\boldsymbol{X}}[h(t|\boldsymbol{X})].$$

Now, we calculate both components of the formula above, separately. For the first part, it holds

$$\mathbb{E}_{(X_2, X_3)}[h(t|(x_1, X_2, X_3))] = \mathbb{E}_{(X_2, X_3)}\big[h_0(t)\exp(X_3 + l_1(t)X_1 X_2)\big]$$

$$= \mathbb{E}_{(X_2, X_3)}\big[h_0(t)\exp(X_3)\exp(l_1(t)X_1 X_2)\big]$$

$$= h_0(t)\,\mathbb{E}_{X_3}\big[\exp(X_3)\big]\,\mathbb{E}_{X_2}\big[\exp(l_1(t)\,x_1 X_2)\big],$$

where the last step follows from the feature independence. For the second part, we derive

$$\mathbb{E}_{\boldsymbol{X}}\big[h(t|\boldsymbol{X})\big] = \mathbb{E}_{\boldsymbol{X}}\big[h_0(t)\exp(X_3 + l_1(t)X_1 X_2)\big]$$

$$= \mathbb{E}_{\boldsymbol{X}}\big[h_0(t)\exp(X_3)\exp(l_1(t)X_1 X_2)\big]$$

$$= h_0(t)\,\mathbb{E}_{X_3}\big[\exp(X_3)\big]\,\mathbb{E}_{(X_1, X_2)}\big[\exp(l_1(t)X_1 X_2)\big], \tag{A31}$$

where, again, the last step follows from the feature independence. Together, we get the first-order pure FD effect of $x_1$ as

$$f_L(t|\boldsymbol{x}) = h_0(t)\,\mathbb{E}_{X_3}\big[\exp(X_3)\big]\,\mathbb{E}_{X_2}\big[\exp(l_1(t)\,x_1 X_2)\big] - h_0(t)\,\mathbb{E}_{X_3}\big[\exp(X_3)\big]\,\mathbb{E}_{(X_1,X_2)}\big[\exp(l_1(t)\,X_1 X_2)\big]$$

$$= h_0(t)\,\mathbb{E}_{X_3}\big[\exp(X_3)\big]\,\Big\{\mathbb{E}_{X_2}\big[\exp(l_1(t)\,x_1 X_2)\big] - \mathbb{E}_{(X_1,X_2)}\big[\exp(l_1(t)\,X_1 X_2)\big]\Big\}. \tag{A32}$$

The braced term in Eq. (A32) depends on $l_1(t)$ and therefore on $t$, unless $l_1(t)$ is constant (or the distributions of $X_1$, $X_2$ are degenerate in a way that makes both expectations equal and independent of $l_1(t)$). Thus, in general $f_L(t|\boldsymbol{x})$ is time-dependent. This demonstrates that for the subset $L = \{1\} \subset Z = \{1,2\}$, $L$ appears as time-dependent in the FD decomposition despite $g_{\{1\}}(\boldsymbol{x})$ being time-independent in $G$.

The same argument applies to the survival function $S(t|\boldsymbol{x})$, since it is a further monotone non-linear transform of the hazard function $h(t|\boldsymbol{x})$. □

*Proof of (ii) by Construction.* Consider the three-dimensional case $\boldsymbol{x} = (x_1, x_2, x_3)$ and define

$$G(t|\boldsymbol{x}) = x_3 + l_1(t)\,x_1 x_2. \tag{A33}$$

Note that the interaction term $g_{\{1,2,3\}}(\boldsymbol{x}) = 0$ and thus time-independent as stated in the Definition in Sec. 3.1. Now we define a set $L = \{1,2,3\}$ and it holds $L \supset Z = \{1,2\}$. We again assume mutually independent features. The third-order pure FD effect $f_L$ is defined using the Möbius Transformation (see Eq. (A13)) as

$$f_L(t|\boldsymbol{x}) = \sum_{S \subseteq L}(-1)^{|L|-|S|}\,\mathbb{E}_{\boldsymbol{X}_{\bar{S}}}\Big[h(t|\boldsymbol{X})\big|\boldsymbol{X}_S = \boldsymbol{x}_S\Big]$$

$$= \sum_{S \subseteq \{1,2,3\}}(-1)^{3-|S|}\,\mathbb{E}_{\boldsymbol{X}_{\bar{S}}}\Big[h(t|\boldsymbol{X})\big|\boldsymbol{X}_S = \boldsymbol{x}_S\Big].$$

We define the following shorthand functions:

$$A(t) := \mathbb{E}_{(X_1,X_2)}\big[\exp(l_1(t)X_1 X_2)\big], \qquad\qquad B_1(t,x_1) := \mathbb{E}_{X_2}\big[\exp(l_1(t)\,x_1 X_2)\big],$$
$$B_2(t,x_2) := \mathbb{E}_{X_1}\big[\exp(l_1(t)\,X_1 x_2)\big], \qquad\qquad\qquad C := \mathbb{E}_{X_3}\big[\exp(X_3)\big].$$

Using these and the feature independence, we get

| | | | |
|---|---|---|---|
| $\mathbb{E}_{\boldsymbol{X}}[h(t|\boldsymbol{X})] = h_0(t)$ | $C$ | $A(t)$ | (sign: $-1$), |
| $\mathbb{E}_{(X_2,X_3)}[h(t|\boldsymbol{X})|X_1 = x_1] = h_0(t)$ | $C$ | $B_1(t,x_1)$ | (sign: $1$), |
| $\mathbb{E}_{(X_1,X_3)}[h(t|\boldsymbol{X})|X_2 = x_2] = h_0(t)$ | $C$ | $B_2(t,x_2)$ | (sign: $1$), |
| $\mathbb{E}_{(X_1,X_2)}[h(t|\boldsymbol{X})|X_3 = x_3] = h_0(t)$ | $\exp(x_3)$ | $A(t)$ | (sign: $1$), |
| $\mathbb{E}_{X_3}[h(t|\boldsymbol{X})|\boldsymbol{X}_{(1,2)} = \boldsymbol{x}_{(1,2)}] = h_0(t)$ | $C$ | $\exp(l_1(t)\,x_1 x_2)$ | (sign: $-1$), |
| $\mathbb{E}_{X_2}[h(t|\boldsymbol{X})|\boldsymbol{X}_{(1,3)} = \boldsymbol{x}_{(1,3)}] = h_0(t)$ | $\exp(x_3)$ | $B_1(t,x_1)$ | (sign: $-1$), |
| $\mathbb{E}_{X_1}[h(t|\boldsymbol{X})|\boldsymbol{X}_{(2,3)} = \boldsymbol{x}_{(2,3)}] = h_0(t)$ | $\exp(x_3)$ | $B_2(t,x_2)$ | (sign: $-1$), |
| $\mathbb{E}_{\varnothing}[h(t|\boldsymbol{X})|\boldsymbol{X} = \boldsymbol{x}] = h_0(t)$ | $\exp(x_3)$ | $\exp(l_1(t)\,x_1 x_2)$ | (sign: $1$). |

With the signs $(-1)^{3-|S|} \in \{-1,+1\}$, the terms factorize as

$$f_{\{1,2,3\}}(t|\boldsymbol{x}) = h_0(t)\,\big(\exp(x_3) - C\big)\,\big[A(t) - B_1(t,x_1) - B_2(t,x_2) + \exp(l_1(t)\,x_1 x_2)\big].$$

The factor $\exp(x_3) - C$ is time-independent, while the bracket carries the entire $t$-dependence via $l_1(t)$. For non-degenerate $(X_1, X_2)$ and non-constant $l_1(t)$, the bracket is not identically zero; hence $f_{\{1,2,3\}}(t|\boldsymbol{x})$ is (in general) time-dependent although $g_{\{1,2,3\}} \equiv 0$ in $G$.

The same argument applies to the survival function $S(t|\boldsymbol{x})$, since it is a monotone non-linear transformation of the hazard $h(t|\boldsymbol{x})$. □

### A.4. Proof of Proposition 3.5

*Proof.* Let $G(t|\boldsymbol{x}) = \sum_{j=1}^{p} \beta_j x_j$ be a linear CoxPH model with mutually independent features and $\beta_1, \ldots, \beta_p \neq 0$, and define $h(t|\boldsymbol{x}) = h_0(t) \exp(G(t|\boldsymbol{x}))$ (Eq. (1)) and $S(t|\boldsymbol{x}) = \exp\left(-\int_0^t h(u|\boldsymbol{x})\,du\right)$ (Eq. (2)).

For any pair of features $i \neq j$, the hazard satisfies

$$h(t|\boldsymbol{x}) = h_0(t) \exp(\beta_i x_i) \exp(\beta_j x_j) \prod_{k \neq i,j} \exp(\beta_k x_k),$$

so the functional component corresponding to $\{i,j\}$ is

$$f_{\{i,j\}}(t|x_i, x_j) = \mathbb{E}[h(t|\boldsymbol{X})|X_i = x_i, X_j = x_j] - \mathbb{E}[h(t|\boldsymbol{X})|X_i = x_i] - \mathbb{E}[h(t|\boldsymbol{X})|X_j = x_j] + \mathbb{E}[h(t|\boldsymbol{X})]$$
$$= h_0(t) \underbrace{\prod_{k \neq i,j} \mathbb{E}[\exp(\beta_k X_k)]}_{\neq 0} \Big( \exp(\beta_i x_i) - \mathbb{E}[\exp(\beta_i X_i)] \Big) \Big( \exp(\beta_j x_j) - \mathbb{E}[\exp(\beta_j X_j)] \Big).$$

Since $\beta_i, \beta_j \neq 0$ and the exponential function is strictly monotone, there exist values $x_i$ and $x_j$ for which $\exp(\beta_i x_i) \neq \mathbb{E}[\exp(\beta_i X_i)]$ and $\exp(\beta_j x_j) \neq \mathbb{E}[\exp(\beta_j X_j)]$, implying $f_{\{i,j\}}(t|x_i, x_j) \neq 0$.

By the same argument, higher-order products yield nonzero SurvFD components, and since $S(t|\boldsymbol{x})$ is a strictly monotone function of the integral of $h(t|\boldsymbol{x})$, its decomposition also contains nonzero interactions. $\qquad\square$

### A.5. Proof of Theorem 3.6

*Proof.* Let $G(t|\boldsymbol{x})$ be as in Eq. (7), and let $j$ be a feature with no direct effect on $G$, i.e., $\{j\} \in \mathcal{I}_{id}$ and $G$ does not depend on $X_j$. Thus, we have

$$G(t|\boldsymbol{x}) = G(t|\boldsymbol{x}_{\{\bar{j}\}}),$$

which implies the joint marginal feature distribution of $\boldsymbol{X}_{\{\bar{j}\}}$ to be $\mathbb{P}(\boldsymbol{X}_{\{\bar{j}\}}) = \mathbb{P}(\boldsymbol{X})$.

**Marginal FD:** By definition,

$$f_{\{j\}}^{\text{marginal}}(t|\boldsymbol{x}) = \mathbb{E}[G(t|\boldsymbol{X}_{\{\bar{j}\}})|X_j = x_j] - \mathbb{E}[G(t|\mathbf{X})]$$
$$= \int G(t|\mathbf{X}_{\{\bar{j}\}})\,d\mathbb{P}(\mathbf{X}_{\{\bar{j}\}}) - \int G(t|\mathbf{X})\,d\mathbb{P}(\mathbf{X})$$
$$= \int G(t|\mathbf{X})\,d\mathbb{P}(\mathbf{X}) - \int G(t|\mathbf{X})\,d\mathbb{P}(\mathbf{X})$$
$$= 0,$$

so both integrals are identical, yielding

$$f_{\{j\}}^{\text{marginal}}(t|\boldsymbol{x}) \equiv 0.$$

**Conditional FD:** By definition,

$$f_{\{j\}}^{\text{conditional}}(t|\boldsymbol{x}) = \mathbb{E}[G(t|\mathbf{X}_{\{\bar{j}\}})|X_j = x_j] - \mathbb{E}[G(t|\mathbf{X})]$$
$$= \int G(t|\mathbf{X})\,d\mathbb{P}(\mathbf{X}_{\{\bar{j}\}}|X_j = x_j) - \int G(t|\mathbf{X})\,d\mathbb{P}(\mathbf{X}).$$

If $X_j$ is dependent on a time-dependent feature $X_k$ (i.e., $\{k\} \in \mathcal{I}_d$), then

$$\mathbb{P}(\mathbf{X}_{\{\bar{j}\}}|X_j = x_j) \neq \mathbb{P}(\mathbf{X}_{\{\bar{j}\}}),$$

so conditioning on $X_j$ changes the distribution of $X_k$ in the integral. Since $G$ depends on $X_k$, this produces a nonzero contribution:

$$f_{\{j\}}^{\text{conditional}}(t|\boldsymbol{x}) = \int G(t|\mathbf{X}) \, d\mathbb{P}(\mathbf{X}_{\{\bar{j}\}}|X_j = x_j) - \int G(t|\mathbf{X}) \, d\mathbb{P}(\mathbf{X}) \not\equiv 0.$$

Hence, the conditional FD component of $X_j$ is nonzero and if $X_k$ is time-dependent, the effect can vary with $t$, while the marginal FD is always zero. $\square$

### A.6. Relationship between SurvFD and Shapley Interactions

Here, we show how the definition of Shapley interactions relates to our introduced SurvFD definition.

Shapley interactions, defined as in Eq. (10), are a weighted sum over all subsets $M$ of the discrete derivative defined by

$$\Delta_K(M) := \sum_{L \subseteq K} (-1)^{|K|-|L|} \nu(t|M \cup L).$$

The definition of the discrete derivative relates to another game theoretic concept, which is the *Möbius Transform* (Grabisch & Roubens, 1999). The Möbius Transform at some timepoint $t \in \mathcal{T}$ is defined by the discrete derivative for $M = \emptyset$:

$$\Delta_K(\emptyset) := \sum_{L \subseteq K} (-1)^{|K|-|L|} \, \nu(t|L).$$

It has been shown that both Shapley values and Shapley interaction indices can be represented by the Möbius Transform (Grabisch & Roubens, 1999).

The Möbius Transform also directly relates to the functional decomposition as shown by (Fumagalli et al., 2025). In our time-dependent setting, for $\nu(t|K) = \mathbb{E}_{\boldsymbol{X}_{\bar{K}}}[F(t|X)|\boldsymbol{X}_K = \boldsymbol{x}_K]$, i.e., where the value function is defined by the joint marginal effect of features in $K$ for some $t$, the Möbius Transform corresponds to the pure SurvFD effects:

$$f_K(t|\boldsymbol{x}) := \sum_{L \subseteq K} (-1)^{|K|-|L|} \, \mathbb{E}_{\boldsymbol{X}_{\bar{L}}}[F(t|X)|\boldsymbol{X}_L = \boldsymbol{x}_L].$$

It has been shown that Shapley interaction indices recover the pure effects of the Möbius Transform when they are computed up to the full feature order, i.e., when all subsets $K \subseteq \{1, \ldots, p\}$ are considered (Bordt & von Luxburg, 2023). In this case, the Shapley interaction indices coincide with the pure effects $f_K(t|\mathbf{x})$ from the functional decomposition when the value function $\nu(t|K)$ is defined as above. When the computation is restricted to interactions up to a predefined order $n$, higher-order Möbius terms ($|K| > n$) are redistributed among lower-order Shapley effects. Nevertheless, even in this truncated setting, the resulting Shapley interaction indices remain expressible as linear combinations of the underlying Möbius components.

### A.7. Time and Memory Complexity of SurvSHAP-IQ

The exact method has exponential time complexity $O(|T| \cdot 2^p)$, where $p$ is the number of features and $|T|$ is the number of timepoints of interest. The memory complexity is $O(2^p)$, because explanations can be computed sequentially and therefore do not require storing results for all timepoints simultaneously. The regression-based SurvSHAP-IQ approximation reduces the computation to polynomial time: $O(|T| \cdot (|B| \cdot p^{2k} + p^{3k}))$, where $k$ denotes the interaction order and $B$ is the number of basis samples. The corresponding memory complexity is $O(|B| \cdot p^k + p^{2k})$, which again does not depend on $|T|$ due to sequential computation across time.

# B. Empirical Validation

Code to reproduce all empirical validations and experiments is available on GitHub[2].

## B.1. Experiments with Simulated Data and Risk Scores under Independence Assumption

### B.1.1. SURVSHAP-IQ ATTRIBUTIONS

**Simulation Setting.** For each simulation scenario, the hazard function for an individual observation is given by

$$h(t|\boldsymbol{x}) = h_0(t)\exp(G(t|\boldsymbol{x})), \tag{B34}$$

where $G(t|\boldsymbol{x})$ specifies the log-risk function. The exact forms of $G(t|\boldsymbol{x})$ for the ten scenarios are listed in Tab. B.1. The features are independently drawn as $x_1, x_2, x_3 \sim \mathcal{N}(0,1)$, and the model parameters are fixed to $h_0(t) = 0.03$, $\beta_1 = 0.4$, $\beta_2 = -0.8$, $\beta_3 = -0.6$, $\beta_{12} = -0.5$, and $\beta_{13} = 0.2$. Event times $t^{(i)}$ are simulated using the approach of Bender et al. (2005) for proportional hazards models and Crowther & Lambert (2013) for non-proportional hazards models, implemented via the simsurv package (Brilleman et al., 2021). Right-censoring is applied by setting $t^{(i)} = 70$ for all observations with $t^{(i)} \geq 70$.

*Table B.1.* Overview of the ten simulation scenarios.

| # | $G(t|\boldsymbol{x})$ |
|---|---|
| (1) | $x_1\beta_1 + x_2\beta_2 + x_3\beta_3$ (linear $G(t|\boldsymbol{x})$, TI) |
| (2) | $x_1\beta_1\log(t+1) + x_2\beta_2 + x_3\beta_3$ (linear $G(t|\boldsymbol{x})$, TD main) |
| (3) | $x_1\beta_1 + x_2\beta_2 + x_3\beta_3 + x_1x_3\beta_{13}$ (linear $G(t|\boldsymbol{x})$, TI, interaction) |
| (4) | $x_1\beta_1\log(t+1) + x_2\beta_2 + x_3\beta_3 + x_1x_3\beta_{13}$ (linear $G(t|\boldsymbol{x})$, TD main, interaction) |
| (5) | $x_1\beta_1 + x_2\beta_2 + x_3\beta_3 + x_1x_3\beta_{13}\log(t+1)$ (linear $G(t|\boldsymbol{x})$, TD interaction) |
| (6) | $\beta_1 x_1^2 + \beta_2\frac{2}{\pi}\arctan(0.7x_2) + \beta_3 x_3$ (generalized additive $G(t|\boldsymbol{x})$, TI) |
| (7) | $\beta_1 x_1^2\log(t+1) + \beta_2\frac{2}{\pi}\arctan(0.7x_2) + \beta_3 x_3$ (generalized additive $G(t|\boldsymbol{x})$, TD main) |
| (8) | $\beta_1 x_1^2 + \beta_2\frac{2}{\pi}\arctan(0.7x_2) + \beta_3 x_3 + \beta_{12}x_1x_2 + \beta_{13}x_1x_3^2$ (generalized additive $G(t|\boldsymbol{x})$, TI, interaction) |
| (9) | $\beta_1 x_1^2\log(t+1) + \beta_2\frac{2}{\pi}\arctan(0.7x_2) + \beta_3 x_3 + \beta_{12}x_1x_2 + \beta_{13}x_1x_3^2$ (generalized additive $G(t|\boldsymbol{x})$, TD main, interaction) |
| (10) | $\beta_1 x_1^2 + \beta_2\frac{2}{\pi}\arctan(0.7x_2) + \beta_3 x_3 + \beta_{12}x_1x_2 + \beta_{13}x_1x_3^2\log(t+1)$ (generalized additive $G(t|\boldsymbol{x})$, TD interaction) |

TI = time-independent, TD = time-dependent

In total, $n = 1{,}000$ observations are simulated and randomly split into a training set ($n = 800$) and a test set ($n = 200$). Two models are then fitted to the training data: (1) a standard Cox proportional hazards (CoxPH) model including only main effects, and (2) a gradient-boosted survival analysis model (GBSA) using a CoxPH loss with regression trees as base learners. All hyperparameters are kept at their default values as implemented in the scikit-survival package (Pölsterl, 2020). Additional implementation details and code are provided in the online supplement.

**Model performance.** Table B.2 reports the Concordance Index (C-Index) and Integrated Brier Score (IBS) as overall performance metrics. GBSA consistently outperforms the CoxPH model across all scenarios, except for the two linear specifications without interactions, where the correctly specified CoxPH model performs comparably. These results highlight the ability of GBSA to capture complex, non-linear relationships beyond the scope of the CoxPH model.

---

[2]https://github.com/bips-hb/survshapiq

*Table B.2.* Comparison of model performance across ten simulation scenarios for CoxPH and GBSA using Integrated Brier Score (IBS) and Concordance Index (C-Index). Lower IBS and higher C-Index indicate better performance.

| Model | Scenario | IBS ↓ | C-Index ↑ |
|-------|----------|-------|-----------|
| CoxPH | (**1**) linear $G(t\|\boldsymbol{x})$ TI, no interactions | 0.143 | 0.759 |
| | (**2**) linear $G(t\|\boldsymbol{x})$ TD main, no interactions | 0.118 | 0.788 |
| | (**3**) linear $G(t\|\boldsymbol{x})$ TI, interactions | 0.173 | 0.725 |
| | (**4**) linear $G(t\|\boldsymbol{x})$ TD main, interactions | 0.172 | 0.707 |
| | (**5**) linear $G(t\|\boldsymbol{x})$ TD interactions | 0.202 | 0.673 |
| | (**6**) generalized additive $G(t\|\boldsymbol{x})$ TI, no interactions | 0.151 | 0.673 |
| | (**7**) generalized additive $G(t\|\boldsymbol{x})$ TD main, no interactions | 0.134 | 0.645 |
| | (**8**) generalized additive $G(t\|\boldsymbol{x})$ TI, interactions | 0.162 | 0.655 |
| | (**9**) generalized additive $G(t\|\boldsymbol{x})$ TD main, interactions | 0.137 | 0.655 |
| | (**10**) generalized additive $G(t\|\boldsymbol{x})$ TD interactions | 0.169 | 0.658 |
| GBSA | (**1**) linear $G(t\|\boldsymbol{x})$ TI, no interactions | 0.152 | 0.742 |
| | (**2**) linear $G(t\|\boldsymbol{x})$ TD main, no interactions | 0.126 | 0.776 |
| | (**3**) linear $G(t\|\boldsymbol{x})$ TI, interactions | 0.149 | 0.780 |
| | (**4**) linear $G(t\|\boldsymbol{x})$ TD main, interactions | 0.149 | 0.743 |
| | (**5**) linear $G(t\|\boldsymbol{x})$ TD interactions | 0.145 | 0.780 |
| | (**6**) generalized additive $G(t\|\boldsymbol{x})$ TI, no interactions | 0.151 | 0.696 |
| | (**7**) generalized additive $G(t\|\boldsymbol{x})$ TD main, no interactions | 0.115 | 0.703 |
| | (**8**) generalized additive $G(t\|\boldsymbol{x})$ TI, interactions | 0.149 | 0.697 |
| | (**9**) generalized additive $G(t\|\boldsymbol{x})$ TD main, interactions | 0.111 | 0.744 |
| | (**10**) generalized additive $G(t\|\boldsymbol{x})$ TD interactions | 0.151 | 0.702 |

TI = time-independent, TD = time-dependent.
IBS = Integrated Brier Score (lower is better). C-Index = Concordance Index (higher is better).

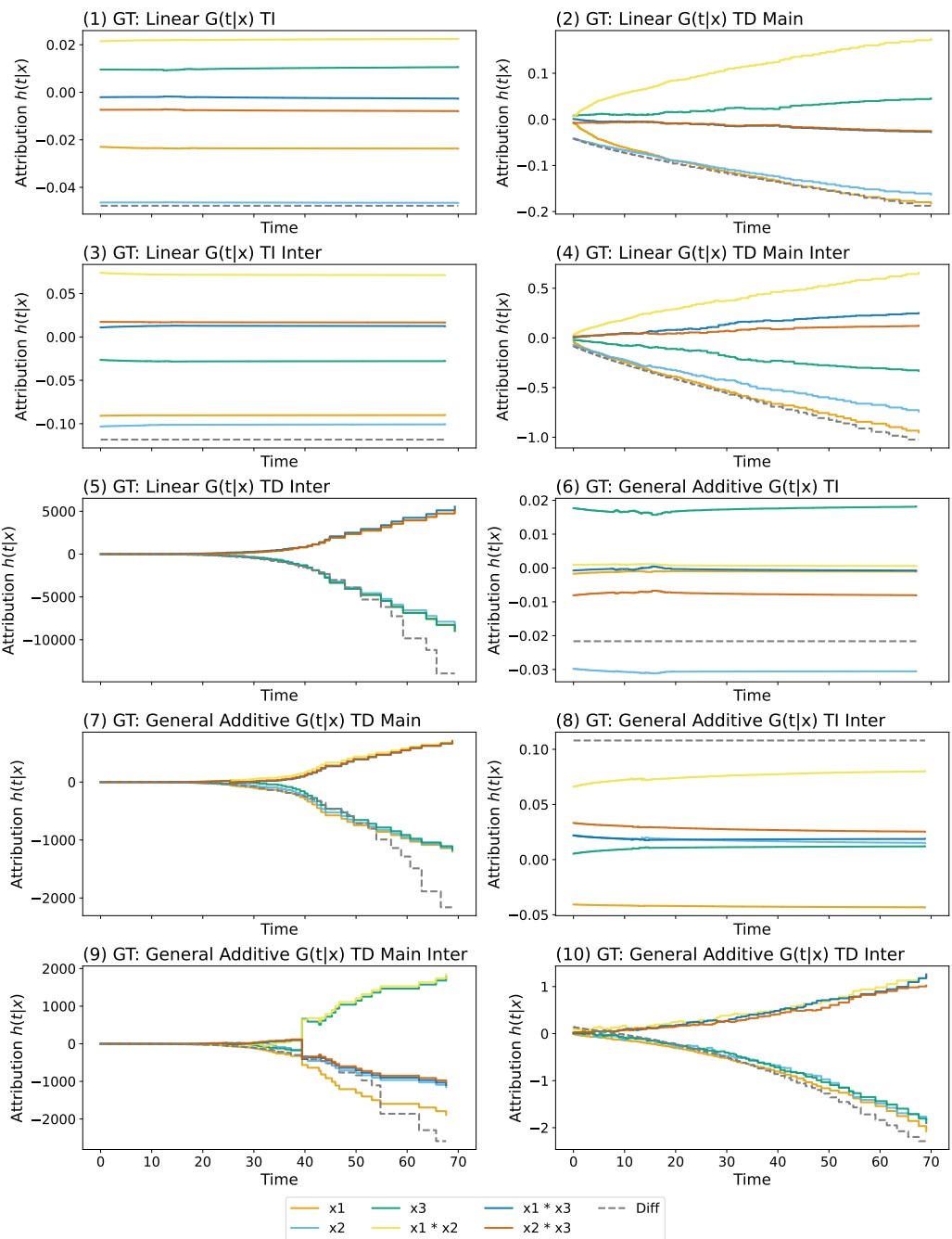

*Figure B.1.* Exact SurvSHAP-IQ decomposition attribution curves for a selected observation $[-1.2650, 2.4162, -0.6436]$ computed on the ground-truth hazard functions of all ten simulation scenarios. The plots show feature- and interaction-specific hazard attributions (colored curves, Y-axis) over time (X-axis). The reference curve (differences in individual hazard and mean hazard) is overlaid in a dashed grey line.

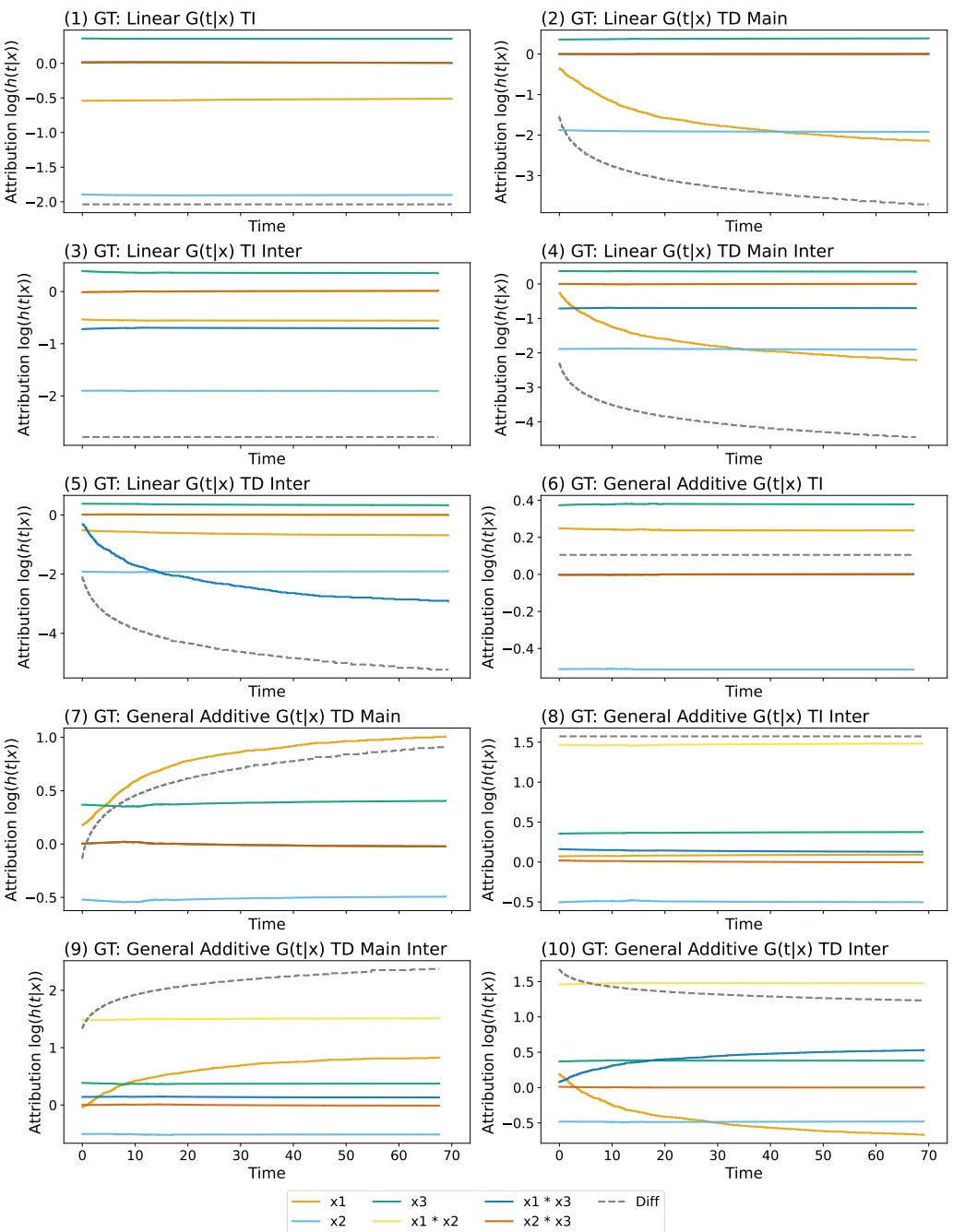

*Figure B.2.* Exact SurvSHAP-IQ decomposition curves for a selected observation $[-1.2650, 2.4162, -0.6436]$ computed on the ground-truth log-hazard functions of all ten simulation scenarios. The plots show feature- and interaction-specific log-hazard attributions (colored curves, Y-axis) over time (X-axis). The reference curve (differences in individual log-hazard and mean log-hazard) is overlaid in a dashed grey line.

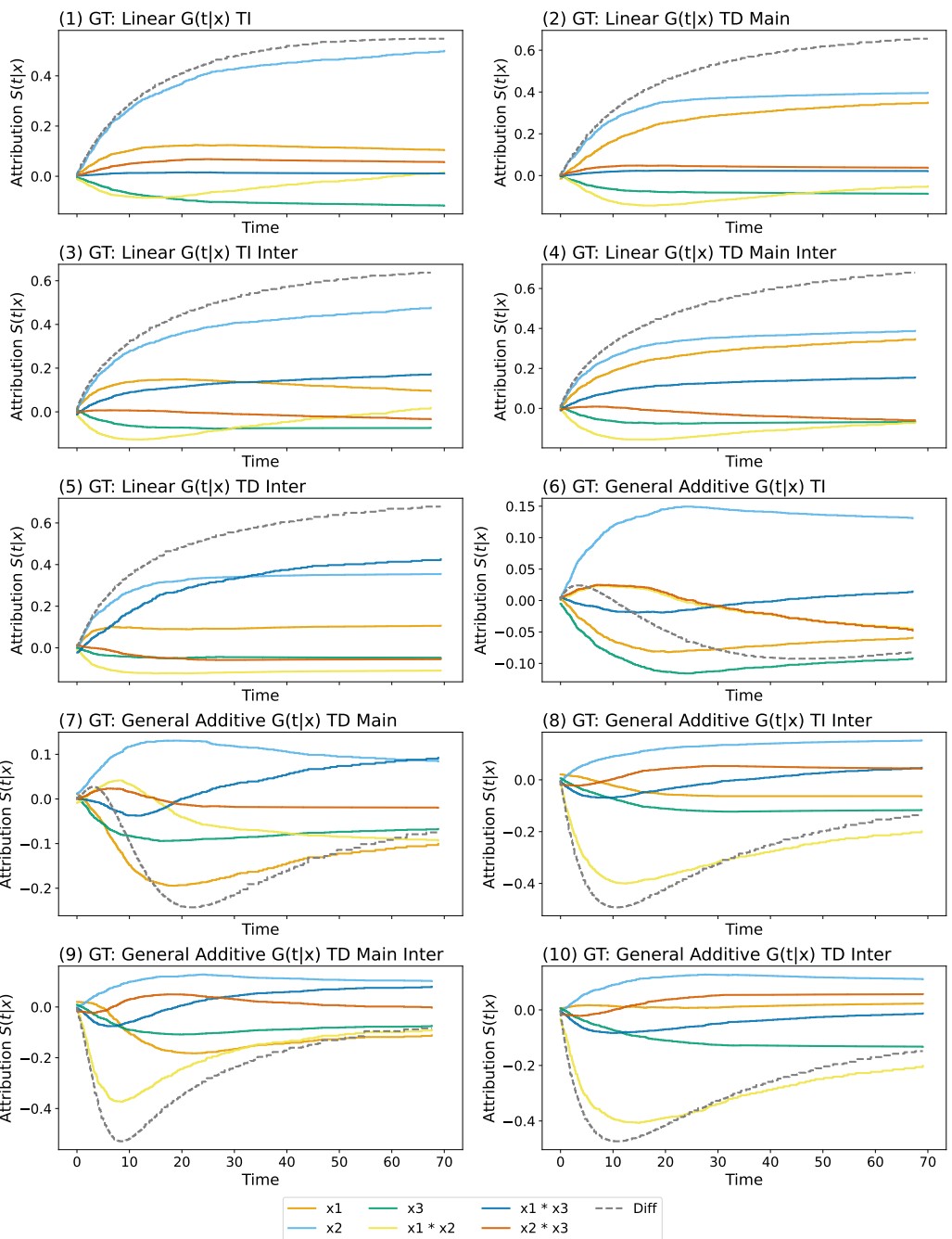

*Figure B.3.* Exact SurvSHAP-IQ decomposition curves for a selected observation $[-1.2650, 2.4162, -0.6436]$ computed on the approximated ground-truth survival functions of all ten simulation scenarios. The plots show feature- and interaction-specific survival attributions (colored curves, Y-axis) over time (X-axis). The reference curve (differences in individual survival and mean survival) is overlaid in a dashed grey line.

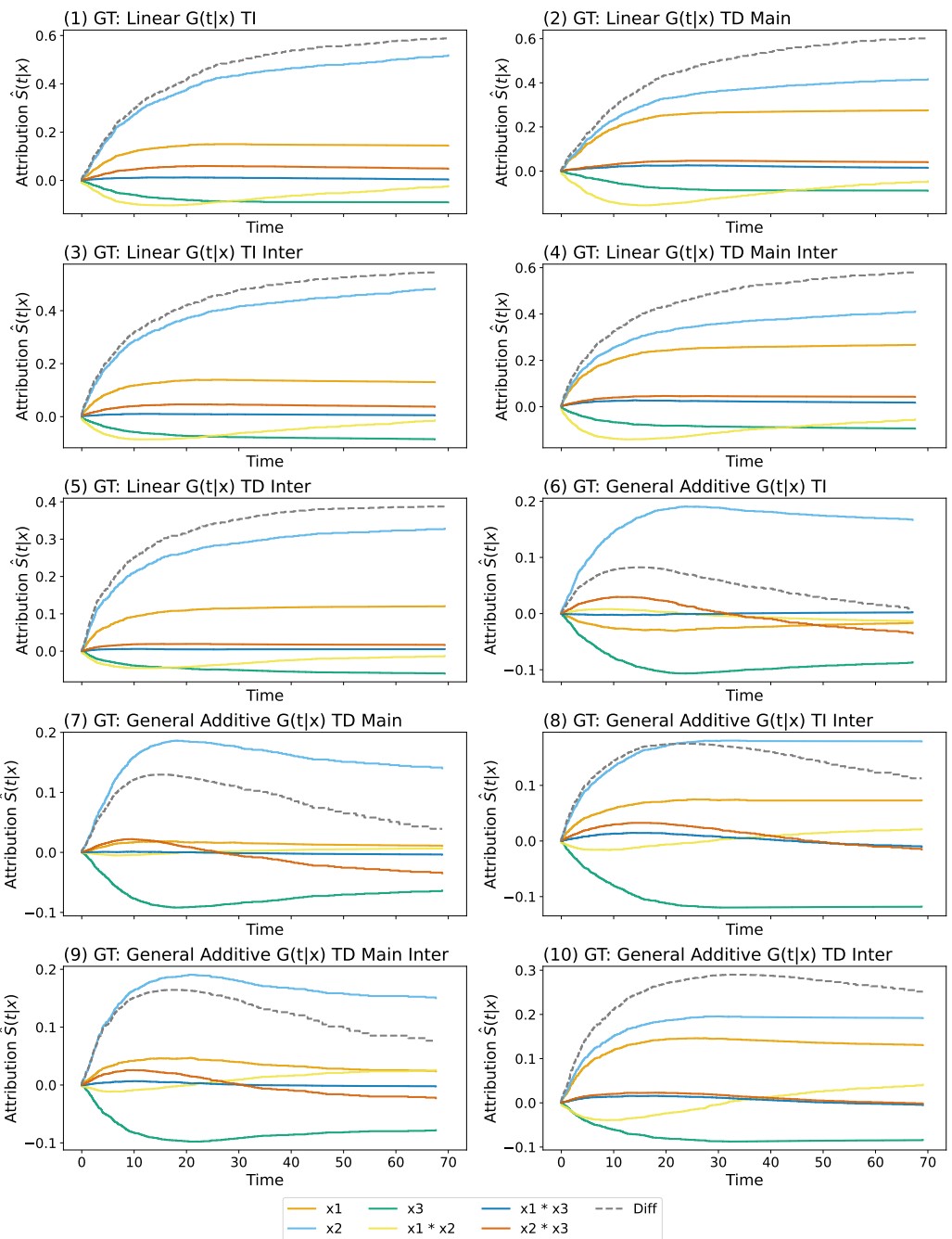

*Figure B.4.* Exact SurvSHAP-IQ decomposition curves for a selected observation $[-1.2650, 2.4162, -0.6436]$ computed on the predicted survival functions of fitted CoxPH for all ten simulation scenarios. The plots show feature- and interaction-specific survival attributions (colored curves, Y-axis) over time (X-axis). The reference curve (differences in individual survival and mean survival) is overlaid in a dashed grey line.

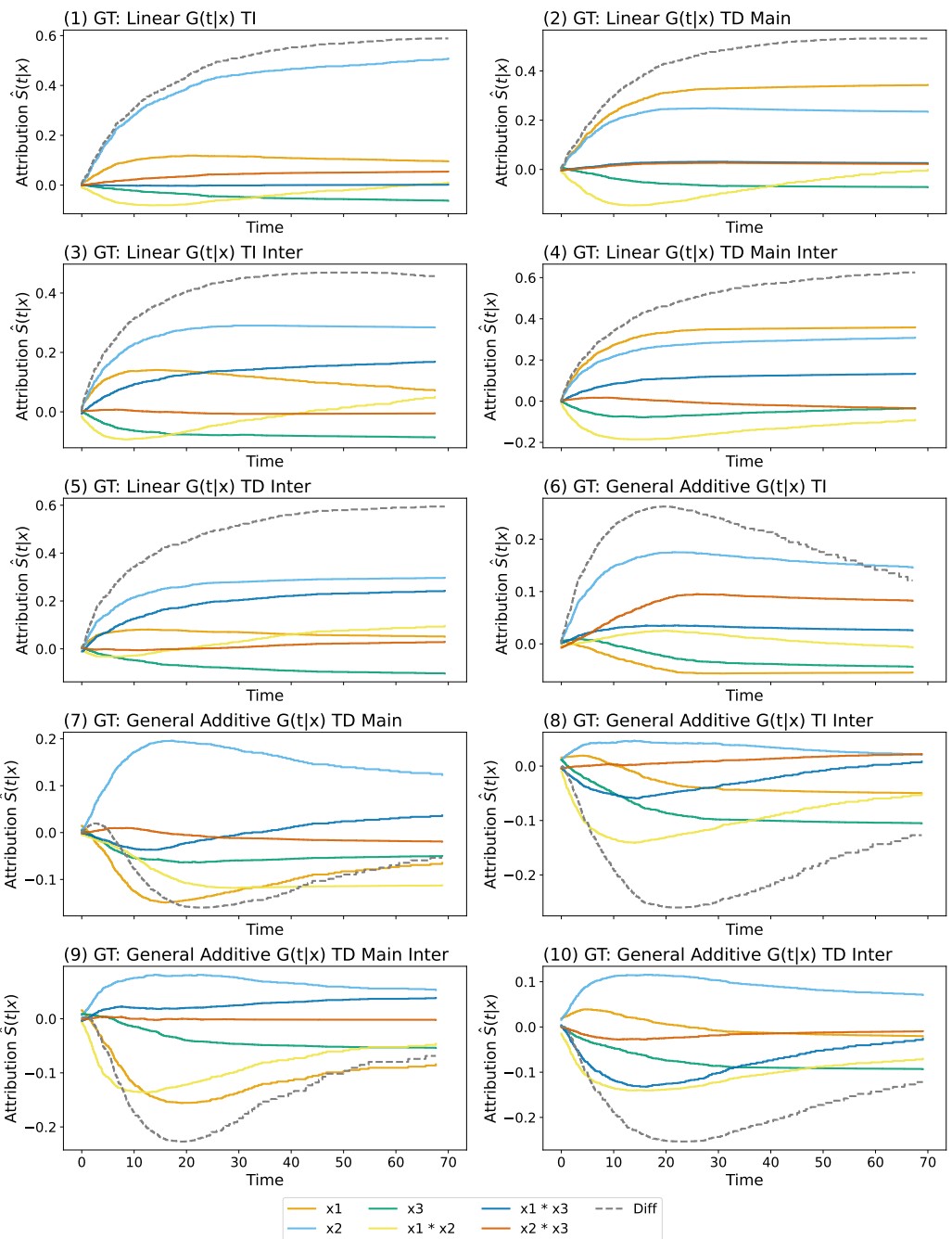

*Figure B.5.* Exact SurvSHAP-IQ decomposition curves for a selected observation $[-1.2650, 2.4162, -0.6436]$ computed on the predicted survival functions of fitted GBSA models for all ten simulation scenarios. The plots show feature- and interaction-specific survival attributions (colored curves, Y-axis) over time (X-axis). The reference curve (differences in individual survival and mean survival) is overlaid in a dashed grey line.

### B.1.2. LOCAL ACCURACY

The concept of *local accuracy* originates from Shapley values and states that the sum of all feature attributions equals the difference between an individual prediction and the expected prediction over the data (Lundberg & Lee, 2017):

$$\sum_{j=1}^{p} \phi_j = F(\boldsymbol{x}) - \mathbb{E}_{\boldsymbol{x}}[F(\boldsymbol{x})].$$

In survival analysis, this property must hold *at each timepoint*, since survival probabilities evolve over time and decrease monotonically. To account for this, Krzyziński et al. introduce a *time-dependent local accuracy measure*:

$$\Phi(t) = \sum_{M \subseteq P: |M| \leq k} \phi_M^{(k)}(t|\boldsymbol{x}) \tag{B35}$$

$$\sigma(t) = \sqrt{\frac{\mathbb{E}\left[(F(t|\boldsymbol{x}) - \mathbb{E}[F(t|\boldsymbol{X})] - \Phi(t))^2\right]}{\mathbb{E}\left[F(t|\boldsymbol{X})^2\right]}}. \tag{B36}$$

This formulation normalizes errors by the magnitude of the target function, giving proportionally more weight to discrepancies at later timepoints where survival probabilities are smaller. From Fig. B.6, we observe that local accuracy values are consistently very low for the ground-truth hazard and log-hazard, as these quantities can be exactly decomposed by SurvSHAP-IQ. They are slightly higher for the survival function, which is approximated from the hazard, and highest for predicted survival functions, where model bias, smoothing, and regularization contribute to approximation error.

To summarize performance over the full time horizon, the local accuracy can be averaged:

$$\bar{\sigma} = \frac{1}{|T|} \sum_{t \in T} \sigma(t),$$

yielding a single measure of how well the SurvSHAP-IQ decomposition reconstructs the model or ground-truth function across time (see Tab. B.3).

*Table B.3.* Average local accuracy over time comparing ground-truth hazard, log-hazard, and survival function with predicted survival from CoxPH and GBSA models, evaluated on ten simulated scenarios. Values are low, indicating high accuracy of the decomposition.

| Scenario | $h(t|\boldsymbol{x})$ | $\log(h(t|\boldsymbol{x}))$ | $S(t|\boldsymbol{x})$ | $\hat{S}_{CoxPH}(t|\boldsymbol{x})$ | $\hat{S}_{GBSA}(t|\boldsymbol{x})$ |
|---|---|---|---|---|---|
| (1) linear $G(t|\boldsymbol{x})$ TI, no inter. | <0.00001 | <0.00001 | <0.00001 | 0.00315 | 0.00552 |
| (2) linear $G(t|\boldsymbol{x})$ TD main, no inter. | <0.00001 | <0.00001 | 0.00200 | 0.00416 | 0.00425 |
| (3) linear $G(t|\boldsymbol{x})$ TI, inter. | <0.00001 | <0.00001 | <0.00001 | 0.00317 | 0.00180 |
| (4) linear $G(t|\boldsymbol{x})$ TD main, inter. | <0.00001 | <0.00001 | 0.00160 | 0.00408 | 0.00216 |
| (5) linear $G(t|\boldsymbol{x})$ TD inter. | <0.00001 | <0.00001 | 0.00215 | 0.00258 | 0.00349 |
| (6) general additive $G(t|\boldsymbol{x})$ TI, no inter. | <0.00001 | <0.00001 | <0.00001 | 0.00157 | 0.01061 |
| (7) general additive $G(t|\boldsymbol{x})$ TD main, no inter. | <0.00001 | <0.00001 | 0.00174 | 0.00044 | 0.01295 |
| (8) general additive $G(t|\boldsymbol{x})$ TI, inter. | <0.00001 | <0.00001 | <0.00001 | 0.00039 | 0.00946 |
| (9) general additive $G(t|\boldsymbol{x})$ TD main, inter. | <0.00001 | <0.00001 | 0.00181 | 0.00028 | 0.01350 |
| (10) general additive $G(t|\boldsymbol{x})$ TD inter. | <0.00001 | <0.00001 | 0.00178 | 0.00196 | 0.00902 |

TI = time-independent, TD = time-dependent, inter. = interaction.
<0.00001 indicates negligible decomposition error.

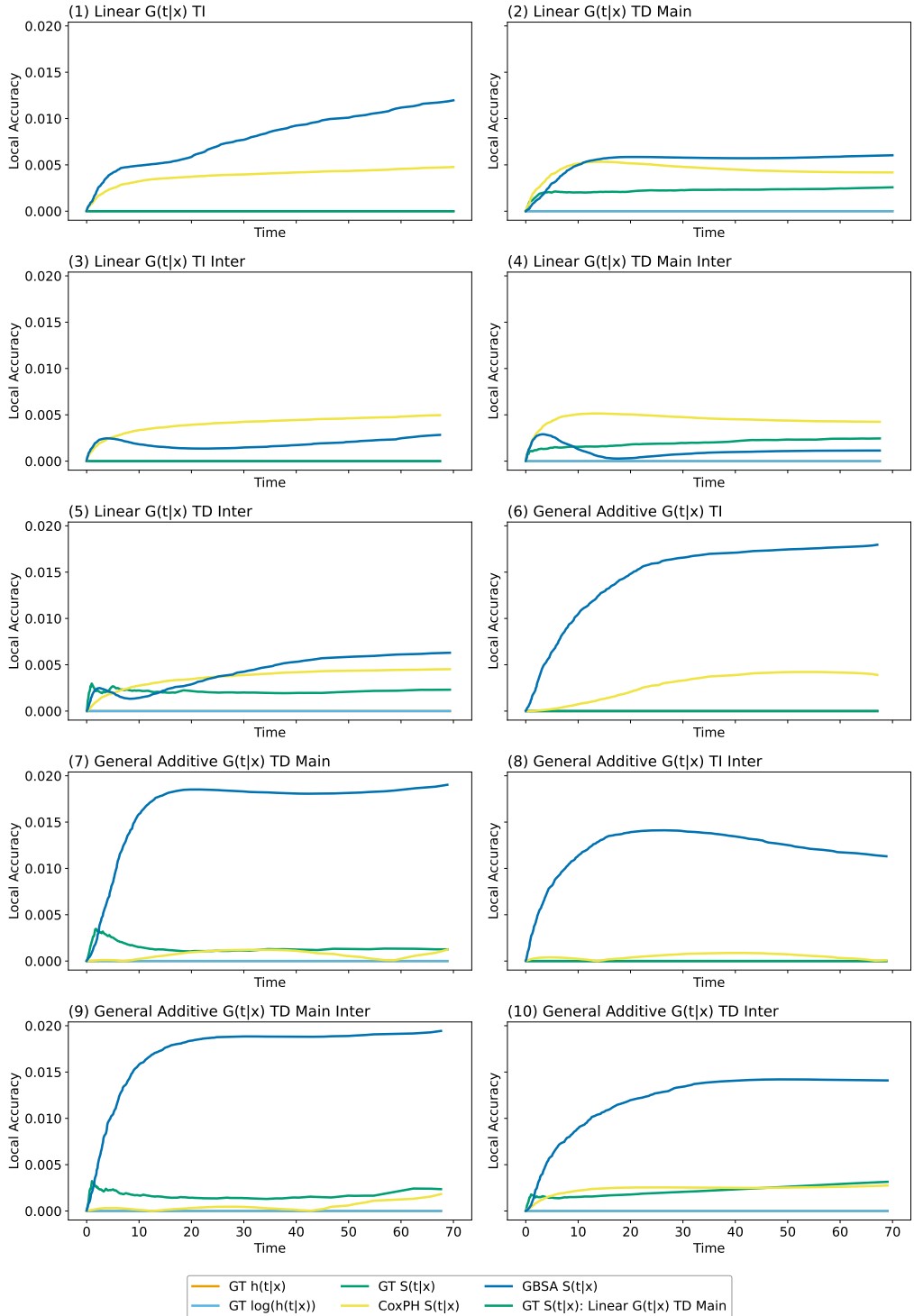

*Figure B.6.* Local accuracy curves over time (X-axis) computed for the ground-truth (GT) hazard, log-hazard, survival and predicted survival functions of CoxPH and GBSA models (colored curves, Y-axis) for all observations from the 10 simulation scenarios. Local accuracy is lower for ground-truth functions than for model prediction functions.

## B.2. Experiments with Simulated Data and Risk Scores with Dependent Features

**Correlated features with marginal SurvFD.** We simulate the same ten scenarios as described in Sec. B.1.1 under identical conditions, except that the three features $x_1, x_2, x_3$ are now generated based on a multivariate normal distribution $(x_1, x_2, x_3) \sim \mathcal{N}(\mathbf{0}, \Sigma)$, where $\Sigma$ has unit variances and pairwise correlation $\rho \in \{0, 0.2, 0.5, 0.9\}$. For each experimental setting, we compute the exact order-2 SurvSHAP-IQ decomposition (i.e., interventional SurvSHAP-IQ; cf. Eq. (10)) using log-hazard SHAP value functions evaluated on the full dataset. To assess how dependencies between features impact the decomposition and the separation of time-dependent and time-independent effects, the resulting attributions in Table B.4 are averaged over time and their standard deviations are computed, for the same single randomly selected observation as in Sec. B.1.1 $[x_1 = -1.265, x_2 = 2.416, x_3 = -0.644]$. We observe that especially in scenarios without interaction effects (scenarios **(1 - 2)** and **(6 - 7)**), feature dependence has minimal impact on the attribution results, as indicated by similar mean and standard deviation values across correlation levels. In contrast, for scenarios with time-dependent interactions in the ground truth, the model can sometimes misattribute part of the interaction effect to lower-order main effects. For instance, in scenario **(5)**, the time-dependent interaction between $x_1$ and $x_3$ is partially attributed to the main effect of $x_1$ for higher feature correlations ($\rho = 0.5$ and $\rho = 0.9$), reflected in increased standard deviations and deviations in the mean values. However, this misattribution is not universal: in scenario **(10)**, the time dependence is correctly attributed solely to the interaction $x_1 x_3$, even for higher correlations. Again, we note that analyzing a different observation leads to qualitatively symmetric conclusions. In summary, feature dependencies introduce an additional layer of complexity for interpretation. Crucially, using the joint marginal distribution as the reference distribution in SurvFD yields attributions that are "true to the model", in that they directly reflect main and interaction effects learned by the model.

**Marginal SurvFD vs. conditional SurvFD.** Additionally, we simulate a simple scenario to illustrate the difference between marginal and conditional FD stated in Th. 3.6. We define the log-risk function $G(t|\boldsymbol{x})$ as:

$$G(t|\boldsymbol{x}) = x_1 \beta_1 \cdot \log(t+1) + x_2 \beta_2,$$

with the hazard defined as in Eq. (B34). Three features $(x_1, x_2, x_3)$ are drawn from a multivariate normal distribution with zero mean and unit variances, where $x_1$ and $x_3$ are strongly correlated with $\rho(x_1, x_3) = 0.9$, while $x_2$ is independent of both. The model parameters are fixed to $h_0(t) = 0.03$, $\beta_1 = 0.8$, and $\beta_2 = -0.4$. Event times $t^{(i)}$ are generated using the approaches of Bender et al. (2005) for proportional hazards and Crowther & Lambert (2013) for non-proportional hazards models, as implemented in the simsurv package (Brilleman et al., 2021). Right-censoring is applied by setting $t^{(i)} = 70$ for all observations with $t^{(i)} \geq 70$, resulting in a final sample size of $n = 1,000$.

We compute the exact order-2 SurvSHAP-IQ decomposition using both interventional (marginal FD) and observational (conditional FD) SurvSHAP-IQ with log-hazard SHAP value functions evaluated on the full dataset (see Fig. B.7) for the same randomly selected observation as in the previous experiments. Consistent with Thm. 3.6, interventional SurvSHAP-IQ correctly recovers the model's effects ($G(t|\boldsymbol{x})$), i.e., a time-dependent effect for $x_1$, a time-independent negative effect for $x_2$, and no contribution from $x_3$, as well as no interaction effects. In contrast, observational SurvSHAP-IQ is influenced by the correlation between $x_1$ and $x_3$ and consequently produces time-dependent main and interaction effects involving all three features $x_1, x_2$, and $x_3$, reflecting the effect induced by conditioning on correlated features. This behavior is expected for conditional FD, as it produces explanations that are "true to the data", reflecting dependencies present in the data distribution.

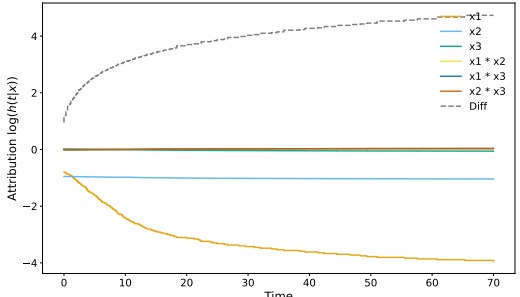 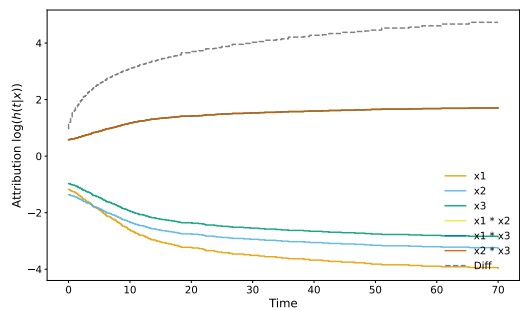

*Figure B.7.* Exact SurvSHAP-IQ decomposition curves for a selected observation $[-1.2650, 2.4162, -0.6436]$ computed on the ground-truth log-hazard functions. **Left:** Interventional SurvSHAP-IQ. **Right:** Conditional SurvSHAP-IQ. Colored curves show feature- and interaction-specific log-hazard attributions, dashed grey shows the individual–mean log-hazard difference.

*Table B.4.* Time-wise mean and standard deviation of the exact SurvSHAP-IQ decomposition results for a selected observation $[-1.2650, 2.4162, -0.6436]$ computed on the ground-truth log-hazard functions for all ten simulation scenarios for four different correlation levels ($\rho = 0, 0.2, 0.5, 0.9$). Each cell reports the mean over time of the SurvSHAP-IQ attributions, with the standard deviation given in parentheses.

| Scenario | $\rho$ | $x_1$ | $x_2$ | $x_3$ | $x_1 * x_2$ | $x_1 * x_3$ | $x_2 * x_3$ |
|---|---|---|---|---|---|---|---|
| **(1)** | 0.0 | $-0.5167(0.0073)$ | $-1.9000(0.0040)$ | $0.3750(0.0053)$ | $0.0015(0.0026)$ | $0.0015(0.0026)$ | $0.0015(0.0026)$ |
| | 0.2 | $-0.5245(0.0079)$ | $-1.9449(0.0024)$ | $0.3619(0.0042)$ | $0.0040(0.0028)$ | $0.0040(0.0028)$ | $0.0040(0.0028)$ |
| | 0.5 | $-0.5061(0.0089)$ | $-1.9581(0.0012)$ | $0.3860(0.0029)$ | $0.0069(0.0022)$ | $0.0069(0.0022)$ | $0.0069(0.0022)$ |
| | 0.9 | $-0.4979(0.0137)$ | $-1.9305(0.0003)$ | $0.4067(0.0008)$ | $-0.0065(0.0024)$ | $-0.0065(0.0024)$ | $-0.0065(0.0024)$ |
| **(2)** | 0.0 | $-1.2849(0.3242)$ | $-1.9084(0.0103)$ | $0.3709(0.0107)$ | $0.0042(0.0051)$ | $0.0042(0.0051)$ | $0.0042(0.0051)$ |
| | 0.2 | $-1.2688(0.2982)$ | $-1.9437(0.0090)$ | $0.3580(0.0100)$ | $0.0010(0.0047)$ | $0.0010(0.0047)$ | $0.0010(0.0047)$ |
| | 0.5 | $-1.2745(0.2850)$ | $-1.9512(0.0078)$ | $0.3902(0.0072)$ | $0.0059(0.0030)$ | $0.0059(0.0030)$ | $0.0059(0.0030)$ |
| | 0.9 | $-1.4045(0.3241)$ | $-1.9261(0.0050)$ | $0.4021(0.0037)$ | $0.0020(0.0009)$ | $0.0020(0.0009)$ | $0.0020(0.0009)$ |
| **(3)** | 0.0 | $-0.5292(0.0093)$ | $-1.9179(0.0069)$ | $0.3692(0.0118)$ | $0.0076(0.0040)$ | $-0.7246(0.0147)$ | $0.0076(0.0040)$ |
| | 0.2 | $-0.3951(0.0058)$ | $-1.9451(0.0092)$ | $0.5337(0.0076)$ | $-0.0007(0.0050)$ | $-0.8541(0.0125)$ | $-0.0007(0.0050)$ |
| | 0.5 | $-0.0569(0.0039)$ | $-1.9513(0.0048)$ | $0.8032(0.0065)$ | $0.0043(0.0036)$ | $-1.1692(0.0125)$ | $0.0043(0.0036)$ |
| | 0.9 | $0.3315(0.0013)$ | $-1.9312(0.0079)$ | $1.2310(0.0031)$ | $0.0010(0.0044)$ | $-1.5902(0.0171)$ | $0.0010(0.0044)$ |
| **(4)** | 0.0 | $-1.2694(0.3575)$ | $-1.8941(0.0147)$ | $0.3625(0.0230)$ | $0.0008(0.0063)$ | $-0.6957(0.0193)$ | $0.0008(0.0063)$ |
| | 0.2 | $-1.1529(0.3533)$ | $-1.9246(0.0130)$ | $0.5527(0.0192)$ | $-0.0113(0.0058)$ | $-0.8689(0.0211)$ | $-0.0113(0.0058)$ |
| | 0.5 | $-0.8682(0.2893)$ | $-1.9602(0.0105)$ | $0.8177(0.0161)$ | $0.0033(0.0042)$ | $-1.1774(0.0234)$ | $0.0033(0.0042)$ |
| | 0.9 | $-0.6189(0.2559)$ | $-1.9138(0.0135)$ | $1.2438(0.0118)$ | $-0.0070(0.0050)$ | $-1.6047(0.0269)$ | $-0.0070(0.0050)$ |
| **(5)** | 0.0 | $-0.6205(0.0395)$ | $-1.8934(0.0298)$ | $0.3462(0.0277)$ | $-0.0042(0.0132)$ | $-1.6098(0.6748)$ | $-0.0042(0.0132)$ |
| | 0.2 | $-0.2137(0.0722)$ | $-1.9351(0.0361)$ | $0.7980(0.0712)$ | $-0.0109(0.0144)$ | $-2.0708(1.0903)$ | $-0.0109(0.0144)$ |
| | 0.5 | $0.6105(0.2621)$ | $-2.0000(0.0383)$ | $1.4463(0.2373)$ | $0.0251(0.0143)$ | $-2.9425(1.6941)$ | $0.0251(0.0143)$ |
| | 0.9 | $1.7828(0.8613)$ | $-1.8874(0.0813)$ | $2.6366(0.7617)$ | $-0.0237(0.0320)$ | $-4.2969(2.9830)$ | $-0.0237(0.0320)$ |
| **(6)** | 0.0 | $0.2493(0.0028)$ | $-0.5200(0.0037)$ | $0.3690(0.0022)$ | $0.0024(0.0016)$ | $0.0024(0.0016)$ | $0.0024(0.0016)$ |
| | 0.2 | $0.2771(0.0036)$ | $-0.5265(0.0049)$ | $0.3608(0.0029)$ | $-0.0015(0.0024)$ | $-0.0015(0.0024)$ | $-0.0015(0.0024)$ |
| | 0.5 | $0.2528(0.0034)$ | $-0.5270(0.0035)$ | $0.3952(0.0019)$ | $-0.0058(0.0018)$ | $-0.0058(0.0018)$ | $-0.0058(0.0018)$ |
| | 0.9 | $0.2346(0.0048)$ | $-0.5319(0.0036)$ | $0.4016(0.0020)$ | $0.0030(0.0022)$ | $0.0030(0.0022)$ | $0.0030(0.0022)$ |
| **(7)** | 0.0 | $0.5912(0.0620)$ | $-0.5017(0.0132)$ | $0.3872(0.0123)$ | $-0.0069(0.0048)$ | $-0.0069(0.0048)$ | $-0.0069(0.0048)$ |
| | 0.2 | $0.6790(0.0839)$ | $-0.5178(0.0145)$ | $0.3647(0.0110)$ | $-0.0076(0.0057)$ | $-0.0076(0.0057)$ | $-0.0076(0.0057)$ |
| | 0.5 | $0.5864(0.0643)$ | $-0.5292(0.0160)$ | $0.3993(0.0146)$ | $-0.0023(0.0065)$ | $-0.0023(0.0065)$ | $-0.0023(0.0065)$ |
| | 0.9 | $0.5932(0.0628)$ | $-0.5228(0.0144)$ | $0.4028(0.0137)$ | $-0.0039(0.0059)$ | $-0.0039(0.0059)$ | $-0.0039(0.0059)$ |
| **(8)** | 0.0 | $0.0745(0.0046)$ | $-0.4871(0.0168)$ | $0.3809(0.0036)$ | $1.4782(0.0170)$ | $0.1313(0.0035)$ | $-0.0047(0.0027)$ |
| | 0.2 | $0.0852(0.0039)$ | $-0.4774(0.0140)$ | $0.3684(0.0033)$ | $1.4849(0.0133)$ | $0.1462(0.0036)$ | $-0.0030(0.0020)$ |
| | 0.5 | $0.2817(0.0033)$ | $-0.2313(0.0227)$ | $0.4091(0.0018)$ | $1.2423(0.0162)$ | $0.1055(0.0038)$ | $0.0002(0.0013)$ |
| | 0.9 | $0.3982(0.0022)$ | $-0.1146(0.0138)$ | $0.3776(0.0005)$ | $1.1111(0.0077)$ | $0.1738(0.0059)$ | $0.0021(0.0006)$ |
| **(9)** | 0.0 | $0.4128(0.0729)$ | $-0.4833(0.0214)$ | $0.3695(0.0125)$ | $1.4678(0.0179)$ | $0.1412(0.0060)$ | $0.0033(0.0049)$ |
| | 0.2 | $0.4809(0.0817)$ | $-0.4745(0.0240)$ | $0.3592(0.0114)$ | $1.4732(0.0152)$ | $0.1542(0.0067)$ | $0.0081(0.0051)$ |
| | 0.5 | $0.6504(0.0551)$ | $-0.2314(0.0347)$ | $0.4083(0.0113)$ | $1.2472(0.0211)$ | $0.1054(0.0068)$ | $0.0001(0.0045)$ |
| | 0.9 | $0.7944(0.0545)$ | $-0.1038(0.0290)$ | $0.3779(0.0060)$ | $1.1042(0.0158)$ | $0.1712(0.0046)$ | $-0.0004(0.0027)$ |
| **(10)** | 0.0 | $-0.2919(0.0837)$ | $-0.4879(0.0109)$ | $0.3833(0.0034)$ | $1.4744(0.0103)$ | $0.3383(0.0342)$ | $0.0004(0.0030)$ |
| | 0.2 | $-0.2986(0.0862)$ | $-0.4734(0.0128)$ | $0.3700(0.0028)$ | $1.4826(0.0111)$ | $0.3714(0.0364)$ | $-0.0040(0.0030)$ |
| | 0.5 | $-0.0843(0.0642)$ | $-0.2266(0.0139)$ | $0.4266(0.0024)$ | $1.2368(0.0119)$ | $0.3058(0.0295)$ | $0.0016(0.0023)$ |
| | 0.9 | $-0.0658(0.0793)$ | $-0.1430(0.0145)$ | $0.3493(0.0010)$ | $1.1327(0.0084)$ | $0.4381(0.0445)$ | $0.0159(0.0042)$ |

## B.3. Experiments with Real-world Applications

**Datasets.** We use the following well-established datasets used in research on survival analysis (Drysdale, 2022):

- `actg` (Hammer et al., 1997) for predicting time to AIDS diagnosis or death (in days), with 1151 observations and 4 numeric features: the patient's `age` at the study enrollment (in years), baseline `cd4` cell count in blood, months of prior Zidovudine medicine use (in months), and Karnofsky performance scale (`karnof`), where: 100 means *normal, no complaint, no evidence of disease*; 90 means *normal activity possible, minor signs/symptoms of disease*; 80 means *normal activity with effort, some signs/symptoms of disease*; and 70 means *cares for self, normal activity/active work not possible*.

- `Bergamaschi` (Bergamaschi et al., 2006) for predicting breast cancer survival based on gene expression data, with 82 observations and 10 numeric features.

- `smarto` (Simons et al., 1999) for predicting survival in patients with either clinically manifest atherosclerotic vessel disease, or marked risk factors for atherosclerosis, based on 3873 observations and 17 numeric features.

- `support2` (Knaus et al., 1995) for predicting survival of seriously ill hospitalized adults based on clinical data, physiology scores, and clinically valid survival estimates, with 9105 observations and 24 numeric features.

- `phpl04K8a` (Shedden et al., 2008) for predicting lung cancer (adenocarcinomas) survival based on gene expression data, with 442 observations and 20 numeric features.

- `nki70` (Van De Vijver et al., 2002) for predicting breast cancer survival based on gene expression data, with 144 observations and 76 features, incl. 70 genomic features.

Furthermore, we extend the analysis to include the recently released dataset for predicting the survival of eye cancer (uveal melanoma), based on routine histologic and clinical variables (Donizy et al., 2022). In this case, several machine learning algorithms were applied, along with feature importance methods to explain their results. There are 150 observations (patients) in the training set and 77 in the validation set. We focus on 9 numerical features, including: the patient's `age` (in years), largest tumor diameter at its base (`max tumor diameter`), `mitotic rate` per $mm^2$, `tumor thickness`, and various cell `nucleus` measurements.

**Models.** We train models using the `scikit-survival` package (Pölsterl, 2020). For `actg`, a random survival forest with `n_estimators = 300` and `max_depth = 6` achieves an out-of-bag C-index score of 0.723 (0.914 on the training set). Similarly, for `nki70`, it achieves an out-of-bag C-index score of 0.726 (0.954 on the training set). For the approximator benchmark with `Bergamaschi`, `smarto`, `support2` and `phpl04K8a`, we train baseline RSF models with `n_estimators = 100` and `max_depth = 3`, achieving a training C-index score of 0.897, 0.737, 0.725, 0.729, respectively. Note that we do not aim for high performance or tune the models, since our goal is not to interpret the explanations themselves in this case. Following Donizy et al. (2022), we train a GBSA model with `n_estimators = 100`, `max_depth = 4`, and `learning_rate = 0.05`, which achieves a C-index score of 0.927 on the training set and 0.758 on the validation set.

**Explanations.** To compute and approximate SurvSHAP-IQ explanations, we build upon the open-source `shapiq` software (Muschalik et al., 2024). For `actg`, we use the exact computer with marginal imputation to explain the survival predictions in 41 evenly-distributed timepoints for a random sample of 200 observations. For uveal melanoma, we use the regression-based approximator with a budget of $2^9$ and marginal imputation to explain the survival predictions in 41 evenly distributed timepoints for the validation set of 77 observations. For `nki70`, we use the regression-based approximator with a budget of $2^{15}$ and marginal imputation to explain survival in 31 timepoints for all 144 observations. For the four benchmark datasets, we analyse four different approximators: Monte Carlo (Fumagalli et al., 2023), permutation-based (Tsai et al., 2023), SVARM (Kolpaczki et al., 2024), and regression-based (Fumagalli et al., 2024). For `Bergamaschi` with 10 features, we use marginal imputation to explain survival predictions in 41 evenly distributed timepoints for all 82 observations. We compute the 'ground truth' explanation exactly, allowing for the measurement of approximation error for budgets $= \{2^6, 2^7, 2^8, 2^9\}$. For the three larger datasets (`smarto`, `support2`, `phpl04K8a`), we simplify the evaluation to make computing the 'ground truth' exactly feasible. Specifically, we restrict the set of features to 16, use baseline imputation with the mean, and explain the model at 11 evenly distributed timepoints for a random sample of 20 observations. Consequently, we measure the approximation error for budgets $= \{2^9, 2^{10}, 2^{11}, 2^{12}, 2^{13}\}$. We repeat all calculations 30 times and report the average with the standard error.

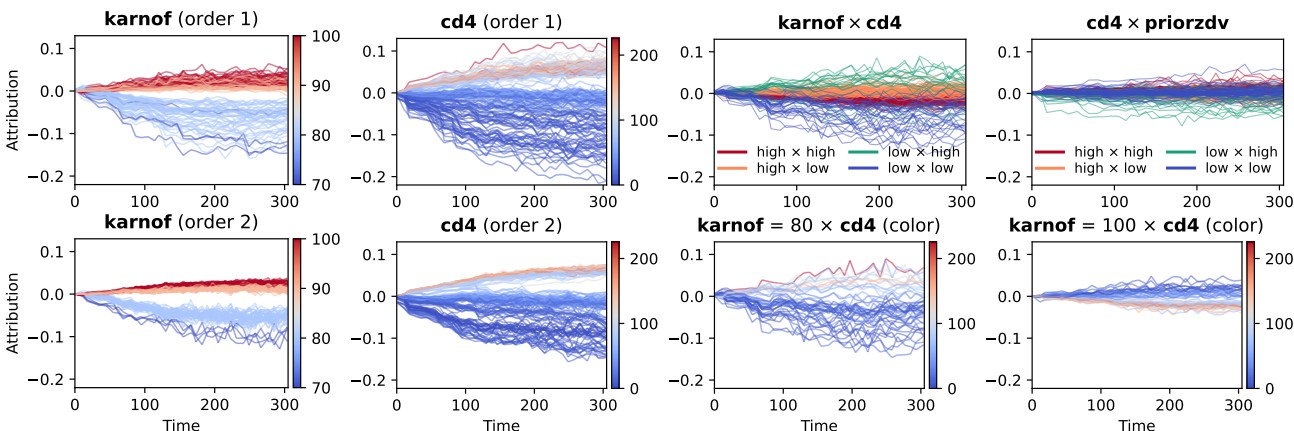

*Figure B.8.* Explanation of an RSF model predicting survival in patients treated for the HIV-1 infection. **Top left:** Shapley value attributions (order 1) for multiple observations (curves) with different feature values (in color). **Bottom left:** SurvSHAP-IQ individual effects (order 2). We observe that these are smoother compared to attribution curves, indicating distincter relationships with feature values. **Right:** SurvSHAP-IQ interaction effects (order 2). We observe a significant interaction between the `karnof` and `cd4` features, which is 'hidden' in the Shapley value simplified explanation. The remaining interaction effects are shown in Figure B.9.

**A simple illustrative example for `actg`.** We first explain a random survival forest (RSF) predicting time to AIDS diagnosis or death in HIV-1 patients using the `actg` dataset (Hammer et al., 1997). The RSF (300 trees, depth 6) trained on four numerical features (`karnof`, `cd4`, `priorzdv`, `age`) achieves an out-of-bag C-index of $0.723$. Figure B.8 shows SurvSHAP-IQ explanations for 200 patients. The model relies most on `karnof` and `cd4`, where low values (blue) reduce survival predictions and higher values (grey–red) increase them. Second-order effects exhibit lower in-group variance than first-order ones—particularly across the four `karnof` color groups—because the decomposition accounts for interactions (Fig. B.8, right). Our method recovers the strong `karnof` × `cd4` interaction, identified by the RSF model, suggesting interpretable models like CoxPH could benefit from explicitly including such terms. We defer visualizations of the remaining, less-important interaction effects to Figure B.9.

**Lack of important gene interactions in `nki70`.** We run the SurvSHAP-IQ approximation on an RSF model with 76 (genomic) features, which effectively comprises $\binom{76}{2} = 2850$ interaction terms, using a relatively large budget of $2^{15} = 32768$ sampled coalitions. We find that no interaction between two features is more important than their first-order effects, i.e., no interaction appears among the top-76 important terms as measured by either the mean absolute attribution value or the standard deviation. Such a result effectively supports the use of an *additive* CPH model for predicting breast cancer survival, as in the original work (Van De Vijver et al., 2002). Figure B.10 shows exemplary survival model explanations for the four most important features and pair-wise interactions.

**Extended results.** Figure B.11 extends Figure 5, showing the SurvSHAP-IQ explanation of the GBSA model for the uveal melanoma task. Figure B.12 extends Figure 7, showing results of the approximation benchmark. Additional implementation details and code are provided in the online supplement.

## B.4. Experiments with multi-modal TCGA-BRCA Dataset

In this multi-modal example, we use data from *The Cancer Genome Atlas Breast Invasive Carcinoma* (TCGA-BRCA) project (Lingle et al., 2016), which is provided in preprocessed form via Kaggle[3]. We train a multi-modal DeepHit model and use SurvSHAP-IQ to analyze interactions in the survival predictions of individual patients. Additional details and code are provided in the online supplement.

### B.4.1. DATASET

The TCGA-BRCA dataset comprises high-resolution histopathological whole-slide images (WSIs) of tissue samples and tabular clinical features, including age, surgical procedure, tumor characteristics, and TNM-based staging information (Amin et al., 2017). In total, dataset includes 1,097 patients, of whom 990 have both patched WSIs and complete clinical data.

---

[3]Available at `https://www.kaggle.com/datasets/jmalagontorres/tcga-brca-survival-analysis`

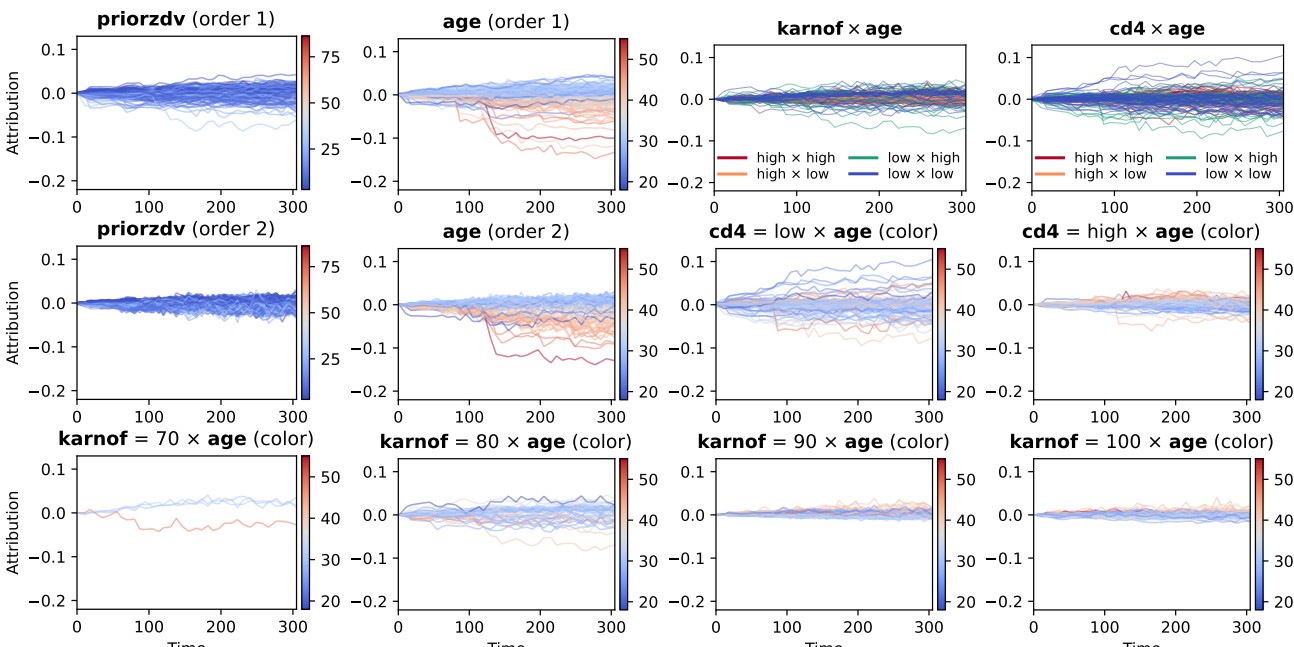

*Figure B.9.* Extended Figure B.8. The remaining interaction effects in an explanation of an RSF model predicting the survival in patients treated for the HIV-1 infection.

**Images.** The image modality consists of extremely high-resolution histopathological WSIs of tissue samples, often spanning tens of thousands of pixels. Since processing full slides is computationally infeasible, standard practice in computational pathology is to divide each slide into smaller, non-overlapping patches. Accordingly, the Kaggle dataset provides 1000×1000 patches at 20×magnification, primarily containing tissue. The number of patches varies substantially across patients (50–2,343), with an average of 523 patches per patient. We used the UNI2-h feature extractor (Chen et al., 2024), a large vision transformer (ViT-H/14), which was pretrained on over 100 million histopathology images from more than 20 tissue types, to encode each patch into a compact numerical representation. The encoder maps each resized 224×224 patch to a 1,536-dimensional feature vector capturing the morphological content of the tissue region. Features are extracted once and stored, so training and explaining the survival model only requires loading precomputed vectors rather than processing raw image patches.

**Clinical tabular features.** In addition to image data, we use 8 clinical variables, resulting in 21 features after dummy encoding the categorical ones (see Table B.5). These **8 variables correspond to the players** in our Shapley-based analysis (Sec. B.4.4). Categorical variables, such as tumor staging, are dummy encoded using clinically appropriate reference categories (e.g., T2 for T-Stage, N0 for N-Stage, and Stage II for overall staging (Amin et al., 2017)). The two continuous variables (patient age and number of examined lymph nodes) are standardized to the range $[0, 1]$.

*Table B.5.* Description of the clinical features of the TCGA-BRCA dataset. Dummy-encoded variables use the listed reference category as baseline.

| Feature | Levels | Type | Reference |
|---|---|---|---|
| Age | – | Continuous | – |
| Lymph Nodes | Lymph nodes examined | Count | – |
| Surgery | lumpectomy, simple mastectomy, other | Categorical | Radical mast. |
| Menopause | Indeterminate, pre-menopausal | Categorical | Post-menopausal |
| T-Stage | T1, T3, T4, TX | Categorical | T2 |
| N-Stage | N1, N2, N3, NX | Categorical | N0 |
| M-Stage | M1, MX | Categorical | M0 |
| Stage | I, III, IV, X | Categorical | Stage II |

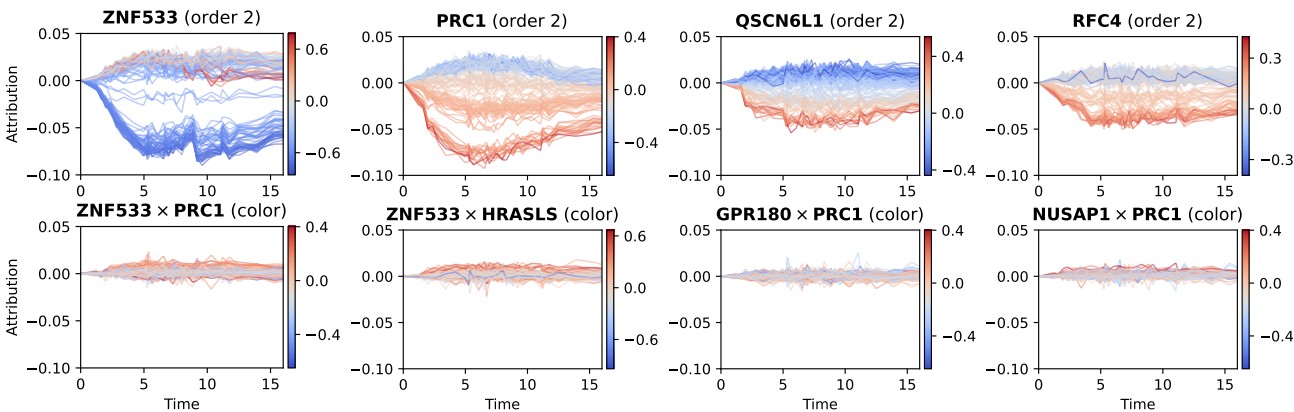

*Figure B.10.* Explanation of an RSF model predicting breast cancer survival. **Top:** SurvSHAP-IQ individual effects (order 2) for four most important gene expressions. **Bottom:** SurvSHAP-IQ interaction effects (order 2) for four most important gene interactions.

### B.4.2. MODEL

The central challenge in image-based pathology is aggregating a varying number of patch features into a single patient-level representation (i.e., a bag of patches per patient). We use gated attention from the field of multiple-instance learning (Ilse et al., 2018), which learns an importance weight $\alpha_k$ for each patch. The patient-level image representation is then a weighted sum, i.e., $z_{\text{img}} = \sum_k \alpha_k e_k$, where $e_k$ is the 1,536-dimensional feature vector from the UNI2-h feature extractor. Patches with higher weights contribute more than others, or more intuitively, the model learns to focus on survival-relevant tissue regions. This allows us to use an arbitrary number of patches for a single patient's prediction. We use six patches for the prediction, which are the **top-6 patches** according to their attention weights to serve **as players** in SurvSHAP-IQ.

The image representation is combined with clinical features via concatenation (both are mapped to a 64-dimensional vector using dense neural networks), including element-wise interactions, i.e., $z_{\text{fused}} = (z_{\text{img}}, z_{\text{tab}}, z_{\text{img}} \odot z_{\text{tab}})$. The fused representation is mapped to a *DeepHit* head (Lee et al., 2018) using the `pycox` package (Kvamme et al., 2019), providing discrete-time survival probabilities. The DeepHit architecture provides the following survival quantities:

- **Probability mass function (PMF):** $p(t|\boldsymbol{x}) = \mathbb{P}(T = t|\boldsymbol{x})$, the probability that the event occurs exactly at time $t$.

- **Survival function:** $S(t) = \mathbb{P}(T > t|\boldsymbol{x})$, the probability of surviving beyond time $t$.

Since DeepHit predicts these quantities at discrete time points $t_1 < t_2 < \cdots < t_L$, they are related by

$$S(t|\boldsymbol{x}) = 1 - \sum_{t_i < t} p(t_i|\boldsymbol{x}).$$

### B.4.3. TRAINING

We train the model using the DeepHit loss (Lee et al., 2018), which combines a partial negative log-likelihood encouraging high probability for the observed event time with a ranking loss that penalizes incorrect orderings of predicted survival times across patients (with weight $\alpha = 0.4$). Survival times are discretized into 16 bins using quantile-based cuts, ensuring that each bin contains approximately the same number of events. We split the 990 patients into training (70%, 693 patients), validation (15%, 149 patients), and test (15%, 148 patients) sets using stratified splitting by event status. We use AdamW optimization with a learning rate of $5 \cdot 10^{-5}$, weight decay of $5 \cdot 10^{-3}$, and a batch size of 16. Training runs for at most 200 epochs with early stopping (patience 25, monitored on validation loss). Since patients have varying numbers of patches, we randomly sample up to 150 patches per patient at each epoch, providing implicit data augmentation as each epoch sees a different subset of patches. Dropout is applied at a rate of 0.5 throughout the network, with a reduced rate of 0.25 before the gated attention mechanism to preserve patch-level information. With this setup, we achieve a C-index score of 0.7375 and an integrated Brier score of 0.1512 on the test set.

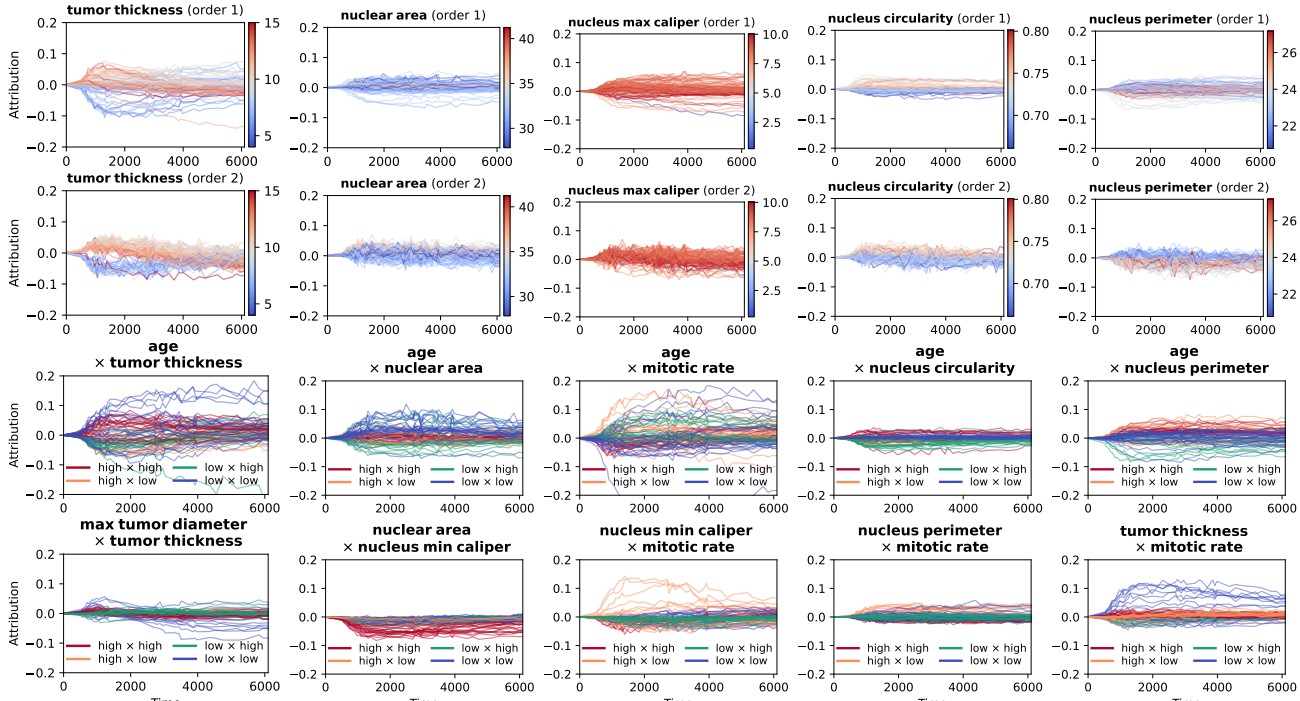

*Figure B.11.* Extended Figure 5 with a narrowed Y-axis scale for readability. The remaining attribution and interaction effects in an explanation of a GBSA model predicting the survival of patients diagnosed with uveal melanoma, a type of eye cancer.

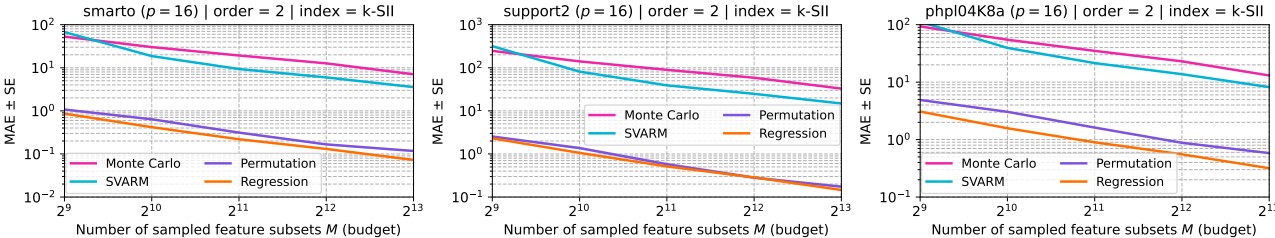

*Figure B.12.* Extended Figure 7. Mean absolute error of different SurvSHAP-IQ approximators as a function of budget for the three survival tasks.

### B.4.4. EXPLANATION WITH SURVSHAP-IQ

To compute Shapley interaction explanations, we build upon the open-source `shapiq` package (Muschalik et al., 2024). We define the cooperative game with **14 players**: the 6 highest-attention patches plus the 8 clinical features (Table B.5). We use the exact computation with marginal imputation to explain survival predictions at one of the 16 discrete time points of the DeepHit architecture. For marginal imputation, we use all other patches from the same patient for image modality and the dataset-wide distribution for the clinical features. In Figure 6, the lymph nodes examined feature and M-Stage are omitted due to negligible effects.

### B.5. Hardware and Software Setup

Experiments were conducted using a 64-bit Linux platform running Ubuntu 22.04 LTS with two AMD EPYC Genoa 9534 64-Core processors (128 cores, 256 threads total), 1.5 terabytes of RAM, and eight NVIDIA RTX 6000 Ada Generation GPUs (each with 48 GB memory). Additionally, for the TCGA-BRCA dataset, a 64-bit Linux platform running Ubuntu 24.04 LTS with an AMD Ryzen Threadrippper 3960X (24 cores, 48 threads) CPU including 256 gigabyte RAM and two NVIDIA Titan RTX GPUs was used.

