# OpenReview forum: "Functional Decomposition and Shapley Interactions for Interpreting Survival Models"
_ICML.cc/2026/Conference — ICML 2026 regular_

### Official Review · Reviewer_MKCW · 2026-03-02

**Soundness:** 3
**Presentation:** 2
**Significance:** 2
**Originality:** 3
**Overall Recommendation:** 4
**Confidence:** 3

**Summary:**

The paper studies how to decompose and interpret feature interactions in survival models, where predictions such as hazard and survival functions are time-dependent and non-additive. It introduces Survival Functional Decomposition (SurvFD), which extends functional ANOVA to survival functions by separating predictions into pure main and higher-order interaction effects while distinguishing time-dependent from time-independent components. It analyzes how this structure changes across log-hazard, hazard, and survival scales and under feature dependence. Building on this theory, the authors propose SurvSHAP-IQ, a time-indexed Shapley interaction method that quantifies higher-order interactions at specific timepoints and provides locally accurate additive explanations. Simulations and real-world cancer survival experiments demonstrate how the approach recovers and visualizes interaction structures in both classical and deep survival models.

**Compliance With Llm Reviewing Policy:**

Affirmed.

**Key Questions For Authors:**

1. How should practitioners choose between explaining the log-hazard, hazard, or survival scale in applied settings, given that SurvFD shows scale-dependent, transformation-induced interactions that may not reflect the underlying risk score?

2. In the presence of strong feature dependence, what practical guidance do you recommend for selecting marginal versus conditional reference distributions when the goal is clinical or scientific interpretation rather than model debugging?

3. How do you see SurvFD and SurvSHAP-IQ extending to non-multiplicative hazard formulations or competing-risks and recurrent-event settings, where the log-hazard decomposition no longer has a clear additive structure?

**Limitations:**

The authors have adequately discussed the limitations of this paper.

**Strengths And Weaknesses:**

Strengths:

1. The idea of extending ANOVA-based functional decomposition to survival analysis seems novel.

2. It extends Shapley interaction method to survival outcomes.

Weaknesses:

1. The method is restricted to multiplicative hazards and the second-order effect of features in rather small datasets (up to 70 features). These restrictions are grounded in the design of the method and are challenging to overcome.

2. The non-linear transformation of hazard/survival may induce interactions that are not genuine, e.g., from the underlying G(t|x).

3. Even when interactions can be found and estimated, interpreting and visualising these are still difficult.

4. If features are correlated/dependent, the explanation can vary depending on the reference distribution. This compound with the weaknesses 2 and 3.

---

> ### Author Rebuttal · Authors · 2026-03-31
>
> We thank the reviewer for the detailed feedback and recognizing the novelty of extending functional decomposition and Shapley interactions to survival outcomes.
>
> We address each point below.
>
> **Weaknesses**
>
> **W1: Restriction to multiplicative hazards.** We agree and acknowledge this limitation (Sec. 5). However, multiplicative hazard models remain the dominant paradigm in survival analysis, so our framework covers a practically relevant class. Importantly, only the theoretical decomposition results require multiplicative hazards; **SurvFD and SurvSHAP-IQ are otherwise model-agnostic** (Sec. 4.2).
>
> **W1: Second-order effects and scalability.** Our method supports arbitrary interaction orders. We computed three-way interactions for smaller datasets, where we found negligible magnitudes relative to pairwise interactions, consistent with the effect hierarchy principle [a]. We will clarify this in the revision. For scalability, we demonstrated SurvSHAP-IQ on 76 features ($2,850$ pairwise interactions) using efficient approximation algorithms (Figs. 7, B.12). Recent advances in scaling Shapley interactions to vision-language models [b] and LLMs [c] suggest promising directions for extending to thousands of features (e.g., genomics). While scaling remains an open challenge, our work takes an important step in offering a principled perspective on survival explanations through interactions. We will make this more explicit in the discussion section.
>
> **W2: Non-linear transformations may induce non-genuine interactions.** Prop. 3.5 and Cor. 3.4 formally characterize when scale transformations induce interactions not present in $G(t∣x)$. Rather than a limitation, we view this as one of our key contributions: practitioners should be aware that survival-scale interactions may be artifacts of transformation.
>
> **W3: Interpreting and visualizing interactions is difficult.** We agree this is challenging and an open HCI research problem [d]. Our paper contributes time-indexed attribution curves  (Fig. 4), grouped interaction plots (Fig. 6), and network plots (Fig. 5). More intuitive visualizations are an important future work direction, which we will add to the discussion.
>
> **W4: Feature dependence.** See Q2 below.
>
> **Questions**
>
> **Q1: Choosing between scales.** The appropriate scale should be determined by the problem: survival for clinical prognoses, hazard for instantaneous risk monitoring. The SurvFD decomposition can only recover ground-truth effects on the log-hazard scale (Thms 3.2, 3.3). Hazard/survival scales introduce transformation-induced interactions (Cor. 3.4, Prop. 3.5), but remain relevant for interpreting predictions.
>
> *Recommendation:* Compute explanations on the scale of interest for the problem at hand. While interactions appearing on the hazard/survival scale may be transformation-induced, they are relevant for interpreting results. We will add guidance on scale selection to the revision.
>
> **Q2: Marginal vs. conditional reference distributions.** In the presence of strong feature dependence, marginal and conditional reference distributions are both valid but reflect different interpretations: marginal approaches are “true to the model,” while conditional approaches are “true to the data” [e,f].
>
> For clinical or scientific interpretation, the choice should be guided by the question of interest: marginal references are more suitable for hypothetical or interventional reasoning, whereas conditional references better reflect associations within the observed data distribution. When dependencies are strong, grouping correlated features and interpreting them jointly can be a useful alternative [g], and extending this to survival models is direction for future work. We will include a more detailed discussion of this topic in the revised version of the paper.
>
> **Q3: Extensions to other settings.**
>
> *Non-multiplicative hazards:* SurvFD and SurvSHAP-IQ apply to any prediction function; only the theoretical results (Thm. 3.3-3.6) do not apply and require further research.
>
> *Competing risks and recurrent events:* SurvFD applies directly to cause-specific log-hazards and log-intensities (e.g., Andersen-Gill), where the additive structure preserves our theoretical results. For prediction functions such as cumulative incidence or marginal mean functions, SurvFD remains applicable, but transformation-induced interactions may arise (Prop. 3.5, Cor. 3.4).
>
> **References**
>
> [a] Effect Hierarchy, Wu & Hamada, Wiley 2009. \
> [b] Explaining Similarity in Vision-Language Encoders, Baniecki et al., NeurIPS 2025. \
> [c] Proxy-SPEX: Sparse Feature Interactions in LLMs, Butler et al., NeurIPS 2025. \
> [d] Human-Centered XAI Survey, Rong et al., IEEE TPAMI 2024. \
> [e] True to Model or Data?, Chen et al., arXiv 2020. \
> [f] A Comparative Study of Methods for Estimating Model-Agnostic Shapley Value Explanations, Olsen et al., DMKD 2024. \
> [g] Grouped feature importance and combined feature effect plots, Au et al., DMKD 2022.

---

> > ### Author Rebuttal · Reviewer_MKCW · 2026-04-01
> >
> > I acknowledge the authors' rebuttal and as my score is already on the positive side, there will be no further changes.

---

### Official Review · Reviewer_FiSB · 2026-03-12

**Soundness:** 3
**Presentation:** 2
**Significance:** 3
**Originality:** 3
**Overall Recommendation:** 4
**Confidence:** 2

**Summary:**

The paper presents Survival Functional Decomposition (SurvFD) which decomposes interactive effects into time-dependent and time-independent components. Characterization of log-hazard, hazard, and survival functions’ decomposition is provided. The paper then presents SurvSHAP-IQ that adapts Shapley interaction for time-to-event prediction.

**Compliance With Llm Reviewing Policy:**

Affirmed.

**Final Justification:**

I decided to maintain my score after reading the author's rebuttal.

**Key Questions For Authors:**

1. To capture the time-dependent and time-independent components, why wouldn’t one consider the panel data models (fix-effect/random-effect methods) that directly characterize different components? What are the differences between the proposed decomposition and those methods?

2. For higher order effect (|M|>3), would one expect a diminishing effect as |M| become larger?

3. In theoretical analysis, could the authors clarify what are the main novel technical tools used beside the well-defined Shapley interaction?

4. In Figure 7, why is regression having a large SE while other methods seem to not have any SE at all?

**Limitations:**

yes

**Strengths And Weaknesses:**

Soundness:  I am not familiar with theoretical results in this space, but the experiments look comprehensive to me.

Presentation: The presentation on survival analysis is clear, but the presentation on functional decomposition could be improved. For example, in Eq 5, $\bar{M}$ is undefined, this makes the entire subsection hard to follow (and $\bar M$ is defined later in Sec 2.3).

Significance: Characterizing the higher order effect in survival analysis is an important topic. The SurvSHAP-IQ decomposition provides a new perspective to calculate the first and second order components (individual and interaction effects). This could be of interest to practitioners in the field.

Originality: The paper is well-motivated: functional decomposition was not designed to disentangle time-dependent and time-independent components, but because of the additive nature of log hazard rate, it is natural to consider extensions.

---

> ### Author Rebuttal · Authors · 2026-03-31
>
> We thank the reviewer for the constructive feedback and for recognizing that our work addresses “an important topic”, is “well-motivated”, and offers “a new perspective” to the field that “could be of interest for practitioners”.
>
> We address all concerns below and hope our responses sufficiently clarify the issues raised.
>
> **Weaknesses**
>
> **W1: Presentation of $\bar{M}$.** Thank you for pointing this out. For clarification: $\bar{M} := P \setminus M$ denotes the complement of feature subset $M$ with respect to $P = {1, \ldots, p}$. We will move this definition to its first use in Eq. 5.
>
> **Questions**
>
> **Q1: Panel data models (FE/RE) vs. SurvFD.** Panel data methods and SurvFD address fundamentally different problems. Panel data models are designed for repeated measurements per individual, decomposing variation into individual-specific time-constant heterogeneity ($\alpha_i$) and time-varying effects:
>
> $$y_{it} = \alpha_i + x_{it}'\beta + \varepsilon_{it}$$
>
> In contrast, our setting has only one observation per individual. The time-dependency is instead reflected in the *output*. Panel data models address the question of how to account for unobserved individual heterogeneity across repeated measurements. SurvFD instead considers how a feature's effect evolves over the survival time horizon, e.g. in $G(t|x) = \beta_1 x_1 \log(t+1) + \beta_2 x_2$, the effect of $x_1$ varies with $t$ (time-dependent) while $x_2$ has a constant effect (time-independent). Thus, the feature values of each individual are measured once, only their influence on the outcome is time-dependent.
>
> **Q2: Diminishing effects for higher-order interactions.** Yes, diminishing effects of higher-order interactions are expected. The effect hierarchy and sparsity principles [a, b] suggest lower-order effects tend to dominate, though this is a heuristic backed by experience rather than a universal law.
>
> We confirm this empirically: in the nki70 dataset (Appendix B.3), we computed all $\binom{76}{2} = 2850$ pairwise interactions and no interaction ranked among the top-76 effects. We also computed three-way interactions for smaller datasets and found negligible magnitudes compared to pairwise interactions. This motivates our focus on second-order effects, though SurvSHAP-IQ is extendable to arbitrary order in principle.
>
> **Q3: Novel technical contributions.** Our aim is to bridge interaction quantification with post-hoc interpretability in survival settings—a problem not previously addressed.
>
> *1. First principled functional decomposition for survival.* We introduce SurvFD, which decomposes any survival prediction function into time-dependent and time-independent pure effects, and characterizes how this decomposition behaves across prediction scales. Thms. 3.2, 3.3, Cor. 3.4, and Prop. 3.5 establish when time-dependent effects propagate and when scale transformations induce spurious interactions, explaining why additive explanations can fail on clinical scales even when the model is additive on the log-hazard scale.
>
> *2. First Shapley interaction estimator for time-dependent outcomes.* Extending Shapley interactions to time-indexed functions requires careful treatment of the value function across time. SurvSHAP-IQ provides this extension.
>
> Further theoretical development for general probabilistic/functional outputs remains needed [c]. We view our work as an important step, providing both theory and practical tools where none existed.
>
> **Q4: Large SE for regression in Fig. 7.** The large SE only occurs at a budget of $2^6$. With $p=10$ features, there are $\binom{10}{1} + \binom{10}{2} = 55$ terms. At budget $2^6=64$, the regression system is barely overdetermined, causing high variance. At higher budgets, regression becomes most accurate. Other methods (Monte Carlo, SVARM, Permutation) show smaller SE because they are sampling-based with higher bias but lower variance at small budgets. These results are consistent with previous findings for SHAP-IQ estimation [d, e]. Thank you for pointing this out, we will extend the description of this phenomenon given additional space.
>
> **References**
>
> [a] Effect Hierarchy, Wu & Hamada, Wiley 2009.\
> [b] Sparsity Principle, Box & Meyer, Technometrics 1986. \
> [c] Shapley Compositions in the Simplex, Noé et al., 2025. \
> [d] KernelSHAP-IQ, Fumagalli et al., ICML 2024. \
> [e] SHAP-IQ, Fumagalli et al., NeurIPS 2023.

---

> > ### Author Rebuttal · Reviewer_FiSB · 2026-04-03
> >
> > I thank the authors for their response and decide to maintain the score.

---

### Official Review · Reviewer_9DhC · 2026-03-12

**Soundness:** 3
**Presentation:** 2
**Significance:** 2
**Originality:** 3
**Overall Recommendation:** 4
**Confidence:** 3

**Summary:**

The paper tackles the functional decomposition of the risk, hazard, and survival functions for survival problems. First, it proposes the survival-function decomposition SurvDF for both the risk and the survival function and shows under which conditions the dependent and independent feature sets coincide. Second, it proposes Shapley interactions for time-dependent survival outcomes.

**Compliance With Llm Reviewing Policy:**

Affirmed.

**Final Justification:**

Score raised from weak reject to weak accept.

**Key Questions For Authors:**

- I noticed that the synthetic data do not take censoring into account (i.e., no censored data were generated). How does that affect the evaluation?
(\lambda) is mentioned in line 300 only once, without stating its purpose.
- Although the evaluation uses real data, I feel that the reported results have a very narrow scope and are not backed by findings from the biological literature.
- It was stated that the full results on the synthetic data can be found in Figures B.2–B.5; however, I could not find the one for Attribution (h(t|x)).
- I may be wrong, but in the first column of Figure 3 the diff seems always parallel to (x_1). Does that suggest something is wrong? The odd thing is that in the top figure, (x_1) is time-dependent. Similar behavior appears in the bottom figure where (x_1) is time-independent.
- Why do all lines in Figure 6 start at attribution equal to zero at time 0 on the whole validation set? Also, why does the age have both negative and positive attributions in the first column? That would mean that for some individuals the risk increases with age while for others it decreases with age. I feel that such points are missing in the manuscript.
- I am wondering why you chose DeepHit for TCGA-BRCA, since it is better suited to competing risks.
- On high-dimensional data: although the paper discusses scalability, it does not discuss limitations of the proposed approach on high-dimensional survival data such as TCGA mRNA (~60k genes).

**Limitations:**

Yes

**Strengths And Weaknesses:**

Strength:
- Addressing time-dependent interactions in survival analysis is a problem of high importance in the field.
- The paper backs almost every step with theoretical justifications and proofs.
Weakness:
- Sometimes the written claims are hard to follow and to attribute. For example, in the introduction you state: "Moreover, although additive decomposition is natural on the log-hazard scale, transformations to interpretable scales, such as hazard or survival functions, induce additional time-dependent effects and interactions not present on the log-hazard scale." It was not clear whether this is an already known result or something that is proven and discussed later in the text.
- Although the paper is theoretically justified, some parts are difficult to follow, especially those related to the SHAP computation—for instance, the definition of the function (v) is omitted in the main text and only given in the appendix, which makes the presentation less comprehensible for non-expert readers. The same applies to the use of (\bar{M}) in Eq. (5) without defining it.
- After reading the paper more than twice and being familiar with the topic, I am still unsure how to apply the proposed decompositions to a new method. This points to a lack of clear guidance on how the decomposition can be applied to new methods.

---

> ### Author Rebuttal · Authors · 2026-03-31
>
> We sincerely thank the reviewer for their time and for recognizing that our work addresses a "problem of high importance in the field" and is theoretically well justified. We address each concern below and hope that our responses adequately resolve the questions raised.  If the reviewer finds these responses satisfactory, we would appreciate a reconsideration of the evaluation.
>
> **Weaknesses**
>
> **W1-2: Presentation issues.** We appreciate these suggestions. For the camera-ready version, we will: (1) clarify that transformation-induced interactions are our novel contribution (Prop. 3.5, Cor. 3.4), (2) define $\bar{M} := P \setminus M$ at first use in Eq. 5, (3) include the value function definition in the main text.
>
> **W3: Applicability to new methods.** SurvFD provides the theoretical foundation for SurvSHAP-IQ (Sec. 3.2, A.6). SurvSHAP-IQ is model-agnostic and applies to any survival model outputting prediction functions, yielding decompositions up to order $k$, as demonstrated with CoxPH, GBSA, RSF, and DeepHit. No adaptation is required for new models. Code is available as a supplement on GitHub.
>
> **Questions**
>
> **Q1: Censoring in synthetic data.** Censoring is applied: all observations with $t^{(i)} \geq 70$ are right-censored (Sec. 4.1, line 302). The parameter $\lambda = 0.03$ is the baseline hazard rate $h_0(t)$ in Eq. 3. We will clarify this in the revision.
>
> **Q2: Scope & biological backing.** Our evaluation spans multiple clinical domains and data modalities: uveal melanoma survival, AIDS diagnosis, breast and lung cancer survival, and multi-modal TCGA-BRCA combining images with clinical variables. Our real-world analyses connect to clinical literature. For uveal melanoma (Fig. 6), age, tumor diameter, and mitotic rate are established prognostic factors [a]. For TCGA-BRCA, the interaction between node-negative status and lumpectomy reflects the known favorable early-stage prognosis. We will strengthen these connections in the revision.
>
> **Q3: Missing hazard attribution figure.** Thank you—Fig. B.1 shows the hazard attributions. We correct the reference.
>
> **Q4: Fig. 3: $x_1$ parallel to diff.** This is expected behavior, not an error. The difference curve (dashed grey) represents individual minus mean prediction over the full data. By the efficiency axiom of Shapley values, the sum of all attribution curves exactly equals this difference. In scenario (9) (top left plot in Fig. 3), $x_1$ is the dominant time-dependent effect, so its attribution largely drives the shape of the difference curve; the other effects contribute minimally, making $x_1$ appear parallel to the diff. In scenario (10) (lower left plot in Fig. 3), $x_1$ is defined as time-independent in the log-hazard ground-truth $G(t|x)$, but the estimated effect absorbs part of the time-dependent interaction $x_1 x_3$ through downward propagation (Thm. 3.3). As a result, the difference curve is flatter than that of $x_1$ due to the opposing behavior of the $x_1 x_3$ interaction, and is therefore less parallel to $x_1$ than in scenario (9).
>
> **Q5: Fig. 6: Zero attribution at $t=0$ and age effects.** At $t=0$, the survival probability equals 1 for all individuals, so $S(0|x) - \mathbb{E}[S(0|X)] = 0$, making all attributions 0 by construction. Regarding age: the upper plot shows Shapley values (k = 1), where the effect of age varies across individuals due to interactions, tending to be positive at lower ages (blue) and negative at higher ages (red). The middle plot (k = 2) isolates main effects, reducing variance, while the lower plot shows interaction effects. We add further interpretation in the revision.
>
> **Q6: DeepHit.** DeepHit directly predicts discrete-time survival probabilities without assuming multiplicative hazards, making it well-suited for multi-modal deep learning where hazard assumptions may not hold. While DeepHit supports competing risks, it is not restricted to them [b].
>
> **Q7: High-dimensional scalability.** We acknowledge that ~60k features remains challenging and will discuss this limitation more explicitly in the final paper. Our nki70 experiment (76 features, 2,850 interactions) used efficient approximation achieving polynomial complexity. Recent advances scaling Shapley interactions to vision-language models and LLMs [c,d] suggest promising directions. For genomics, dimensionality reduction or pathway-level aggregation before explanation may be practical.
>
> We hope these clarifications address the reviewer's concerns and demonstrate that the paper makes solid contributions. We are committed to improving our work in the revision.
>
> **References**
>
> [a] Machine Learning Models for Uveal Melanoma Prognosis, Donizy et al., European Journal of Cancer 2022.\
> [b] DeepHit: A Deep Learning Approach to Survival Analysis with Competing Risks, Lee et al., AAAI 2018.\
> [c] Explaining Similarity in Vision-Language Encoders, Baniecki et al., NeurIPS 2025.\
> [d] Proxy-SPEX: Sparse Feature Interactions in LLMs, Butler et al., NeurIPS 2025.

---

> > ### Author Rebuttal · Reviewer_9DhC · 2026-04-02
> >
> > I am increasing my score, but I sincerely hope that the presentation issues will be addressed and improved in the revised version.

---

> > > ### Author Response · Authors · 2026-04-07
> > >
> > > We thank the reviewer for the careful re-evaluation of our manuscript and for increasing their score. We are glad that our rebuttal helped clarify the contributions and address the concerns. In the revised version, we will incorporate all the improvements outlined in the rebuttal, including addressing the presentation issues.

---

### Official Review · Reviewer_mTic · 2026-03-16

**Soundness:** 3
**Presentation:** 4
**Significance:** 3
**Originality:** 3
**Overall Recommendation:** 5
**Confidence:** 3

**Summary:**

This paper addresses a gap in interpreting machine learning survival models: existing explanation methods quantify individual feature attributions but cannot capture feature interactions. The authors propose SurvFD, a framework that decomposes survival predictions into time-dependent and time-independent interaction effects, and SurvSHAP-IQ, a practical Shapley-based estimator built on top of it. The methods are validated on simulated data and applied to real-world clinical datasets, including a multi-modal deep learning model combining histopathology images with clinical features.

**Compliance With Llm Reviewing Policy:**

Affirmed.

**Key Questions For Authors:**

The authors only showed the method’s results on GBSA/RSF models across the main manuscript and supplemental material. Can the authors demonstrate how SurvFD behaves on models that are inherently interpretable?

What quantitative metrics can be incorporated into the study to make statistical comparisons with other methods? This is quite a broad question, but finding even a single metric would strengthen this manuscript significantly.

**Limitations:**

No... just as the authors state: "There are many potential societal consequences of our work, none which we feel must be specifically highlighted here."

The authors can freely state how their specific design choices have societal impact without taking away from their technical novel contributions.

**Strengths And Weaknesses:**

The paper is very well-written by the authors and easy to follow. They summarized the landscape of related literature in a concise, yet precise manner. The authors also state their novel contributions to the field very clearly. The empirical validation of both synthetic and real data on machine learning and deep learning models presented convincing qualitative evidence.

As is a common difficulty with most model interpretability papers, the authors did not define any quantitative metrics for SurvFD’s success. The authors also successfully explored the limitations of their method, in both model design (multiplicative hazards) and study design (second-order effects / <70 feature datasets)

---

> ### Author Rebuttal · Authors · 2026-03-31
>
> We thank the reviewer for their thoughtful feedback and positive assessment of the paper by noting that the paper is “very well-written”, provides a “concise, yet precise” overview of the literature, clearly states its novel contributions, presents “convincing qualitative evidence” through empirical validation, and “successfully” explores key limitations in both model and study design.
>
> In the following we address the remaining questions.
>
> **Questions**
>
> **Q1: Application to inherently interpretable models.** The definition of SurvFD does not depend on whether the underlying prediction model is inherently interpretable or a black box model. The approach is model-agnostic and uniquely defined based on the chosen reference distribution.
>
> At the same time, we agree that inherently interpretable models can make validation more straightforward, as they provide an implicit model-based  ground truth. For this reason our experimental setup already includes results on the Cox Proportional Hazards (CoxPH) model (Figs. 4 and B.4). A key finding is that even a basic CoxPH model without explicit interaction terms exhibits non-zero interaction effects under SurvFD (Prop. 3.5), because the transformation from the additive log-hazard scale to the multiplicative survival scale inherently induces interactions. We believe this is a practically important insight for anyone interpreting survival models.
>
> **Q2: Quantitative metrics for comparison.** This is a valid concern and, as correctly pointed out by the reviewer, a general problem in interpretability papers. We want to address it from multiple angles.
>
> First, we emphasize that we introduced a comprehensive set of evaluation metrics to validate both the theoretical properties of SurvFD and the correctness of its empirical estimator, SurvSHAP-IQ. (1) *Local accuracy* (Eq. B.35-B.36, Table B.3, Fig. B.6), which measures how well SurvSHAP-IQ reconstructs the original prediction, confirming near-perfect decomposition quality across all scenarios; (2) *Ground-truth recovery* (Sec. 4.1), verifying correct identification of time-dependent vs. time-independent effects on synthetic data with known interaction structures (Thms 3.2–3.3); and (3) *Approximation error* (Figs. 7, B.12), benchmarking approximation algorithms against exact computation.
>
> However this is solely internal validation. We agree with the reviewer that comparing with other IML methods is inherently challenging. SurvSHAP-IQ is, to our knowledge, the first method to estimate interaction effect curves in survival models. Existing methods, like SurvLIME [a], SurvSHAP(t) [b], and GradSHAP(t) [c] target only first-order attributions. This makes direct method-to-method comparison infeasible, as there is simply no baseline to compare against. More broadly, quantitative comparison in interpretable machine learning is inherently difficult because different explanation methods often measure fundamentally different quantities. As discussed by Bordt & von Luxburg [d] and Chen et al. [e], even methods that appear similar (e.g., interventional vs. observational SHAP) answer different questions and are "correct" with respect to different ground truths. Krishna et al. [f] further demonstrate that disagreement between explanation methods does not necessarily indicate that one is wrong – they may simply be explaining different aspects of model behavior. This makes unified benchmarking across IML methods a largely open problem.
>
> **Limitations**
>
> As suggested by the reviewer, we will also extend the discussion of limitations in Sec. 5 and the broader impact of our work in the final version of the paper.
>
> **References**
>
> [a] SurvLIME: A Method for Explaining Machine Learning Survival Models, Kovalev et al., Knowledge-Based Systems 2020.\
> [b] SurvSHAP(t): Time-dependent Explanations of Machine Learning Survival Models, Krzyziński et al., Knowledge-Based Systems 2023.\
> [c] Gradient-based Explanations for Deep Learning Survival Models, Langbein et al., ICML 2025.\
> [d] From Shapley Values to Generalized Additive Models and back, Bordt & von Luxburg, AISTATS 2023.\
> [e] True to the Model or True to the Data?, Chen et al., arXiv 2020.\
> [f] The Disagreement Problem in Explainable Machine Learning: A Practitioner's Perspective, Krishna et al., TMLR 2024.

---

### Decision · Program_Chairs · 2026-04-30

**Decision:**

Accept (regular)

**Comment:**

After the author/reviewer discussion, the reviewers unanimously leaned toward acceptance, and in reading over the discussion, I'm in favor of recommending the paper for acceptance as well. I recommend that the authors follow through with the promised edits of improving presentation clarity and being more thorough with the discussion of limitations.